# Coevolutionary transitions from antagonism to mutualism explained by the Co-Opted Antagonist Hypothesis

Christopher A. Johnson [1,2,3✉], Gordon P. Smith [1,4], Kelsey Yule [1,5], Goggy Davidowitz[6], Judith L. Bronstein[1] & Régis Ferrière[1,7,8]

There is now good evidence that many mutualisms evolved from antagonism; why or how, however, remains unclear. We advance the Co-Opted Antagonist (COA) Hypothesis as a general mechanism explaining evolutionary transitions from antagonism to mutualism. COA involves an eco-coevolutionary process whereby natural selection favors co-option of an antagonist to perform a beneficial function and the interacting species coevolve a suite of phenotypic traits that drive the interaction from antagonism to mutualism. To evaluate the COA hypothesis, we present a generalized eco-coevolutionary framework of evolutionary transitions from antagonism to mutualism and develop a data-based, fully ecologically-parameterized model of a small community in which a lepidopteran insect pollinates some of its larval host plant species. More generally, our theory helps to reconcile several major challenges concerning the mechanisms of mutualism evolution, such as how mutualisms evolve without extremely tight host fidelity (vertical transmission) and how ecological context influences evolutionary outcomes, and vice-versa.

[1] Dept. of Ecology and Evolutionary Biology, University of Arizona, P.O. Box 210088, Tucson, AZ, USA. [2] Institute of Integrative Biology, ETH Zürich, Universitäetstrasse 16, Zürich, Switzerland. [3] Dept. of Ecology and Evolutionary Biology, Princeton University, 106a Guyot Hall, Princeton, NJ, USA. [4] Dept. of Neurobiology and Behavior, Cornell University, 215 Tower Road, Ithaca, NY, USA. [5] Biodiversity Knowledge Integration Center, Arizona State University, 734W Alameda Drive, Tempe, AZ, USA. [6] Dept. of Entomology, University of Arizona, 1140 E. South Campus Dr., Tucson, AZ, USA. [7] Institut de Biologie de l'ENS (IBENS), École Normale Supérieure CNRS UMR 8197, 46 rue d'Ulm, Paris, France. [8] iGLOBES International Research Laboratory, École Normale Supérieure, Université Paris Sciences & Lettres CNRS UMI 3157, University of Arizona, 845N Park Avenue, Tucson, AZ, USA. ✉email: cjohns21@uw.edu

Mutualisms play essential roles in the generation of biodiversity[1]; yet, the mechanisms driving mutualism evolution remain enigmatic[2]. While many mutualisms evolved de novo via the exchange of costless by-products, many other mutualisms are thought to have originated from antagonistic interactions[3]. Several studies have elucidated ecological or evolutionary conditions driving transitions from mutualism to antagonism (mutualism breakdown)[4,5], yet limited theory has been developed to mechanistically explain the converse, evolutionary transitions from antagonism to mutualism (but see[6,7]). Models of virulence predict evolution of reduced antagonism, perhaps toward mutualism, in cases of parasitism where host fidelity is high and availability of alternate hosts is low[8,9]. In these models, host fidelity is achieved when lineages of parasites and their host are tightly linked through vertical transmission from parent to offspring. Compelling evidence for the 'virulence theory' comes from interactions involving microbial parasites[10,11]. The fact that horizontal transmission of mutualistic partners is common in nature, however, is a major challenge to virulence theory[12], leaving the mechanisms of mutualism largely unresolved for some of the most conspicuous partnerships on Earth.

We develop a general eco-coevolutionary framework for investigating the evolutionary transition from antagonism to mutualism and use data-based modeling to advance and evaluate a new hypothesis: the Co-Opted Antagonist (COA) Hypothesis, whereby co-option of an antagonistic species to perform a beneficial function changes the ecology of the interaction and, simultaneously, the interaction evolves to net mutualism. We evaluate the COA by studying plant-insect interactions in which insects are pollinators and herbivores of the same plant species[13–15]. This type of interactions may be surprisingly common in nature; for example, over half of Lepidoptera in a central European community (54% of 995 species) nectar at (and thus potentially pollinate) at least one of their larval host plants[16]. These observations lead us to posit that interactions between some plants and insects may have transitioned from pure herbivory to pollination as plants evolved to co-opt insect search behavior for host plants to disperse their gametes.

To evaluate the COA hypothesis we develop data-based eco-coevolutionary models for a community in which a lepidopteran insect initially oviposits on, but does not pollinate, its larval host plants, which are pollinated by other species. Evolutionary co-option of the antagonist is achieved if plant traits evolve in a way that allows the antagonist to pollinate it (e.g., larger flowers shaped so as to allow the insect to nectar and transfer pollen). The evolutionary integration of two consumer-resource interactions, herbivory and nectar-feeding (with pollination as a by-product) may achieve the tight coupling required to induce strong selective pressures on plants (beyond those induced by other pollinators) and the insect. This coupling may in turn create the ecological conditions for plant-insect coevolution to drive pollination benefits beyond trophic costs, thus assimilating the herbivore as a pollinator (either alongside or replacing other pollinators). From the COA hypothesis, we predict that community context, in particular the degree of specialization of the herbivory and nectar-feeding interactions, should influence the transition: generalist larvae that can feed on multiple host plant species may facilitate the transition by ameliorating herbivory costs, whereas generalist adults feeding at multiple nectar sources may hinder the transition by amplifying oviposition costs.

## Results

We develop an eco-coevolutionary framework of evolutionary transitions from antagonism to mutualism (Methods), which we adapt to derive a fully-ecologically parameterized eco-coevolutionary model for a small community involving the hawkmoth *Manduca sexta* and its associated plant species in southern Arizona, USA (Box 1; Fig. 1). This community is an excellent system to evaluate the COA hypothesis because—while relatively specialized compared to most plant-pollinator or plant-herbivore communities in nature—it involves multiple plant species and is small enough for tractability. This allows us to test the effects of alternative trophic interactions (larval host plants and nectar sources) on the transition to mutualism and cast the COA hypothesis in slightly more general terms as a step towards studying full communities.

We link population dynamics and trait evolution via the adaptive dynamics framework[17,18]. We study four scenarios involving *M. sexta* and: (1) each *Datura* species separately, (2) both *Datura* species together, (3) each *Datura* species and an alternative larval host plant, and (4) each *Datura* species and an alternative nectar source. In each scenario, an ancestral insect oviposits on, but does not pollinate, ancestral *Datura* species $i$ and nectar feeds at other plants. We model coevolution of plant traits, such as flower size ($x_i^B$), that allow the insect to pollinate it and subsequently increase pollination benefits (seed set of pollinated flowers, $b_i$) as well as plant and insect traits driving attraction and defense. Attraction entails coevolution of plant traits such as the production of volatiles ($x_i^V$) that attract the insect and insect traits such as its sensitivity to olfactory cues ($y_i^V$) to locate larval host plants. In the model, coevolution of attraction (via $x_i^V$ and $y_i^V$) changes the visitation rate, $v_i$, which affects both nectaring and oviposition[19]. Defense entails coevolution of plant traits such as the production of chemical defenses ($x_i^H$) and insect traits such as increasing tolerance to plant secondary compounds ($y_i^H$). Coevolution of defense in the model (via $x_i^H$ and $y_i^H$) affects the herbivory rate, $h_i$. In the model, plant traits trade off with competitive ability (e.g., mutant plants with highly-attractive flowers or strong chemical defenses have low competitive ability) and insect traits trade off with oviposition (e.g., mutant insects with high sensitivity to olfactory cues or high tolerance to plant defenses produce fewer eggs).

**Ancestral antagonism.** An ancestral insect persists as a pure antagonist of its exclusive larval host plant species $i$ ($b_i = 0$) when:

$$f_i s_i > 1 \tag{1}$$

where $f_i$ is insect lifetime fecundity and $s_i$ is larval success (probability of maturing rather than dying) on plant species $i$ (Methods). This defines the interaction breakdown boundary (dashed gray lines in Fig. 2) below which the ancestral insect is

**Box 1**

We parameterize our eco-coevolutionary model using the hawkmoth *Manduca sexta* (Sphingidae) and its associated plant species in southern Arizona, USA, a relatively specialized study system (Fig. 1). In southern Arizona, *M. sexta* both pollinates and oviposits (lays eggs) on *Datura wrightii* and *D. discolor* (Solanaceae)[19,30], which are co-blooming congeners[25]. *D. wrightii* flowers are highly attractive to moths[20] whereas *D. discolor* flowers are much less fragrant and less rewarding[25,26]. Both *Datura* species are highly self-compatible, but moth-pollinated flowers set several times more seeds than do autonomous self-pollinated flowers (Supplementary Data 1;[30]). *M. sexta* is the primary pollinator of *D. wrightii*[40], but a single larva can completely defoliate its host plant[41]. In this region, *M. sexta* also oviposits on, but does not pollinate, *Proboscidea parviflora* (Martyniaceae)[21,34] and nectar-feeds at, but does not oviposit on, the bat-pollinated *Agave palmeri* (Asparagaceae) and a limited range of other species[40].

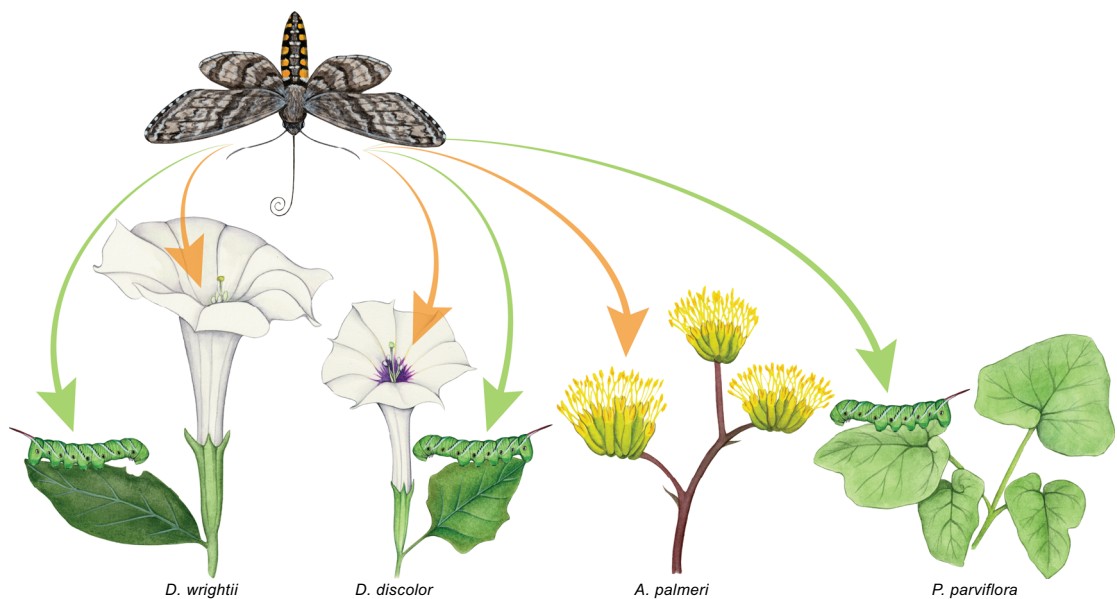

**Fig. 1 Study system involving the hawkmoth *Manduca sexta* and its associated plants in southern Arizona, USA.** *M. sexta* nectar-feeds (orange arrows) and oviposits (green arrows) at *D. wrightii* and *D. discolor*. The outcome of these interactions could be affected by *M. sexta* interactions with the alternative nectar source, *A. palmeri*, or the alternative larval host plant, *P. parviflora*. Paintings by Julie Johnson (Life Science Studios).

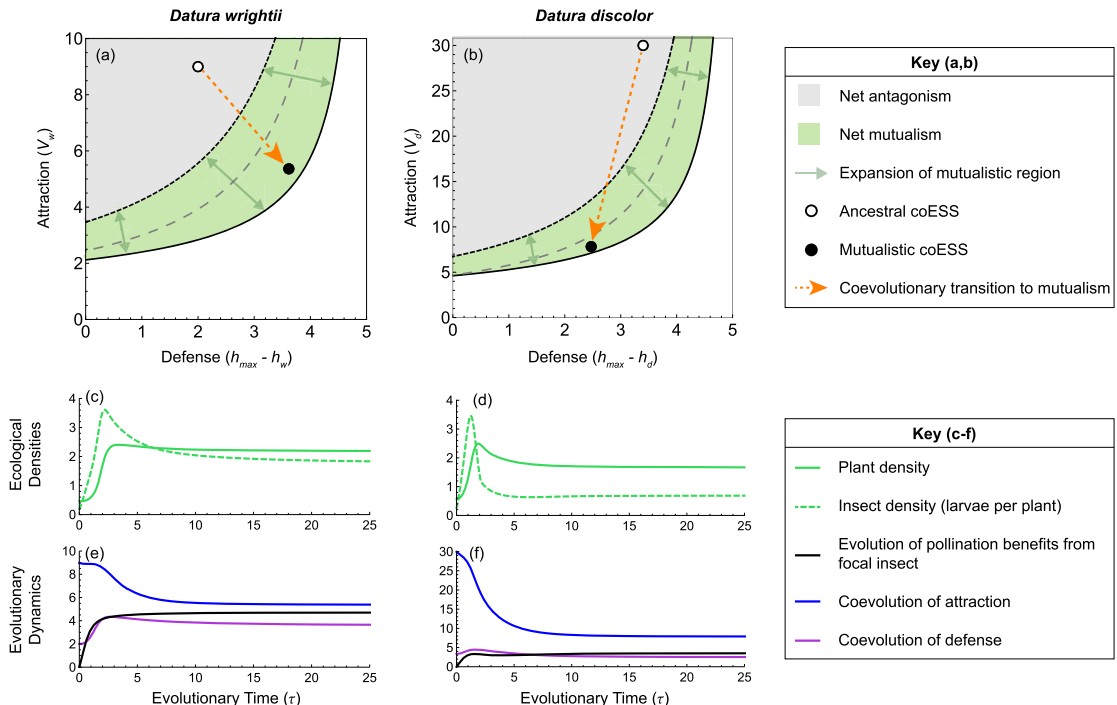

**Fig. 2 Coevolution from antagonism to mutualism in one-plant species communities.** Panels (**a**, **b**) show interaction outcomes as a function of attraction ($v_i$) and defense ($h_{max} - h_i$), where $h_{max}$ is the maximum herbivory rate (Methods). Parameter space regions depict ecological outcomes and arrows highlight coevolutionary effects. The ancestral insect persists as a pure antagonist above the dashed gray line (Eq. 1) and is extinct below. Co-option of the antagonist as a pollinator and the evolution of pollination benefits ($b_i > 0$) expands the green mutualistic region by moving the interaction breakdown boundary (Eq. 3; solid black line) away from the interaction transition boundary (Eq. 2; dotted black line), as depicted by the green arrows (see Supplementary Movies 1 and 2). The insect persists as a net antagonist despite being co-opted as a pollinator within the gray regions and goes extinct within the white regions. Simultaneous to the evolution of pollination benefits, coevolution of attraction and defense drives the transition to net mutualism, as depicted by the orange arrows pointing from the ancestral coESSs (white points) to the new coESSs (black points). Empirical estimates of the coESSs are not included here (as in Fig. 3) due to data limitations. Panels (**c**, **d**) plot the equilibrium densities of each plant species (solid green lines) and insect larvae per plant of each *Datura* species (dashed green lines) over evolutionary time, $\tau$. Panels (**e**, **f**) plot the coevolutionary dynamics of pollination benefits ($b_i$; black lines), attraction ($v_i$; blue lines), and defense ($h_i$; purple lines).

extinct. For an ancestral insect with similar trait values as *M. sexta* to persist with ancestral plant species with similar trait values as *D. wrightii* or *D. discolor*, the coevolved visitation and herbivory rates (white points in Fig. 2) must be above the interaction breakdown boundary.

**Ecological conditions for the transition from antagonism to mutualism.** An interaction involving an insect that pollinates its exclusive larval host plant species $i$ ($b_i > 0$) is mutualistic when:

$$f_i s_i < 1 \qquad (2)$$

*Methods.* Equation (2) is the interaction transition boundary (dotted black lines in Fig. 2, which replace the dashed gray lines from the ancestral interaction) above which the interaction is net antagonistic (gray regions of Fig. 2) and below which the interaction is net mutualistic (green regions of Fig. 2). Equation (2) reveals an inherent conflict of interest between plant and insect: high insect life-time fecundity ($f_i$) and larval success ($s_i$) benefit the insect but not the plant, and vice-versa. Equation (1) now gives the invasion criterion above which the insect can increase from low density and persist as a net antagonist of the plant despite being co-opted as a pollinator because its herbivory costs to the plant exceed its pollination benefits.

The new interaction breakdown boundary (solid black lines in Fig. 2) below which the insect always goes extinct (white regions of Fig. 2) is now approximated (Methods) by:

$$f_i s_i (1 + b_i) > 1 \qquad (3)$$

Equation (3) shows that pollination benefits, $b_i$, effectively create a parameter region in which the insect persists as a net mutualist by buoying plant density and thus lowering the interaction breakdown boundary (Eq. 3 vs. Eq. 1). Within this mutualistic region, however, the insect cannot invade from very low density because, when the insect is rare, it cannot buoy plant density sufficiently to maintain a positive per capita growth rate (mathematically, the insect's invasion criterion—Eq. 1—cannot hold when Eq. 2 is satisfied). The mutualistic region is therefore characterized by bistability (see Supplementary Figure 1) and an Allee threshold above which the insect persists as a net mutualist of the plant and below which it goes extinct.

The interaction transition boundary (Eq. 2) is the exact reverse condition of the interaction breakdown boundary in the ancestral interaction (Eq. 1). Thus, co-option of the antagonist as a pollinator (i.e., $b_i > 0$) does not itself create mutualism because the phenotype of any viable ancestral interaction remains in the antagonistic regions of Fig. 2 following co-option. Integration of the antagonist as a pollinator, however, fundamentally changes the ecology of the interaction by making mutualism possible (within the green regions of Fig. 2 defined by Eqs. 2 and 3) and by reshaping the evolution of plant and insect traits governing the outcome of the interaction by changing their underlying selection gradients (Methods).

Given a herbivorous insect that can pollinate, the evolutionary transition to mutualism requires that selection yields a coevolutionary stable state (coESS) at which pollination benefits exceed herbivory costs for the plant (within the green regions of Fig. 2). We model simultaneous coevolution of three responses: pollination (via $x_i^B$, affecting pollination benefits from the insect, $b_i$), attraction (via $x_i^V$ and $y_i^V$, affecting visitation rate, $v_i$) and defense (via $x_i^H$ and $y_i^H$, affecting herbivory rate, $h_i$). The question becomes: can coevolution drive the system across the interaction transition boundary from antagonism to mutualism?

**Coevolutionary transitions from antagonism to mutualism.** Plant evolution of pollination benefits from the antagonist ($b_i$) creates and expands the green mutualistic regions of Fig. 2, as depicted by the green arrows (Supplementary Movies 1 and 2). Simultaneous to the evolution of pollination benefits, coevolution of attraction and defense leads to a coESS at which the interaction has transitioned to net mutualism, as depicted by the orange arrows in Fig. 2 pointing from the ancestral coESS (white points) to the new coESS (black points). This is true irrespective of ancestral trait values: for any initial trait values in the ancestral interaction (above the dashed gray lines in Fig. 2), coevolution of pollination benefits, attraction, and defense following the co-option of the antagonist as a pollinator leads the interaction to mutualism. Coevolution, in turn, affects the ecology of the interactions. As pollination benefits, attraction and defense coevolve (Fig. 2e, f; Supplementary Fig. 2), plant and insect densities increase such that both species attain greater equilibrium density following the evolutionary transition to mutualism (Fig. 2c, d).

The same results apply to a model in which both plant species co-occur (Fig. 3) and thus coevolve with each other and the antagonist. We model this case based on data from our experiments involving both *D. wrightii* and *D. discolor*[19]. Comparing the predicted coESS with independent data provides a strong validation of the model—at least of the selection gradients on attraction and defense around the contemporary state. The coESS for attraction and defense predicted by the model (black points in Fig. 3) are well within the standard errors our empirical estimates (blue crossbars in Fig. 3). Evolution leads to a coESS that would have driven evolutionary purging of the insect in either one-plant species community (i.e., the coESS in Fig. 3 would be in the extinction regions of Fig. 2), underscoring the importance of multiple larval host plants for the system's eco-evolutionary stability. Coevolutionary dynamics cycle transiently around the coESS (Fig. 3e, f) due to varying selective pressures on each plant species and the antagonist, driving slight oscillations in the ecological equilibria (Fig. 3c, d).

The predicted coESS for pollination benefits from the antagonist (seed set of flowers pollinated by *M. sexta*) and oviposition efficiency (average number of eggs laid by a female *M. sexta* after a floral visit) are also within the standard errors of our empirical estimates (Supplementary Table 3). The presence of multiple *Datura* species has other notable qualitative and quantitative effects on evolutionary outcomes: both species evolve lower attraction (lower $v_i$) and greater defense (lower $h_i$) than they do in a one-plant species community (Fig. 2). Intriguingly, the *Datura* species evolve different strategies in response to *M. sexta*: *D. wrightii* evolves high attraction and defense (high $v_w$, low $h_w$), while *D. discolor* evolves low attraction and defense (low $v_d$, high $h_d$). This difference arises because *D. wrightii* receives greater pollination benefits, but is also preferentially visited—and oviposited on—by *M. sexta* (both in the model and field; Supplementary Data 1). *D. wrightii* therefore differentially benefits from *M. sexta* pollination, but must evolve relatively high defense, while *D. discolor* mitigates herbivory costs by evolving low attraction, and therefore requires relatively low defense.

**Coevolutionary transitions in more generalized trophic interactions.** We now assess how alternative plant species that either partition the costs or share the benefits of the insect influence evolutionary transitions from antagonism to mutualism. Our study system (Box 1) is well suited because it is small enough to enable model parameterization and involves both an alternative larval host plant and an alternative nectar source. Recent *M. sexta* trophic generalization to *P. parviflora* and *A. palmeri* are well documented in nature[20,21].

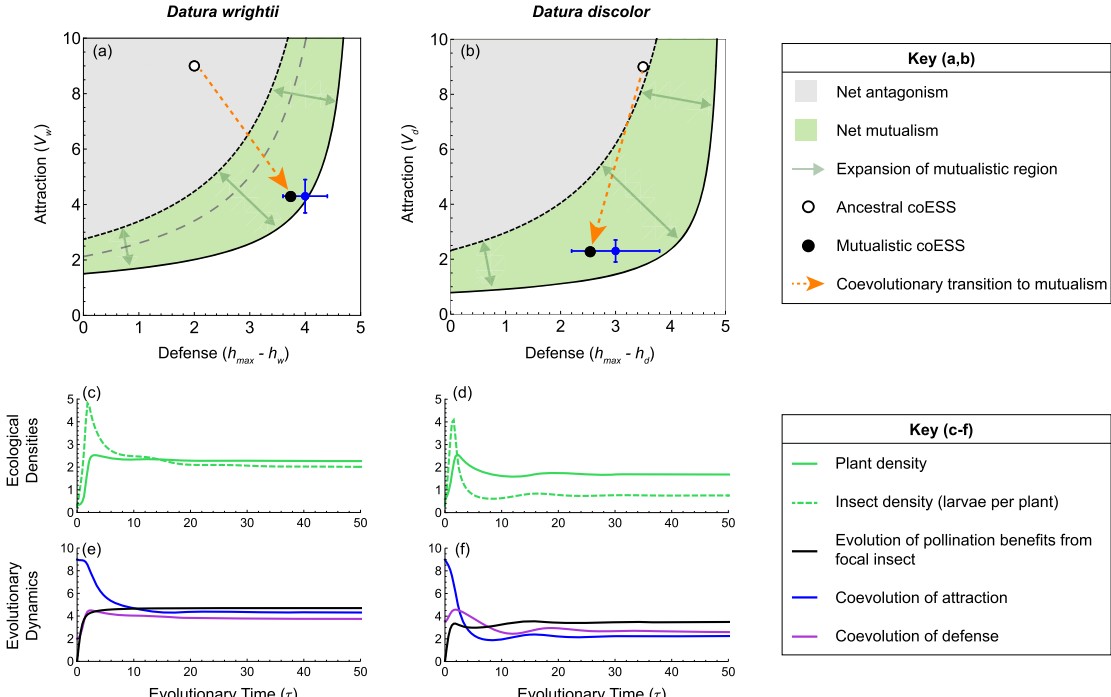

**Fig. 3 Coevolution from antagonism to mutualism in the two-plant species community.** Panels (**a**, **b**) show interaction outcomes as a function of attraction ($v_i$) and defense ($h_{max} - h_i$), where $h_{max}$ is the maximum herbivory rate. The ancestral insect persists as a pure antagonist above the dashed gray line in panel **a** and is extinct below. There is no dashed gray line in panel **b** because the ancestral insect can persist on the ancestral *D. wrightii* alone. The insect persists as a net antagonist within the gray regions and is extinct within the white regions. Plant evolution of pollination benefits ($b_i > 0$) expands the green mutualistic regions by separating the interaction breakdown boundary (solid black line) and the interaction transition boundary (dotted black line), as depicted by green arrows. Simultaneously, coevolution of attraction and defense drives the transition to net mutualism, as depicted by orange arrows from ancestral coESS (white points) to the new coESS (black points). Blue points give empirical estimates of the coESS (panel (**a**): $h_w = 1 \pm 0.4$; $v_w = 4.3 \pm 0.6$; panel (**b**): $h_d = 2 \pm 0.8$; $v_d = 2.3 \pm 0.4$), where the crossbars show variation in leaf consumption[30] and the standard error of floral visitation[19] ($n = 89$ plants for *D. wrightii*; $n = 33$ plants for *D. discolor*) (Methods). While the plant species coevolve in the model (see Supplementary Movie 3), panels (**a**, **b**) are plotted with the other plant species held at its final coESS for clarity. Panels (**c**, **d**) plot the equilibrium densities of each *Datura* species (solid green lines) and insect larvae per plant of each *Datura* species (dashed green lines) over evolutionary time, $\tau$. Panels (**e**, **f**) plot the coevolutionary dynamics of pollination benefits ($b_i$; black lines), attraction ($v_i$; blue lines) and defense ($h_i$; purple lines).

When each *Datura* species and the alternative larval host plant coevolve with *M. sexta*, mutualism can evolve over a large parameter region (Fig. 4a, b) because the alternative larval host plant effectively subsidizes some of the costs to the *Datura* species associated with oviposition. Conversely, the presence of the alternative nectar source narrows the parameter region in which mutualism can arise (Fig. 4c, d) by increasing oviposition and amplifying the antagonistic component of the interactions. In both scenarios, coevolution (Fig. 4i–l) increases the equilibrium densities of both *Datura* species and *M. sexta* (Fig. 4e–h).

## Discussion

We advance and evaluate the Co-Opted Antagonist (COA) Hypothesis as a mechanism for explaining the coevolutionary transition from antagonism to mutualism. The COA mechanism is an eco-coevolutionary process whereby natural selection favors the co-option of an antagonist to perform a beneficial function and coevolution of the interacting species drives the interaction from antagonistic to net mutualistic. The transition involves a counterintuitive evolutionary route through increased interactions with an antagonist. This is an example of a dangerous liaison[22] in which there are potential benefits of an interaction, but also substantial costs. Evolutionary co-option of the antagonist requires the evolution of traits, such as changes in floral structure or phenology in plant species, that allow an antagonist to function as a mutualist and subsequently increase mutualistic benefits obtained from the co-opted antagonist. We model

coevolution of mutualistic benefits and two other responses that drive a transition from antagonism to net mutualism: attraction, which determines the strength of the interaction; and defense, which reduces the costs incurred by the species due to interacting with the antagonist. These traits may have evolved initially in response to other species, but are driven by selective pressures imposed predominantly by the antagonist.

To evaluate the COA hypothesis, we develop a general eco-coevolutionary framework that we adapt to derive a data-based model of a small community in which a herbivorous insect pollinates its larval host plants (Box 1). Very few models focus on interactions in which an insect is both a pollinator and a herbivore of the same plant species (e.g. 23,24), despite evidence that these interactions may be surprisingly common in nature[16]. Our model is based on the well-studied *Manduca-Datura* system and is designed to be parameterized with data from empirical studies. Our eco-coevolutionary model reveals how evolutionary transitions from antagonism to net mutualism can occur via the COA hypothesis for systems embodying this biology.

For both *Datura* species, evolution drove a transition from antagonism to net mutualism in both the one- and two-plant species communities despite tightly-coupled interactions with a voracious herbivore. Importantly, the adapted trait values predicted by the model are statistically indistinguishable from our empirical estimates (Supplementary Table 3). The model is further supported by other predictions that align with empirical findings. The model predicts that *D. wrightii* evolves greater

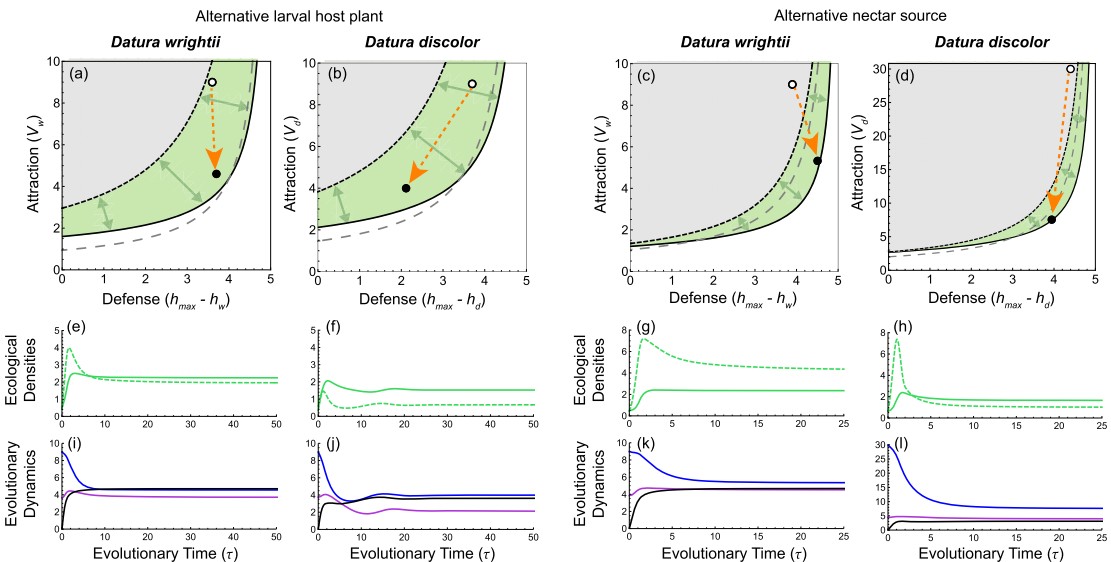

**Fig. 4 Coevolutionary transitions in more generalized trophic interactions.** Interaction outcomes in the presence of an alternative larval host plant (panels **a**, **b**) or an alternative nectar source (panels **c**, **d**) are plotted as a function of attraction ($v_i$) and defense ($h_{max} - h_i$), where $h_{max}$ is the maximum herbivory rate. The ancestral insect persists as a pure antagonist above the dashed gray lines and is extinct below. The insect persists as a net antagonist within the gray regions and is extinct within the white regions. Plant evolution of pollination benefits ($b_i > 0$) expands the green mutualistic regions by moving the interaction breakdown boundary (solid black line) away from the interaction transition boundary (dotted black line), as depicted by green arrows (see Supplementary Movies 4 and 5). Simultaneously, coevolution of attraction and defense drives the transition to net mutualism, as depicted by orange arrows from ancestral coESSs (white points) to new coESSs (black points). Empirical estimates of the coESSs are not included due to data limitations. Panels (**e**–**h**) plot the equilibrium densities of each *Datura* species (solid green lines) and insect larvae per plant of each *Datura* species (dashed green lines) over evolutionary time, $\tau$. Panels (*i*-*l*) plot the coevolutionary dynamics of pollination benefits ($b_i$; black lines), attraction ($v_i$; blue lines), and defense ($h_i$; purple lines).

attraction and defense than does *D. discolor*. *M. sexta* has a strong preference for nectar-feeding at *D. wrightii* over *D. discolor*[19], perhaps reflecting selection on *D. wrightii* for floral traits that attract *M. sexta*. Indeed, *D. wrightii* has flowers that are highly attractive to moths[20], whereas *D. discolor* has far less conspicuous flowers[25,26]. As nectar-feeding and oviposition behaviors are tightly linked in *M. sexta*[19], however, the coevolution of attraction necessarily changes pollination and oviposition simultaneously. For interactions lacking this behavioral linkage, coevolution might favor high floral visitation rates without an associated increase in oviposition risk, which would further facilitate transitions to mutualism. *D. wrightii* is also heavily defended against herbivory, producing proteinase inhibitors and other defensive compounds[27]. The seemingly paradoxical combination of *D. wrightii* being heavily defended and strongly preferred over *D. discolor*[19] is consistent with our model predictions. The strong oviposition preference may be a by-product of a greater visitation rate, as assumed in the model. It could also be that *D. wrightii* is a higher-quality host plant for *M. sexta* than is *D. discolor*[19], and *M. sexta* has evolved to tolerate its defenses. This difference is incorporated to a degree into the model via greater maturation and lower larval mortality on *D. wrightii* than on *D. discolor*.

The COA hypothesis resolves some of the key difficulties of the virulence theory of antagonism-to-mutualism evolution[8,9], while also predicting how mutualism can evolve in many of the most conspicuous partnerships on Earth. In virulence theory, mutualism evolution requires high host fidelity through vertical transmission and an evolutionary decrease in virulence. In the COA, evolutionary co-option also increases coupling with an antagonist, potentially increasing the costs of antagonism. Rather than a partner evolving lower virulence, however, it is the host plant evolving defense that reduces the antagonistic component of the interaction (against selection on the co-opted antagonist for greater antagonism), together with a level of attraction that

increases the by-product benefit of the interaction (pollination, in our model system).

The COA hypothesis also makes novel predictions about the influence of the ecological context of the interaction on evolutionary outcomes, showing that even slight trophic generalization by the insect (from one larval host plant species to two larval host plants and/or nectar sources) can profoundly affect the transition. In the *M. sexta* system, an alternative larval host plant increases the parameter region in which mutualism occurs relative to the one-plant species communities (Fig. 4a, b vs. 2a, b) by effectively subsidizing the costs of herbivory experienced by the *Datura* species. Conversely, an alternative nectar source greatly reduces the parameter region in which mutualism occurs relative to the one-plant species communities (Fig. 4c, d vs. 2a, b) by increasing oviposition and its consequent herbivory costs. The important insight is that lower partner fidelity due to even slight trophic generalization can either facilitate or hinder the transition from antagonism to mutualism depending on the specific resource axis along which the partner's niche broadens.

We have focused on evolutionary transitions from antagonism to mutualism, but there are also well established transitions from mutualism to antagonism or from mutualism to interaction breakdown (i.e., no interaction at all)[4,5]. Our results suggest that evolution can buffer interactions from transitioning to antagonism (as indicated by the distance from the coESSs to the interaction transition boundary in Figs. 2–4), but may predispose interactions to breakdown via partner extinction (as indicated by the proximity of the coESSs to the interaction breakdown boundary in Figs. 2–4). Thus, mutualism breakdown might more often result in partner loss[28,29] rather than a transition back to antagonism. Varying the evolutionary coefficients in the model reveals cases in which coevolution drives evolutionary purging of the antagonist or, more rarely, in which the net antagonism persists despite co-option of the antagonist (Fig. 5). Our model

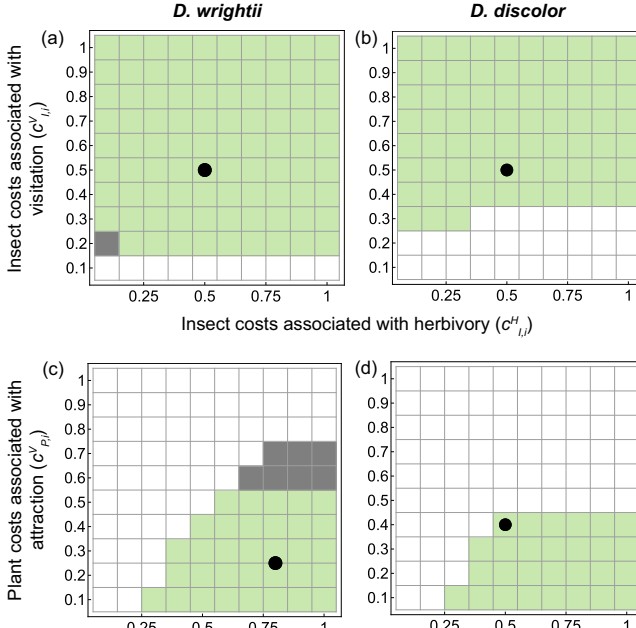

**Fig. 5 Varying the costs associated with plant and insect traits reveals coevolutionary outcomes.** Insect costs associated with herbivory ($c^H_{I,i}$) and visitation ($c^V_{I,i}$) are varied in panels (**a**, **b**) and plant costs associated with defense ($c^H_{P,i}$) and attraction ($c^V_{P,i}$) are varied in panels (**c**, **d**) (Methods). Green regions indicate evolutionary transitions from antagonism to net mutualism. Gray regions indicate that the interaction is net antagonistic despite evolutionary co-option of the antagonist. White regions indicate evolutionary purging of the antagonist. Black points give evolutionary parameter values used in the model (Methods).

results also suggest that environmental change, for example, could result in interaction breakdown or even transitions back to antagonism if it were to drastically alter evolutionary constraints (Fig. 5).

The COA framework reveals several key factors underlying evolutionary transitions from antagonism to mutualism. First, species (plants in our system) must possess traits that evolution can modify to co-opt the antagonist. In our system, for example, plants must have previously evolved flowers to benefit from other pollinators, which evolution could modify to co-opt the antagonist as a pollinator. Otherwise, the interaction would always be antagonistic. Second, species (plants) must be able to mount responses, such as attraction and defense, that could coevolve toward greater mutualistic benefits at lower trophic costs. Otherwise, the interaction would remain net antagonistic despite co-option of the antagonist. Third, if fitness costs are too extreme, the antagonism would either persist (gray regions of Fig. 5) or would disappear via evolutionary purging of the antagonist (white regions of Fig. 5). Finally, species (plants) must experience stronger selective pressures from the antagonist than from other species in the community.

For tractability, our model focuses on a system that is relatively specialized compared to most plant-pollinator and plant-herbivore communities. A critical question for future studies is how other interactions influence evolutionary trajectories from antagonism to mutualism. We briefly highlight two examples of how other interactions might modify the transition to mutualism. First, evolution of traits that attract the antagonist might come at the expense of attracting other pollinators or deterring other herbivores. If evolutionary constraints are too extreme, for example, the antagonism might persist or evolution might purge

the interaction in favor of other pollinators. Second, competition among plant species might modify coevolution, for example, by increasing interactions between the insect and competitively-dominant and abundant plant species. More broadly, extending the COA framework to larger communities is a key step.

The COA hypothesis generates several key predictions that could be tested empirically. We highlight two here. First, the COA predicts that the ecological context of an interaction strongly influences its evolutionary outcome. For example, our models suggest that an interaction is more likely to be mutualistic for species interacting with a focal partner that is more generalized in its antagonistic interactions than its mutualistic interactions (e.g., an insect that has multiple larval host plant species and few nectar sources). Second, the COA predicts that evolution of traits that allow an antagonist to function as a mutualist are not sufficient for an interaction to be mutualistic: coevolution of the interacting species is also necessary. For example, evidence that a herbivorous insect nectars at its larval host plant species does not inherently indicate that it is a mutualist: its benefits as a mutualist must be assessed relative to its costs as an antagonist[30].

The COA hypothesis helps to resolve the evolution of mutualisms that did not evolve de novo from costless interactions. For example, molecular evidence suggests that mutualistic bacteria evolved more frequently from parasitic than from free-living ancestors[11]. While virulence theory posits that it was the parasites that evolved lower virulence, it is also possible that the hosts evolved increased defense that reduced the virulence of co-opted partners in line with the COA hypothesis. Similarly, some insects have co-opted their parasitic endosymbionts for defense against parasitoids[31]. More generally, mycorrhizal mutualisms are thought to have evolved from parasitism[3], perhaps as plants co-opted the fungi to acquire their nutrients; early Angiosperms may have co-opted the foraging behaviors of pollen-feeders to disperse their pollen[3]; and some ant-defense mutualisms may have evolved from antagonism as ants were co-opted for defense[32,33]. Our eco-coevolutionary framework could help to unify ideas from the COA hypothesis, virulence theory[8,9], and other frameworks[6,7] to explain mutualism evolution across these natural systems.

We generated the COA hypothesis from the study of a pollination-herbivory system. However, the COA mechanism—evolutionary co-option and coevolutionary mitigation of antagonism (through defense) and amplification of mutualistic benefits (through attraction)—is not specific to this system and could apply much more broadly, including to microbial, mycorrhizal, and ant-defense mutualisms. Our results call for the integration of two lines of research that have proceeded largely independently: evolution of different interaction types (e.g., mutualism, antagonism) and interaction structures (e.g., specialized, generalized). This integration requires an eco-evolutionary perspective[2] in which the greatest progress will come by combining theoretical and empirical approaches. This study provides a critical step towards integrating theory and data to study the ecological and evolutionary dynamics of biological communities.

## Methods

**General framework for eco-coevolutionary transitions from antagonism to mutualism.** We develop a general framework in which we model interactions between host species $i$ (density $H_i$) and its partner species $k$ (density $F_k$), which are initially purely antagonistic. The model is general, but could be applied broadly to bacterial hosts and parasitic phages or plant hosts and animal or fungal partners, for example. The ecological dynamics of this community (without evolution) are given by:

$$\frac{dH_i}{dt} = g_i H_i\left(1 - \sum_j q_{ij} H_j\right) + \sum_k f_{ik}\left[\beta(H_i, F_k), \alpha(H_i, F_k)\right] \quad (4a)$$

$$\frac{dF_k}{dt} = \sum_i f_{ki}\left[\beta(H_i, F_k), \alpha(H_i, F_k)\right] - \delta_k F_k \quad (4b)$$

The first term of Eq. (4a) describes host population growth in the absence of partner species, where $g_i$ is its intrinsic per capita growth rate and $q_{ij}$ is the competitive effect of host $j$ on host $i$ for other limiting factors. The general function $f_{ik}$ describes the effects of interactions with partner $k$ on host $i$: $\beta(H_i, F_k)$ gives the potential mutualism and $\alpha(H_i, F_k)$ describes the antagonism. In Eq. (4b), the general function $f_{ki}$ gives the effects of interactions with host $i$ on partner $k$ and $\delta_k$ is the partner's per capita mortality rate.

To derive an explicit eco-coevolutionary model, we apply Equation (4) to model interactions between a single host species and its exclusive partner species (for the sake of simplicity) in terms of host traits $x_i$ and partner traits $y_i$ (involved in interactions with host $i$); the ecological dynamics of which are given by:

$$\frac{1}{H_i}\frac{dH_i}{dt} = g_i(1 - q_i H_i) + \frac{b[x_i^B]v[x_i^V, y_i^V]F_k}{S_i + v[x_i^V, y_i^V]F_k} - h[x_i^H, y_i^H]v[x_i^V, y_i^V]F_k \quad (5a)$$

$$\frac{1}{F_k}\frac{dF_k}{dt} = e[y_i^V, y_i^H]v[x_i^V, y_i^V]h[x_i^H, y_i^H]H_i - \delta_k \quad (5b)$$

where $b$ is the mutualistic benefits to the host, $v$ is the visitation rate, $S_i$ is a saturation constant, $h$ is the costs of antagonism to the host and its benefits to the partner, and $e$ is the partner's conversion efficiency. The mutualistic and antagonistic interactions are assumed to contribute additively to host population growth and multiplicatively to partner population growth, assumptions that may be valid for many types of interactions, but will not apply universally. To prevent unbounded population growth in the model, the effects of mutualism on host population growth are assumed to saturates with increasing partner density.

The function $b[x_i^B]$ gives the mutualistic benefits of the partner as a function of host trait $x_i^B$:

$$b[x_i^B] = b_{\max,i}\left(\frac{2}{1 + e^{-B_i' x_i^B}} - 1\right) \quad (6a)$$

where $b_{\max,i}$ gives the maximum mutualistic benefits and $B_i'$ is a saturation constant. The interaction is purely antagonistic when $x_i^B = 0$. As $x_i^B$ increases, the mutualistic benefits $b[x_i^B]$ increase towards $b_{\max,i}$.

The function $v[x_i^V, y_i^V]$ gives visitation rate as a sigmoid function of host trait $x_i^V$ and partner trait $y_i^V$:

$$v[x_i^V, y_i^V] = \frac{v_{\max,i}}{1 + e^{-V_i'(x_i^V + y_i^V)}} \quad (6b)$$

where $v_{\max,i}$ is the maximum visitation rate and $V_i'$ determines how rapidly visitation rate changes as host and partner traits change. As $x_i^V$ or $y_i^V$ increase, the visitation rate increases and approaches $v_{\max,i}$ when $x_i^V + y_i^V \to \infty$. As $x_i^V$ or $y_i^V$ decrease, the visitation rate decreases and approaches zero when $x_i^V + y_i^V \to -\infty$. Negative values of $x_i^V$ indicate that the host species is reducing its attraction to the partner species.

The function $h[x_i^H, y_i^H]$ gives the costs of antagonism to the host and its benefits to the partner, which is described via a sigmoid function of the difference between host trait $x_i^H$ and partner trait $y_i^H$:

$$h[x_i^H, y_i^H] = \frac{h_{\max,i}}{1 + e^{H_i'(x_i^H - y_i^H)}} \quad (6c)$$

where $h_{\max,i}$ gives the maximum antagonism and $H_i'$ determines how antagonism changes as the difference between host and partner traits increases. When $x_i^H > y_i^H$, antagonism declines and approaches zero when $x_i^H - y_i^H \to \infty$, while when $x_i^H < y_i^H$, antagonism increases and approaches $h_{\max,i}$ when $x_i^H - y_i^H \to -\infty$ (unlike $x_i^V$, $x_i^H$ cannot be negative).

Partner traits $y_i^V$ and $y_i^H$ trade off with conversion efficiency via the function $e[y_i^V, y_i^H]$ as defined by:

$$e[y_i^V, y_i^H] = e_{\max,i}e^{-\left(c_{I,i}^V(y_i^V)^2 + c_{I,i}^H(y_i^H)^2\right)} \quad (6d)$$

where $e_{\max,i}$ is the maximum conversion efficiency when interacting with host $i$ (when $y_i^V = y_i^H = 0$), and $c_{I,i}^V$ and $c_{I,i}^H$ determine how rapidly conversion efficiency declines as $y_i^V$ or $y_i^H$ increase, thus quantifying the costliness of traits $y_i^V$ and $y_i^H$, respectively. This trade-off shape was chosen because it is unimodal and constrains conversion efficiency to always be positive. Host trade-offs are defined below (Eq. 8c).

**Host-partner coevolutionary dynamics**. We model coevolution via the adaptive dynamics framework[17,18]. Coevolution of a mutant host trait $x_i^{mut}$ and partner trait $y_i^{mut}$ (for any general traits $x_i$ and $y_i$) is given by:

$$\frac{dx_i^{mut}}{d\tau} = \mu_x \left.\frac{\partial W_H(x_i^{mut}, x_i, y_i)}{\partial x_i^{mut}}\right|_{x_i^{mut} = x_i} \quad (7a)$$

$$\frac{dy_i^{mut}}{d\tau} = \mu_y \left.\frac{\partial W_F(y_i^{mut}, y_i, x_i)}{\partial y_i^{mut}}\right|_{y_i^{mut} = y_i} \quad (7b)$$

where $\tau$ is the evolutionary timescale, $\mu_x$ and $\mu_y$ give, respectively, the rates of host and partner evolution, and $W_H(x_i^{mut}, x_i, y_i)$ and $W_F(y_i^{mut}, y_i, x_i)$ are the invasion fitness (per capita growth rate when rare) of a mutant host and partner species with

trait $x_i^{mut}$ and $y_i^{mut}$ in a resident community with trait $x_i$ and $y_i$, respectively. The partial derivatives $\partial W_H/\partial x_i^{mut}\big|_{x_i^{mut}=x_i}$ and $\partial W_F/\partial y_i^{mut}\big|_{y_i^{mut}=y_i}$ are the selection gradients.

We model coevolution of mutualistic benefits from the focal partner species (via $b$), attraction (via $v$), and defense (via $h$). The invasion fitness of the mutant host and a mutant partner are given by:

$$W_H = g_i\left(1 - q[x_i^{mut}, x_i]H_i^*\right) + \frac{b[x_i^{B,mut}]v[x_i^{V,mut}, y_i^V]F_k^*}{S_i + v[x_i^{V,mut}, y_i^V]F_k^*} - h[x_i^{H,mut}, y_i^H]v[x_i^{V,mut}, y_i^V]F_k^* \quad (8a)$$

$$W_F = e[y_i^{V,mut}, y_i^{H,mut}]v[x_i^V, y_i^{V,mut}]h[x_i^H, y_i^{H,mut}]H_i^* - \delta_k \quad (8b)$$

where $H_i^*$ and $F_k^*$ are species' densities at the ecological equilibrium (of Eq. 5). The functions $b$, $v$, $h$, and $e$ are given by Eq. (6a–d), respectively, where $x_i$ and $y_i$ are replaced with $x_i^{mut}$ in Eq. (8a) and $y_i^{mut}$ in Eq. (8b). The function $q[x_i^{mut}, x_i]$ describes trade-offs between mutant host traits and mutant host competitive ability as defined by:

$$q[x_i^{mut}, x_i] = 1 + c_{H,i}^B\left((x_i^{B,mut})^{s_i^B} - (x_i^B)^{s_i^B}\right) + c_{H,i}^V\left((x_i^{V,mut})^{s_i^V} - (x_i^V)^{s_i^V}\right) + c_{H,i}^H\left((x_i^{H,mut})^{s_i^H} - (x_i^H)^{s_i^H}\right) \quad (8c)$$

If $x_i^{mut} > x_i$ for any trait, the competitive effect experienced by the mutant host is increased by an amount taken to be proportional (for simplicity) to the difference between the trait values, $x_i^{mut} - x_i$, whereas if $x_i^{mut} < x_i$, the competitive effect experienced by the mutant host is decreased by that amount. The coefficients $c_{H,i}^B$, $c_{H,i}^V$, and $c_{H,i}^H$ measure the costs associated with the trade-off for each trait, while the shape parameters $s_i^B$, $s_i^V$, and $s_i^H$ define whether the trade-offs are linear ($s_i = 1$), concave ($s_i < 1$), or convex ($s_i > 1$).

Mutualism can evolve via the COA for all trade-off shapes (Supplementary Fig. 3). Parameter space plots show that the interaction transitions from antagonism to net mutualism when the costs associated with host traits underlying attraction ($c_{H,i}^V$) and defense ($c_{H,i}^H$) are within a range beyond which there is evolutionary purging of the partner (Supplementary Fig. 3a–c). Only with convex trade-offs can the net antagonism persist. The coevolution of mutualism also requires that the costs associated with partner traits underlying visitation ($c_F^V$) and antagonism ($c_F^H$) exceed a threshold (Supplementary Fig. 3d–f) below which there is evolutionary purging of the partner (linear or convex trade-offs) or the net antagonism persists (linear or concave trade-offs). Coevolution of mutualism occurs across greater parameter ranges when the trade-offs are linear or slightly concave because costs increase less rapidly than with convex trade-offs.

**Ecological model of plant-insect interactions**. We tailor the general model (Eq. 4) to model populations of *D. wrightii* (density $P_w$) and *D. discolor* (density $P_d$) interacting with *M. sexta*. We scale the model so that $P_i = 1$ in the absence of *M. sexta*: thus, $P_i > 1$ indicates that pollination benefits exceed herbivory costs, and $P_i < 1$ indicates that herbivory costs exceed pollination benefits. The *Datura* species do not rely obligately on *M. sexta* and, consistent with ecology of the natural community (Box 1), the model incorporates the alternative host plant, *Proboscidea parviflora* (density $P_p$), and the alternative nectar source, *Agave palmeri*. The ecological dynamics of this community (without evolution) are given by:

$$\frac{1}{P_i}\frac{dP_i}{dt} = (1 - P_i) + \frac{b_i v_i A}{H + v_i A} - h_i L_i \quad (9a)$$

$$\frac{dL_i}{dt} = \varepsilon e_i v_i P_i A - m_i h_i L_i - d_i L_i \quad (9b)$$

$$\frac{dA}{dt} = \sum_i \rho_i m_i h_i L_i - d_A A \quad (9c)$$

Equation (9a) describes the population dynamics of plant species $i$ (*D. wrightii*, *D. discolor*, or *P. parviflora*). Equation (9b,c) give the dynamics of *M. sexta*: $L_i$ gives the larvae density on plant species $i$, which recruit into the adult population, $A$. Pollination is described by the term $b_i v_i A/(H + v_i A)$, where $b_i$ is the per capita growth of plant species $i$ due to pollination by the antagonist, $v_i$ is the visitation rate to plant species $i$ per antagonist adult, and $H$ is the saturation constant for pollination. Oviposition is given by $\varepsilon e_i v_i P_i A$, where $e_i$ is the oviposition efficiency (number of eggs laid per floral visit) and $\varepsilon$ is the fractional increase in egg production due to nectar-feeding at *A. palmeri*. Floral visits lead to both pollination and oviposition because these behaviors have been shown to be tightly linked in *M. sexta*[19]. Pollination and oviposition are given by saturating and linear functions, respectively, based on our data (Supplementary Data 1). Herbivory damage is given by the term $h_i L_i$, where $h_i$ is the herbivory rate per larvae on plant species $i$. Larvae mature at rate $m_i h_i L_i$, where $m_i$ is the maturation efficiency (fraction of larvae maturing on plant species $i$). Larval mortality on plant species $i$ is $d_i$, adult mortality is $d_A$, and $\rho_i$ is pupae survival (due to data constraints, we include pupae survival in our estimates of maturation $m_i$, set $\rho_i = 1$, and drop $\rho_i$ from equations hereafter). Equation (9a) gives the dynamics of the alternative larval host plant, *P. parviflora* ($b_p = 0$ and cannot evolve), which can coevolve attraction and defense. The alternative nectar source, *A. palmeri*, is incorporated within the model via the parameter $\varepsilon$.

**Model scaling**. Without the antagonist, plant population growth is given by $g_i(1 - q_iP_i)$, where $g_i$ is the per capita growth rate of plant species $i$ due to autonomous self-pollination or pollination by other species and $q_i$ is plant self-limitation. As $q_i$ is very difficult to quantify in nature, we scale the model so that $P_i = 1$ without the antagonist. We scale plant density ($\hat{P}_i = q_iP_i$), larvae density ($\hat{L}_i = q_iL_i$), herbivory rate ($\hat{h}_i = h_i/q_i$), maturation efficiency ($\hat{m}_i = q_im_i$), and survival of pupae ($\hat{\rho}_i = \rho_i/q_i$); where the hats denote scaled quantities and are dropped elsewhere for clarity. Thus, the model is scaled for parameterization, but is not non-dimensionalized. We then scale $g_i$ to 1 such that pollination benefits, $b_i$, are estimated by the ratio of the seed set of moth-pollinated flowers to autonomously self-pollinated flowers. Parameter estimates are for scaled quantities.

**Interaction breakdown boundary for ancestral interaction in a one-plant species community**. For the ancestral insect to persist, its per capita growth rate must be positive when it is rare (i.e., at $P_i^* = 1$, $L_i^* = 0$, $A^* = 0$). In stage-structured models, the per capita growth rate is given by the dominant eigenvalue ($\lambda_D$) of the matrix:

$$[-m_ih_i - d_i \quad \varepsilon e_i v_i P_i^* m_i h_i - d_A]$$

which is given by:

$$\lambda_D = \frac{1}{2}\left(-d_A - d_i - m_ih_i + \sqrt{(d_A + d_i + m_ih_i)^2 - 4(d_A(d_i + m_ih_i) - \varepsilon e_i v_i m_i h_i)}\right).$$

For the insect to persist, $\lambda_D$ must have a positive real part, which occurs only when the second term in the square root of $\lambda_D$ is negative; i.e., $d_A(d_i + m_ih_i) - \varepsilon e_i v_i m_i h_i < 0$. Rearranging this condition yields: $\left(\frac{\varepsilon e_i v_i}{d_A}\right)\left(\frac{m_ih_i}{m_ih_i + d_i}\right) > 1$. Applying $f_i = \frac{\varepsilon e_i v_i}{d_A}$ and $s_i = \frac{m_ih_i}{m_ih_i + d_i}$, where $f_i$ is insect lifetime fecundity and $s_i$ is the larval success (probability of larvae maturing rather than dying), yields Eq. (1).

**Interaction transition boundary in a one-plant species community**. For the interaction to transition from antagonism to mutualism, equilibrium plant density, $P_i^*$ must exceed one (see "Model scaling"). Setting Eq. (9b) to zero and solving for $P_i^*$ yields: $P_i^* = \frac{m_ih_i + d_i}{\varepsilon e_i v_i}\left(\frac{L_i^*}{A^*}\right)$. Setting Eq. (9c) to zero and rearranging terms then yields: $\frac{L_i^*}{A^*} = \frac{d_A}{m_ih_i}$. Thus, $P_i^* = \frac{m_ih_i + d_i}{\varepsilon e_i v_i}\left(\frac{d_A}{m_ih_i}\right)$ and (rearranging slightly) the condition for mutualism to arise is: $P_i^* = \left(\frac{d_A}{\varepsilon e_i v_i}\right)\left(\frac{m_ih_i + d_i}{m_ih_i}\right) > 1$. Rearranging and applying $f_i = \frac{\varepsilon e_i v_i}{d_A}$ and $s_i = \frac{m_ih_i}{m_ih_i + d_i}$ yields Eq. (2).

**Interaction breakdown boundary in a one-plant species community**. In the ancestral interaction, insect persistence is evaluated by whether or not it can increase from low density, which yields Eq. (1). Within the net mutualistic region, however, the insect cannot increase from very low density because it cannot buoy plant density sufficiently to maintain a positive per capita growth rate (mathematically, Eq. 1 cannot hold when Eq. 2 is satisfied). The mutualistic region is thus characterized by bistability (see Supplementary Figure 1), and the interaction breakdown boundary is determined by the conditions for the coexistence equilibrium to exist. At the coexistence equilibrium, the larval and adult densities are: $L_i^* = \frac{-B + \sqrt{B^2 - 4A_LC_L}}{2A_L}$ and $A^* = \frac{-B + \sqrt{B^2 - 4A_AC_A}}{2A_A}$, where $A_L = \varepsilon e_i v_i^2 h_i d_A$, $A_A = \varepsilon e_i v_i^2 m_i h_i^2$, $B = \varepsilon e_i v_i^2 m_i h_i\left(\frac{1}{f_is_i} + \frac{H}{v_im_i} - (1 + b_i)\right)$, $C_L = \varepsilon e_i v_i H d_A\left(\frac{1}{f_is_i} - 1\right)$, and $C_A = \varepsilon e_i v_i m_i h_i H\left(\frac{1}{f_is_i} - 1\right)$. For the coexistence equilibrium to exist, either $C_L$ and $C_A$ must be negative or $B$ must be negative and $L_i^*$ and $A^*$ must be real. $C_L$ and $C_A$ are negative when $f_i s_i > 1$, which is Eq. (1) and cannot hold within the mutualistic region because Eq. (2) must be satisfied. However, $B$ is negative when $f_is_i\left((1 + b_i) - \frac{H}{v_im_i}\right) > 1$, which is approximated by Eq. (3) when the last term is assumed to be small. For $L_i^*$ and $A^*$ to be real, $B^2 - 4A_LC_L > 0$ and $B^2 - 4A_AC_A > 0$. Assuming that the pollination saturation constant is small (i.e., $H \approx 0$) yields $C_L \approx C_A \approx 0$ such that $L_i^* \approx \frac{-B}{A_L} \approx \frac{m_i}{d_Af_is_i}(f_is_i(1 + b_i) - 1)$ and $A^* \approx \frac{-B}{A_A} \approx \frac{1}{h_i}(f_is_i(1 + b_i) - 1)$, which are both positive when $f_i s_i (1 + b_i) > 1$ as approximated by Eq. (3).

**Interaction transition and breakdown boundaries in a two-plant species community**. These boundaries are analytically intractable and are estimated by simulation (see codes provided online).

**Coevolutionary dynamics of plants and insect**. The effects of plant traits $x_i$ and insect traits $y_i$ on the ecological dynamics of the interactions are given by:

$$\frac{1}{P_i}\frac{dP_i}{dt} = (1 - P_i) + \frac{b[x_i^B]v[x_i^V, y_i^V]A}{H + v[x_i^V, y_i^V]A} - h[x_i^H, y_i^H]L_i \quad (10a)$$

$$\frac{dL_i}{dt} = \varepsilon e[y_i^V, y_i^H]v[x_i^V, y_i^V]P_iA - m_ih[x_i^H, y_i^H]L_i - d_iL_i \quad (10b)$$

$$\frac{dA}{dt} = \sum_i m_ih[x_i^H, y_i^H]L_i - d_AA \quad (10c)$$

We model coevolution of plant-insect interactions using the adaptive dynamics framework[17,18] to link population dynamics and trait coevolution. The coevolution of mutant plant trait $x^{mut}$ and insect trait $y^{mut}$ (for general traits $x$ and $y$) is given by Equation (7). We model the coevolution of pollination benefits from the antagonist, $b_i$ (via mutant plant trait $x_i^{B,mut}$), attraction (via mutant plant trait $x_i^{V,mut}$ and mutant insect trait $y_i^{V,mut}$), and defense (via mutant plant trait $x_i^{H,mut}$ and mutant insect trait $y_i^{H,mut}$). The invasion fitness of a mutant plant is given by:

$$W_{P,i}(x_i^{mut}, x_i, y_i) = \left(1 - q[x_i^{B,mut}, x_i]P_i^*\right) + \frac{b[x_i^{B,mut}]v[x_i^{V,mut}, y_i^V]A^*}{H + v[x_i^{V,mut}, y_i^V]A^*} - h[x_i^{H,mut}, y_i^H]L_i^* \quad (11a)$$

where $P_i^*$, $L_i^*$, and $A^*$ are the densities of the plant, insect larvae per plant, and insect adults, respectively, at the ecological equilibrium (of Eq. 10). The functions $b[x_i^{B,mut}]$, $v[x_i^{V,mut}, y_i^V]$, and $h[x_i^{H,mut}, y_i^H]$, describe the effects of mutant plant traits $x_i^{B,mut}$, $x_i^{V,mut}$, $x_i^{H,mut}$, and $x_i^{H,mut}$ on pollination benefits, attraction, and defense, respectively, which are defined by Eq. (6a–c), where $x_i$ is replaced with $x_i^{mut}$ (where the plant is the host species and the insect is the partner species). The function $q[x^{mut}, x_i]$ defines the trade-offs between mutant plant traits and the competitive ability of mutant plants, which is given by Eq. (8c) (with $s_i = 1$). At a coESS, $x_i^{mut} = x_i$ for all traits such that $q[x^{mut}, x_i] = 1$ and the original definition of $P_i > 1$ indicating that pollination benefits exceed herbivory costs is retained when pollination benefits evolve.

Invasion fitness of a mutant insect is given by the dominant eigenvalue of its system of equations evaluated at the resident equilibrium. In a one-plant species community, the insect invasion fitness is:

$$W_{I,i} = \frac{1}{2}\left(-d_A - d_i - m_ih_i^{mut} + \sqrt{(d_A + d_i + m_ih_i^{mut})^2 - 4\left(d_A(d_i + m_ih_i^{mut}) - \frac{\varepsilon e_i^{mut}v_i^{mut}h_i^{mut}d_A(d_i + m_ih_i)}{e_iv_ih_i}\right)}\right) \quad (11b)$$

where $v_i^{mut}$, $h_i^{mut}$, and $e_i^{mut}$ are functions describing the effects of mutant insect traits on attraction, defense, and mutant oviposition efficiency, respectively, which are given by Eq. (6b–d), where $y_i$ is replaced with $y_i^{mut}$. Invasion fitness of a mutant insect in a two-plant species community is given by the dominant eigenvalue of its system of equations evaluated at the resident equilibrium, which is analytically tractable, but sufficiently complicated that we do not include it here (see codes provided online).

The curves where the selection gradients (see Eqs. 7) become zero give the evolutionary isoclines for the coevolutionary system. The points where the isoclines intersect give the coevolutionary singularities, which are coevolutionary stable states (coESSs) when they are stable for both plants and the insect. For tractability, the local stability of the coevolutionary singularities was assessed by carefully inspecting the selection gradient of each trait in the neighborhood of its coESS with all other traits held at their coESS as well as by simulating coevolutionary dynamics. Importantly, all three plant traits ($x_i^B$, $x_i^V$, and $x_i^H$) and both insect traits ($y_i^V$ and $y_i^H$) all coevolve simultaneously in the model.

**Coevolution of the ancestral antagonistic interaction**. In the ancestral interaction, pollination by the antagonist is impossible ($b_i = 0$) and thus visitation only contributes to oviposition. From the plant perspective, the selection gradients for attraction and defense in the ancestral interaction are given by:

$$\left.\frac{\partial W_{P,i}}{\partial x_i^{V,mut}}\right|_{x_i^{mut}=x_i} = -c_{P,i}^V P_i^* \quad (12a)$$

$$\left.\frac{\partial W_{P,i}}{\partial x_i^{H,mut}}\right|_{x_i^{mut}=x_i} = \frac{h_{max,i}H_i'e^{H_i'(x_i^H - y_i^H)}}{\left(1 + e^{H_i'(x_i^H - y_i^H)}\right)^2}L_i^* - c_{P,i}^H P_i^* \quad (12b)$$

Equation (12a) predicts that selection favors plant traits that reduce attracting the antagonist (e.g., reduced production of volatiles) and lower costs associated with competitive ability. We constrain $x_i^V$ to be non-negative in the ancestral interaction so that $x_i^V = 0$ at the coESS; otherwise, $x_i^V \to -\infty$ and the plant always purges the insect given this model parameterization. Selection balances reduced herbivory damage (first term of Eq. 12b) with costs of reduced competitive ability (second term of Eq. 12b). Selection gradients for insect traits are sufficiently complicated that we do not include them here (see codes provided online); however, selection balances traits that increase visitation and overcome plant defenses with the costs associated with reduced oviposition. The ancestral coESSs are given in Supplementary Table 3.

**Coevolution of pollination benefits, attraction, and defense**. The evolution of mutant plant traits that allow the antagonist to pollinate it ($b_i^{mut} > 0$) initiates the evolution of pollination benefits from the antagonist. The selection gradient for

pollination benefits from the antagonist is given by:

$$\left.\frac{\partial W_{P,i}}{\partial x_i^{B,mut}}\right|_{x_i^{mut}=x_i} = \frac{2b_{\max,i}B_i'e^{-B_i'x_i^B}v[x_i^V,y_i^V]A^*}{\left(1+e^{-B_i'x_i^B}\right)^2\left(H+v[x_i^V,y_i^V]A^*\right)} - c_{P,i}^B P_i^* \quad (13a)$$

Equation (13a) shows that plants evolve traits to benefit from floral visits by the antagonist when selection for increased pollination benefits (first term of Eq. 13a) exceeds the costs associated with reduced competitive ability (second term of Eq. 13a).

In the model, pollination benefits from the antagonist evolve via Eq. (13a) simultaneously with plant and insect traits affecting attraction and defense. The plant selection gradient for attraction is now:

$$\left.\frac{\partial W_{P,i}}{\partial x_i^{V,mut}}\right|_{x_i^{mut}=x_i} = \frac{b[x_i^B]v_{\max,i}V_i'e^{-V_i'(x_i^V+y_i^V)}HA^*}{\left(H\left(1+e^{-V_i'(x_i^V+y_i^V)}\right)+v_{\max,i}A^*\right)^2} - c_{P,i}^V P_i^* \quad (13b)$$

The co-option of the antagonist has fundamentally changed selection on attraction (Eq. 13b vs. Equation 12a), which now balances traits affecting attraction (first term of Eq. 13b) with the costs of reduced competitive ability (second term of Eq. 13b). Co-option of the antagonist also modifies selection on defense (which is still given by Eq. 12b) by changing both trait values and equilibrium densities.

**Model parameterization**. All ecological parameters are estimated from empirical data. Here we parameterize the saturation constant $H$, maturation efficiency $m_i$, larval mortality $d_i$, and adult mortality $d_A$ as well as the parameters for the alternative larval host plant and the alternative nectar source (see "Model validation" for other parameters).

We cannot fit the saturation constant $H$ to data because seed set saturates with even a single floral visit. We therefore estimate $H$ as follows: D. wrightii flowers have a 91% chance of setting fruit[30]; thus, $v_wA/(H+v_wA)=0.91$ for a single visit ($v_wA=1$). Solving $1/(H+1)=0.91$ for $H$ yields: $H=0.1$. $H$ is assumed to be the same for D. discolor as pollination benefits saturate with a single visit for D. discolor. For maturation efficiency $m_i$, only 0.5% of M. sexta larvae on D. wrightii survive through the final larval instar in nature[34]; thus, $m_w=0.005$. As M. sexta suffers 40% lower larval survival on D. discolor (5/8 larvae surviving to pupation) than on D. wrightii (10/10 larvae surviving to pupation) in our experiment[19], we estimate that maturation efficiency is ~40% lower on D. discolor than on D. wrightii; i.e., $m_d=(1-0.4)m_w=0.003$. To estimate larval mortality, we note that larval survival is given by: $m_i=e^{-d_iD_i}$, where $D_i$ is development time. M. sexta has a larval stage of ~20 days on D. wrightii[35] and there is no difference in development on D. wrightii and D. discolor, at least to the 5th instar[19]. Solving for $d_i$ yields: $d_w \approx 0.25$ and $d_d \approx 0.3$. Finally, adults live ~5 days in the wild[36]. Assuming adult mortality is roughly the inverse of the lifespan: $d_A \approx 0.2$.

For the alternative larval host plant, females lay similar numbers of eggs on D. wrightii and P. parviflora[34]; thus, visitation rate and oviposition efficiency are assumed to be the same as with D. wrightii; i.e., $v_p=v_w$ and $e_p=e_w$. Because P. parviflora plants are of similar size and architecture as D. wrightii[34], we assume that herbivory rate on P. parviflora is the same as on D. wrightii; i.e., $h_p=h_w$ (see "Model validation" for estimates of $v_w$, $e_w$, and $h_w$). Only 1% of M. sexta larvae on P. parviflora survive through the final larval stage[34]; thus, $m_p=0.01$. As larvae have roughly the same development time on P. parviflora as on D. wrightii (~20 days[37]), solving $m_p=e^{-d_pD_p}$ yields an estimate of larval mortality on P. parviflora of: $d_p \approx 0.25$.

For the alternative nectar source, A. palmeri provides M. sexta with copious amounts of nectar that females likely utilize for egg production[38]. M. sexta females lay 100–300 eggs/night[39]. If females foraging exclusively on D. wrightii lay the minimum 100 eggs/night and females that also forage at A. palmeri lay the maximum 300 eggs/night, then A. palmerii is estimated to increase oviposition by a factor of: $\varepsilon=3$.

**Model validation**. Pollination benefits ($b_i$), visitation rate ($v_i$), herbivory rate ($h_i$), and oviposition efficiency ($e_i$) all evolve simultaneously in the model. We independently validate the coESSs predicted by the models whenever possible by estimating these parameters using data that were not used to parameterize the models. We estimate $b_i$ via the ratio of the seed set of moth-pollinated flowers to autonomously self-pollinated flowers (autonomously self-pollinated seeds germinate as readily as do outcrossed seeds;[30]). Pollinated D. wrightii and D. discolor flowers set $b_w=4.6\pm0.2$ and $b_d=3.6\pm0.1$ times more seeds, respectively, than do autonomously self-pollinated flowers (D. wrightii: n = 21 fruit; D. discolor: $n=85$ fruit). Moths averaged $v_w=4.3\pm0.6$ floral visits to D. wrightii ($n=89$ plants) and $v_d=2.4\pm0.4$ floral visits to D. discolor ($n=33$ plants) in our experiment[19]. Estimating the herbivory rate is very difficult in nature; however, we can make cursory estimates based on our data. A single M. sexta larvae can consume 1400–1900 cm² of leaves, which is more than many D. wrightii plants in nature[30]. Assuming that an average D. wrightii plant supplies larvae with 1400 cm² of leaves, the variation in leaf consumption (500 cm²) represents ~0.4 plants (=500/1400). Thus, M. sexta larvae are estimated to consume: $h_w \approx 1\pm0.4$ D. wrightii plants. M. sexta larvae consumed roughly two times more D. discolor leaf biomass than D. wrightii leaf biomass based on our cursory estimates from our experiments; thus, $h_d=2h_w \approx 2\pm0.8$. We estimate oviposition efficiency by the slope of a linear

regression of the number of eggs versus the number of floral visits that each plant received from each female moth in our experiments[19], which yields: $e_w=0.6\pm0.1$ ($n=34$ plants) and $e_d=0.6\pm0.2$ ($n=24$ plants) (Supplementary Data 1).

**Estimating evolutionary model parameters**. Directly estimating evolutionary parameters with data is not possible. We therefore use theory to predict how key parameters affect eco-coevolutionary outcomes and to select reasonable parameter estimates. Our approach is as follows. We set the rates of plant and insect evolution to one ($\mu_x=\mu_y=1$); these rates affect the speed of evolution, but not the coESSs. For each trait, we need to estimate the maximum value ($b_{max,i}$, $v_{max,i}$, $h_{max,i}$ and $e_{max,i}$), the coefficient ($R_i'$, $V_i'$, and $H_i'$), and the associated costs ($c_{P,i}^B$, $c_{P,i}^V$, and $c_{P,i}^H$ for plant $i$ and $c_{I,i}^V$ and $c_{I,i}^H$ for the insect). Maximum trait values were chosen to constrain coevolution to a realistic range. We set the coefficients $R_i'$, $V_i'$, and $H_i'$ to one for simplicity because the exact value of any trait $x$ and $y$ are themselves somewhat arbitrary. The costs associated with the traits therefore largely determine the coevolutionary outcomes in the model.

We estimate the costs of each trait by systematically varying the costs of plant traits in the one-plant species community given reasonable values for the insect costs and then systematically varying the costs of insect traits while holding plant costs constant at their chosen values (Fig. 5). Parameter space plots show that the interactions transition from antagonism to net mutualism provided that the costs associated with insect traits underlying visitation ($c_{I,i}^V$) exceed a threshold below which the plant and insect engage in an evolutionary arms-race that results in the evolutionary purging of the antagonist (Fig. 5a, b). Only very rarely does the net antagonism persist. We assigned all insect traits a cost of 0.5 (black points in Fig. 5a, b) and then systematically vary the costs of plant traits associated with attraction and defense.

Parameter space plots show that interactions transition from antagonism to net mutualism when the costs associated with defense are high relative to the costs associated with attraction ($c_{P,i}^H > c_{P,i}^V$); otherwise, coevolution drives evolutionary purging of the antagonist (Fig. 5c, d). When the costs associated with attraction and defense are both fairly high, the net antagonism persists. We assigned values of $c_{P,i}^H$ and $c_{P,i}^V$ to D. wrightii and D. discolor such that the parameters for D. discolor are closer to the threshold at which evolutionary purging occurs than are those of D. wrightii (Fig. 5d vs. 5c), reflecting the smaller range of ecological parameters over which M. sexta can persist with D. discolor versus with D. wrightii (Fig. 2b vs. 2a). Finally, the costs associated with pollination benefits from the antagonist ($c_{P,i}^B$) must be very high for the net antagonism to persist and we never observed evolutionary purging of the insect within the range of values used (see codes provided online). We assigned values of $c_{P,i}^B$ so that pollination benefits to D. wrightii and D. discolor are well below their maximum values. Our estimates of evolutionary parameters are reported in Supplementary Table 2. Evolutionary parameters for P. parviflora are set equal to D. discolor because, in the absence of more information, both species are annual plants that may face broadly similar evolutionary constraints, at least relative to the perennial D. wrightii.

**Reporting summary**. Further information on research design is available in the Nature Research Reporting Summary linked to this article.

## Data availability
Data used to parameterize and validate the models are available within the paper and its Supplementary Information as well as on Zenodo: https://doi.org/10.5281/zenodo.4628187. Our data was recorded in Microsoft Excel (v. 16.48) and was analyzed in R (v. 3.1.0).

## Code availability
Codes are available on Zenodo: https://doi.org/10.5281/zenodo.4628187. The codes were developed in Mathematica (v. 12.0).

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

## Acknowledgements

We thank Julie Johnson (Life Science Studios) for the beautiful illustrations in Fig. 1 and for help designing all of the Figures. We thank Olivia Brinkerhoff, Kimberly Kopplin, Nico Lorenzen, Keane Sullivan, and James Wadsworth for their help in growing plants and rearing larvae as well as Heather Costa for colony maintenance. This work was supported by Science Foundation Arizona grant BSP 0528-13 to C.A.J, U. S. National Science Foundation grant IOS-1053318 to G.D., U. S. National Science Foundation, Dimensions of Biodiversity program (DEB-1831493) to R.F., and the Keep Engaging Youth in Science (KEYS) program of the University of Arizona BIO5 Institute.

## Author contributions

C.A.J., G.P.S., G.D. and J.B. designed the experiments; C.A.J. and G.P.S. performed the experiments and analyzed the data; C.A.J. developed the models; C.A.J., K.Y. and R.F. developed methods to analyze the models; and C.A.J. wrote the manuscript with input from all authors. J.B. and R.F. co-advised the project.

## Competing interests

The authors declare no competing interests.
