## [Peer Review File · Nature Communications]

REVIEWER COMMENTS

Reviewer #1 (Remarks to the Author):

In this manuscript, the authors propose a new hypothesis to explain how a mutualistic interaction can evolve from an antagonistic interaction. They use an eco-evolutionary model, based on an empirical system, to evaluate their Co-opted antagonist hypothesis.

The authors use a system of a moth species and its associated plant species. As an adult, the moth visits plants to nectar feed (and potentially pollinate), while the caterpillars feed on the leaves of the plant. Hence, the interaction changes from an antagonist to a potential mutualist over the life-time of the moth.

In the model, initially the moths do not pollinate the plants and the interaction is therefore fully antagonistic. In the first step of the model, the plants evolve a trait that allows butterflies to pollinate it. However, this does not lead to a mutualism, since the costs of herbivory are still higher than the benefits of pollination. In the second step of the model, the host plant evolves attraction and defense traits, which then leads to the transition of the system from antagonism to mutualism. The authors show results for different type of plant communities (e.g., an additional nectar source).

In general, I really like the paper. It is well written, and I think of interest for many people. Even though there are many systems where an herbivorous insect pollinates some of its host plants, there are only a few studies investigating the evolutionary causes and consequences of such interactions. A particular strength of this manuscript, is that the model is based on and parameterized for a biological system and therefore has a clear basis in biological reality. However, not all assumptions of the system will apply to other systems, and I therefore wonder how general the results of this study are. In particular, I wonder how important the ontogenetic change in interaction type is for the results. I understand that this study is a first step towards understanding the potential of the COA hypothesis, however, I think that the authors should be a bit more careful in generalizing their results.

I find the results in the manuscript complex and I think that the result section needs some rewriting to improve the readability of the manuscript. The authors assume a two-step process, where the host first evolves such that the antagonist can perform a beneficial function. After that, the host evolves to exploit the antagonist such that the interaction becomes mutualistic. It is not completely clear to me why a two-step process is assumed. Would the results be similar in case all three traits evolve simultaneously or not? And if not, how realistic is it to assume that first one trait evolves, and only later the other traits? Some justification and explanation is in my opinion needed for this assumption. In addition, not all model assumptions are clear and some need stronger justification.

I have listed details below.

1) The authors explore a wide variety of scenarios, which makes this study really interesting and complete. However, it is therefore also a bit difficult to follow what is happening. As far as I understand, first parameter r (effect of pollination) evolves in a single-plant system. After that, the authors study the

co-evolution of attraction & defense in both a single-plant system, a two plant-system, and systems with an additional host plant and/or alternative nectar plant. So, the first step of the COA hypothesis is only studied in a single-plant system, while the second step is studied in a variety of communities. I think the result section could be improved by

- a. Stating explicitly that step 1 is only studied in a single-plant system (plus maybe a justification why this is sufficient).
- b. Stating explicitly that it is assumed that r is in the ESS of step 1, and does not evolve in step 2.
- c. Being more specific of the ancestral trait in case of the additional host/nectar plant. Is an r -value assumed that evolved in the single-plant system? Would the r be different otherwise and would that affect the outcome?

2) In addition, in the more complex systems it is not very clear if the two plants evolve simultaneously or if only one plant evolves while the other is in its ESS. Does it matter? Would co-evolution of the two plants change the results?

3) Initially, only parameter r evolves and parameters v and h are fixed. In step 2 parameter r is fixed while parameter v and h evolve. I think it would be helpful if this assumption is clarified in the main text to improve readability. Also some justification is needed why the authors assume a 2-step process. Would the results be very different in case all three parameters can evolve simultaneously? In addition, what happens to parameter r in case a mutualistic interaction has evolved?

4) All three trade-off functions (eq6, eq7b, p10) depend on the difference between the mutant trait and another value (r_i , h_i , v_i). It is not clear what the authors mean by these values. Are these the traits of the resident? Or are these the traits of the ancestral species? Some explanation and justification of the trade-offs is needed. In case the value is the trait of the resident, the trade-off disappears in the CSS, is this realistic?

5) In step 1 of the model, only parameter r (effect of pollination) evolves. The authors assume that an increase of pollination leads to a higher herbivory rate (eq 6, p10). What is the biological reason behind this assumption? Why would an increase in r leads to a higher herbivory rate?

6) In step 2 of the model, I think it is assumed that r (effect of pollination) has the value it evolved to in step 2. In step 1, however, an increase in r increased the herbivory rate due to the trade-off. It is unclear if this is still true in step 2. Some justification is needed in case the herbivory rate is set back to the value where $r = 0$.

7) In the discussion the authors do discuss the possibility of purging of the partner (L274-L280 and SI fig 4). In the absence of pollination ($r = 0$), I expect that it is possible for the plant to purge its partner as well by evolving v & h to values outside the interaction breakdown. A short explanation why the authors don't take this scenario into account would be helpful.

8) Related to my previous comment, the authors state in the discussion that the first step of COA

reshapes selection acting on attraction and defense trait (L220-221). However, in the ms the authors do not study the evolution of these traits before the first step of COA. It is therefore not clear from the ms how the evolution of these traits is reshaped by the change in r .

9) The authors predict that generalist adults feeding at multiple nectar sources may hinder the transition to mutualism by reducing pollination benefits (L47-48, L264). However, the way pollination is modelled, pollination benefits are not affected by alternative host plants, since adult visitation rates are not constraint and not affected by other plants. Therefore, this predication does, in my opinion, not make much sense since it will, at least in this model, never happen.

10) The authors study the evolutionary transition from antagonism to mutualism in a consumer species that has an ontogenetic shift from antagonism to a neutral or mutualistic interaction. How important is this ontogenetic shift for the results? Does the COA hypothesis also works in the absence of complex life cycles? The authors write (L316-L317) that their results are not specific to their specific system and could apply more broadly. I find this statement a bit bold, given that the studied system is quite specific.

11) I am not sure if this is relevant, but would it be helpful to show how evolution of plant traits affect the pollinator dynamics? Does the increase in plant density somehow benefit the moth and does this in turn has a positive effect on plant densities? The eco-evolutionary dynamics are hidden in the manuscript, while it might help readers to understand what is going on.

Minor comments

In figure 2 (p4) the authors show that the predicted ESS is very close to the ESS based on data (cool!). I wonder if there is some information on the ESS in case of an additional host/nectar plant and if the prediction is equally good.

The meaning of the white dots is ambiguous. Sometimes the authors refer to them as the ancestral traits (e.g., L120-121), while sometimes it is referred to as the ancestral ESS (e.g., L139). Is it an ESS? If yes, it seems that these results are not in the manuscript.

L338 Should this not be equation 3 instead of 1?

L339-L340 'Which is scaled such that A gives the adults recruited per plant of species i '. How is this scaled?

L512 should it not be $P_{\hat{i}} = P_i * q_i$?

L513 According to the text herbivory rate h is scaled, but the equation suggests that maturation m is scaled.

L513 Time is scaled as well (L548), this should be explained here.

L516 From the text it is not clear if estimated parameters are the scaled or the unscaled values.

L538 'We vary h in ecological models'. What do the authors mean with the ecological models? There are only results from eco-evo models as far as I can see.

P20/Sup Fig 1: It is unclear what the authors plotted in panels c-f. What are the dark areas? What does this plot tell?

P22/Sup Fig 3: There is no reference in the text to this figure.

Reviewer #2 (Remarks to the Author):

Dear Editors and Authors,

In "Evolutionary transitions from antagonism to mutualism explained by the Co-Opted Antagonist Hypothesis" Johnson et al propose a model of mutualism evolution from antagonism. In this model, a plant initially begins to receive some small benefits (pollination) from a herbivore, but largely suffers defoliation costs. They show that selection on either pollination amount or two traits that separately influence pollination amount in the plant (defense and floral attraction) could drive an evolutionary trajectory that reaches a stable mutualistic state instead of continued antagonism or total breakdown of the interaction. The authors develop an explicit model, which they parameterize with data from a real system (*Manduca sexta* and *Datura* spp), and include a number of relevant extensions.

This work has one very key strength which I want to emphasize before anything else. While the evolution of breakdown vs. maintenance in mutualisms is highly studied, the evolution of mutualism origin in the first place is under-represented in the literature (though perhaps not as woefully as the authors paint it). Further, explicit mathematical models considering population dynamics (here, e.g. plant density), are generally under-represented for mutualisms even without considering evolution – so this contribution fills two-part literature gap. Also, the text is generally well-written, and really quite exciting to read. Despite the criticisms I levy below, I think this article has a good likelihood of being influential in the field.

The authors' treatment of the real system is no less than excellent: they have considered many aspects of relevant biology here, from ecological constraints on traits to even heterospecific pollen. Yet in both the introduction and discussion, it's clear that this mechanism is likely to be a wider phenomenon than just for Lepidoptera and hosts. It could very much extend the impact of this paper to provide at least one generalized presentation of the model -- as the literature gap it fills extends across mutualism types.

The only true major issue I see is that of insect evolution or co-evolution. Here, evolution (or even

behavioral variation-- i.e. insects choosing to oviposit on less defended hosts) in the insect is not considered, except in sudden host shifts. However, it seems confusing to consider only plant evolution when insects are likely to have higher population sizes, and faster generation times – traits that suggest they might do most of the evolution in co-evolutionary scenarios (e.g. Gandon & Michalakis 2002, Bergstrom & Lachmann 2003). Further, the influence of insect evolution could lead to opposite outcomes. As the plant increases herbivory defense and decreases attraction, the obvious insect evolutionary responses (increased tolerance of secondary compounds and increased sensitivity to olfactory cues for finding hosts) are likely to escalate antagonism (push defense down and further into the antagonistic outcome zone). Indeed, *M. sexta* is unaffected by the extreme chemical defense of *D. wrightii*, which seems certain to be an evolutionary response. On the other hand, from a state where hawkmoths nectar exclusively on non-*Datura* species, a hawkmoth mutant that is able to nectar on the plant it oviposits on may have higher relative fitness through more efficient resource use, mitigating plant fitness costs as a byproduct –e.g. the reverse co-option model.

Four minor points:

1) I am confused as to why this model has two separate stages. Seems like in the real world co-option would proceed from increasing floral attraction from 0 or a low value, while leaf attraction would be at some high value. Further, the stages seem to differ in the model only slightly: in the first stage pollination is a trait that mutates on its own, where as in the second it becomes a function of correlated impacts of two mutant traits (defense and attraction). As a sidenote here, I'd like to point out that the correlated traits aspect of stage 2 allows the inclusion of constraint and is one of the neat features of this model.

2) The methods are not clear in places. Much of the explanation for the model and math is in the supplemental text, yet most is key to a basic understanding of the model. Despite careful reading, I am left with a few confusions about what was actually done and what wasn't, though probably only because it exists chopped up into so many different places. Because of this, it is hard for the reader to distinguish ecology only or eco-evo models, figures, and results.

3) This model is somewhat similar to two existing ideas in the literature. For one (evolution of mutualism from vertically or semi-vertically transmitted parasites/antagonists), the authors distinguish their model, but perhaps underemphasize previous work – e.g. suggestion of evolution of ant plant defense from herbivory (Nelson et al 2018) or of ant-hemipteran defense from ant-hemipteran predation (Stadler & Dixon 1999). Yet for another, even more similar idea (evolution of mutualism from herbivory via tolerance – de Mazancourt et al 2001, 2005) the authors do not. I note that a generalized case of the authors' model as I have suggested could provide a unifying framework across these systems.

4) As a slightly different mechanism, could the timing of flowering and herbivory actually induce the plant fitness benefit of nectaring by the herbivore? an early herbivore that also nectars could reduce fitness benefits of late pollinators because the eating/developing larva destroy later developing leaf and

flower tissue. Related to my comment for line 364.

Line by line notes, mostly examples of points above:

Just to quickly note: The readme file for the code looks good, and notebooks seem more than adequate for repeating analyses. I have not checked them in Wolfram Mathematica, as I am not familiar, nor do I currently have access to a machine I can install it on (COVID-19 related)

Line 82: Why "Given" when it is modeled?

Line 98: It took me 3 read-throughs to realize that "Co-option of a herbivore as a pollinator" models are not in Figure 2, even though the paragraph cites Figure 2 panels. Also, this section seems to mostly discuss trait models, even though this section isn't ostensibly about trait models.

Line 118: Defense metric was hard to get, and could use a bit more explanation. ex. "When h_i is high, that means herbivory is near the max herbivory."

Line 125: For panels g and h, I don't know what ESS of "other" species means when the ESS clearly depend on each other (shifted from e and f) – does this mean that one species is held at its single-species ESS? And only one plant evolves in the two plant community?

Also Figure 2: Given that the model is parameterized with data from the experiment, is it really exciting that the ESS matches traits measured in that same experiment? And, if that is exciting, why aren't the measured traits also reported in Figure 3?

Line 126: The concept of isocline isn't explained, unclear what it means, especially in contrast to the arrows in the figures.

Line 140: Just wondering generally, if there's any evidence to support ancestral antagonism in this system.

Line 146: Is it really independent? seems the parameterization came from the same experiment first place...

Line 147: Extinction how? Do stochastic forces push it across into the trait space where the interaction breaks down? It should be specified.

Line 161: When adding a host or a nectar source, why do we assume amelioration of herbivory? Couldn't the reverse be true? (i.e. apparent competition – two hosts support greater than additive larval density and therefore more costs?) In contrast, line 173 I understand well and is a very clear point.

Line 168: Should this be "attraction"? I find the distinction between "attraction" and "visitation" to be

confusing generally.

Line 169: When the insect was initially generalized, where is the starting state on the traits? And why?

Line 181: Took me a long time to get exactly what happens here: first evolution to ESS in a single species (which authors have argued is unstable, line 147); then when porting those plant trait coordinates to the new ecological context of a host shift (Fig 3 upper), the interaction is again antagonism, but eco-evo dynamics then push it back to mutualism again (Fig 3 lower). Could use more explanation.

Figure 3: Is *D. discolor* still in the models? If so, as an evolving or non-evolving host? Not clear. Here is also where I noticed that Supplementary Table 3 suggests there are models with all four plants, yet no results/discussion for them?

In Figure 2, and now 3, I found it hard on first couple read-throughs to decisively identify ecology only and evolutionary scenarios. I think its because there are “initial values” on the ecology only models that just map the interaction outcomes to regions of trait-space -- e.g. there are no evolving values in those panels, correct?.

Line 248: These arguments are confusing. How can *D. wrightii* be both better defended (defined as less eaten which is tied to less larval development) and a better host (more larval development) within the confines of the model? To me, this suggests that co-evolution matters.

Line 250: This paragraph seems to fly in the face of the initial justification (that this model is great because it's not vertical transmission) Further, “vertical” here is not that more attractive plants interact with better pollinators, they just interact with all pollinators more...Are these really the same idea as vertical transmission and partner fidelity as used in the literature?

Line 268: This paragraph is a great contribution. Plus this fits with other literature (see Chomicki et al 2017; Werner et al 2018) where breakdown is generally loss. And is a key example of why generalizing the model & introduction would be very useful to the mutualism research community writ large. – see also lines 303-312.

Line 277: π was not defined previously, thus its not clear what this means.

Line 281: This paragraph is about how often the fitness benefits of pollination by an antagonist would be high enough. Is there anything known on how often host plants where herbivores also nectar are pollen limited?

I didn't fully understand lines 295-303, but I think probably only because I don't understand the example. I think maybe some detail is missing? Or maybe the wording has me confused.

Line 342: What is the half saturation constant for? I intuit from recalling predator feeding equations,

that it could account for re-visitation of already visited plants? But that feels wrong given later text...

Line 361: Seems there are other natural ways to deal with this? E.g. there could be a limited number of resources devoted to flowers? Or maybe I don't understand the problem here?

Line 364 and 378-385. This is great consideration of trade-offs/epistatic links between attraction/herbivory and defense/repulsion and how they influence the evolution of mutualism, but aren't there other relevant mechanisms that would do the same thing – e.g. if single chemicals are both defensive and pollinator repulsive (Kessler 2011)? Interestingly, it seems to me that such a mechanism could induce reliance on hawkmoths as pollinators in the first place – if herbivory in any way makes plants less attractive to other pollinators. I'm also curious as to how this trade-off affects the relationship between attraction and realized defense inherent defense traits (after accounting for D_v , D_r) versus v and h ?

Line 485: How can there be cross-pollination if there is only one individual per species?

Line 488: I was excited to see that the authors considered the possible role / cost / influence of avoiding heterospecific pollen. That really demonstrates a thorough thinking about the system. However this sentence confused me. The words on the page imply that good viable seeds are formed from the transfer of only heterospecific pollen (really?). Yet, I want to understand that adding heterospecific pollen doesn't interfere with the production of seeds from conspecific pollen. Which is it?

Line 524: For which models exactly are v_w and v_d set to these values? I'm confused.

Line 531: Does the $h_w = 1$ and $h_d = 2$ relationship assume plant species and individuals are equal sizes? since herbivory subtracts from density which is individuals? or is density foliage based? Should be clarified.

Line 548: I understood previous model scaling, but somehow not this one. Perhaps more detail would help?

Line 560: Sorry if I'm being dense, but I don't understand the "double" and "half" explanations. If 80% of larvae survived on *P. parviflora*, that means 20% died, and if survivorship on *P. parviflora* is twice that of *D. wrightii* that would be 40% survive on *D. wrightii* (and 60% die). But 60% is not double 20%.

Line 567: What are the joules from? Calories in volume of nectar from one visit?

Bergstrom, C. T., & Lachmann, M. (2003). The Red King effect: when the slowest runner wins the coevolutionary race. *Proceedings of the National Academy of Sciences*, 100(2), 593-598.

Chomicki, G., & Renner, S. S. (2017). Partner abundance controls mutualism stability and the pace of morphological change over geologic time. *Proceedings of the National Academy of Sciences*, 114(15), 3951-3956.

Gandon, S., & Michalakis, Y. (2002). Local adaptation, evolutionary potential and host–parasite coevolution: interactions between migration, mutation, population size and generation time. *Journal of Evolutionary Biology*, 15(3), 451-462.

Kessler, A., Halitschke, R., & Poveda, K. (2011). Herbivory-mediated pollinator limitation: negative impacts of induced volatiles on plant–pollinator interactions. *Ecology*, 92(9), 1769-1780.

de Mazancourt, C., Loreau, M., & Dieckmann, U. (2001). Can the evolution of plant defense lead to plant-herbivore mutualism?. *The American Naturalist*, 158(2), 109-123.

de Mazancourt, C., Loreau, M., & Dieckmann, U. L. F. (2005). Understanding mutualism when there is adaptation to the partner. *Journal of Ecology*, 93(2), 305-314.

Nelsen, M. P., Ree, R. H., & Moreau, C. S. (2018). Ant–plant interactions evolved through increasing interdependence. *Proceedings of the National Academy of Sciences*, 115(48), 12253-12258.

Stadler B, Dixon AFG (1999) Ant attendance in aphids: Why different degrees of myrmecophily? *Ecol Entomol* 24:363–369

Werner, G. D., Cornelissen, J. H., Cornwell, W. K., Soudzilovskaia, N. A., Kattge, J., West, S. A., & Kiers, E. T. (2018). Symbiont switching and alternative resource acquisition strategies drive mutualism breakdown. *Proceedings of the National Academy of Sciences*, 115(20), 5229-5234.

Reviewer #3 (Remarks to the Author):

Reviewers remarks to authors and editor

- What are the noteworthy results?

This exciting and densely packed manuscript introduces a novel mutualism theory. It proposes that mutualisms might arise from parasitism via the evolution of greater host defense rather than reduced parasite virulence.

I really like the fact that the parameters of the model can be estimated from empirical data, which makes it unfortunate that these are relegated to the supplementary material. Success of the model depends on both the ecological context and the specific shapes of the trade-off curves. These dependencies are informed by empirical observations of a specific system, studied by two of the co-authors, comprising the hawk moth *Manduca sexta*, and four plants with which it interacts in the Sonoran Desert.

- Will the work be of significance to the field and related fields? How does it compare to the established literature? If the work is not original, please provide relevant references.

The evolution of mutualism, and transitions between mutualism and pathogenicity, present major challenges to evolutionary biology. In addition to proposing a new way of thinking about the evolution of mutualism, the manuscript adds important ecological realism to a literature that has been dominated by simplistic evolutionary models of mutualism. I find especially compelling the insight that in a two-plant community, an alternative, less-defended, host plant can support a pollinating herbivore during the evolution of mutualism in a more highly defended plant.

- Does the work support the conclusions and claims, or is additional evidence needed?

The work seems to support the conclusions and claims, but could benefit from a clearer, more detailed description. For example, in the introduction, the authors mention night-blooming flowers as a trait that could attract and co-opt an insect herbivore to perform pollination services. Later, they describe modeling the evolution of floral scent as an attractive trait. Yet, they also assume that night-blooming has evolved and is a pre-requisite for the evolution of the pollinating mutualism. It would be helpful to further develop how night-blooming and scent-production interact in the model and in their thinking. For example, how might night-blooming arise and what selection pressures would maintain this trait? Also, of the plants they studied empirically, which bloom at night; which bloom during the day? Finally, is this the trait whose ecological effect is modeled in Equation S1.16?

Why does the interaction transition boundary in the ecological model not shift when the plant evolves night-blooming (e.g., Figure 2a and 2c versus 2b and 2d)? Night-blooming increases visitation rate, which comes with a pollination/oviposition trade-off. Is the trade-off scaled so that these boundaries don't shift? I really need more explanation of the statement in lines 108 – 110.

Why is the subsequent evolution of floral attractant different? How has the “resetting” of the “ecological context” caused this difference? Floral attractant, too, increases visitation rate, which involves the same pollination/oviposition trade-off. Is the assumption of a defense that entails a resource allocation cost, but is not a perfect trade-off, the thing that allows the mutualism to evolve?

The first step in the eco-evo model is that the plant shifts flowering from day to night, which increases visitation, which increases pollination and oviposition/herbivory (trade-off). In the second step, the plant increases floral volatiles, which further increases visitation, and therefore pollination and oviposition/herbivory. However, it is also assumed to increase defense compounds, which decrease herbivory, but not pollination. But then, the evolutionary model adds a genetic correlation between defense and fecundity – i.e., an investment cost of defense. Given its relevance to the initial hypothesis (evolution of greater host defense rather than reduced pathogen virulence), this trait is the most important one in the model. It is not just that plants have “co-opted” the pollinator, but they have also “tamed” it by evolving a defense, which is a way to get pollinated without suffering the cost of herbivory that comes with oviposition. Thus, the structure of the defense cost is crucial to the success of the model. The issue of defense also comes up in one of the bacterial examples the authors provide (lines 299 – 300). I'd like to see more discussion of the defense aspect of the model.

Minor nit-picking:

In line 294, I'm guessing that the term "specious" means something different than the authors think it does: <https://www.merriam-webster.com/dictionary/specious>. They are probably seeking the term, "speciose:" <https://glosbe.com/en/en/speciose>. However, substituting "richer" for "more speciose" would be even better, as "speciose" primarily refers to numbers of taxa within a taxonomic grouping whereas "richness" describes the number of species in a community.

Lines 370 – 371: Floral volatiles are also plant secondary compounds.

- Are there any flaws in the data analysis, interpretation and conclusions? - Do these prohibit publication or require revision?

I don't think there are flaws in the data analysis, interpretation and conclusions, but I would like to see additional description of the way that defense evolution is incorporated into the model.

- Is the methodology sound? Does the work meet the expected standards in your field?

I believe that the methodology is sound and meets the expected standards in my field, but due to my own limitations, I was unable to fully evaluate the mathematical model.

- Is there enough detail provided in the methods for the work to be reproduced?

Yes, if the modeling is a bit more clearly presented.

The modeling effort provides the ecological context in equations 3, which rather confusingly (and driven by the journal's manuscript structure of presenting methods after results) come after the resulting conditions for herbivore persistence (Equation 1), which are visualized in Figure 2.

I had trouble connecting the ecological model to its historical antecedents and so modified its presentation for my own edification. (Editor and/or authors can skip this or use it to determine if I really understood the model. I've put my specific questions/concerns in bold face.)

The ecological model expands a basic Lotka-Volterra predator-prey model to include predation by the larval moth live stage and mutualism by the adult moth life stage. The moth feeds on three different plant species (density of the i th plant species symbolized as P_i) as a larva (density on each plant species symbolized as L_i). The larvae from all the plant hosts combine to form a unified adult moth population (which grows at a rate of A moths per plant). In the equations below, there are four terms without an index. They are A (explained above), dA , which is the uniform adult moth mortality rate, H , which is a

half-saturation constant (of what, I wasn't sure until I sorted out the function), and ϵ , which accounts for an extra proportion of eggs that moths can produce if they feed on nectar from *Agave palmeri*, a plant on which they don't oviposit.

Equation 3a models population growth of four plant species (Did they, as stated in lines 337 – 338, actually model population growth of *Agave palmeri*? If so, does visitation affect its population growth rate?) with self-constrained increase as $P_i(1 - P_i)$, saturating increase due to mutualistic moth pollination (on the two *Datura* species) as $P_i [r_i v_i A / (H + v_i A)]$, and linear decrease due to predation (i.e., herbivory by larvae on *Proboscidea parviflora* and the two *Datura* species) as $h_i L_i P_i$. The per capita increase of plant density due to pollination by an adult moth is symbolized as r_i . The rate at which adult female moths visit each plant species is symbolized as v_i . The rate of plant population increase due to pollination by adult moth visitation is modeled as a saturating Michaelis-Menton function (e.g., $dP/dt = V_{max} S / [K_M + S]$), in which $r_i = V_{max}$, $v_i A = S$, and $H = K_M$). In this case, I think that H is the number of adult moths at which the plant fitness benefit from pollination is half of the total possible benefit provided by infinitely many adult moths. The authors show in the supplemental that for both *Datura* species, pollination benefits saturate with only a single moth visit, which provides an empirical argument for why this value need not vary among plant species. The reduction in plant population growth due to herbivory per larva is depicted as a linear function with slope h_i .

Equation 3b models larval population growth with plant population increase due to adult moths ovipositing onto plants as $P_i A [\epsilon e_i v_i]$. Decrease occurs via larval maturation to the adult stage, $m_i h_i L_i$, and natural larval death, $d L_i$. An adult female that visits a plant oviposits e_i eggs, but can increase her egg production by a fraction ϵ if she has nectared at *Agave palmeri*, a plant on which she will not oviposit. This value is roughly parameterized from empirical data. In the model, oviposition follows a linear function of moth visits, which reflects observations. However, in the supplemental, e_i is defined as the oviposition efficiency, which is the relationship between oviposition rate and floral visitation rate, v_i . This reader is still a bit confused about the meaning of e_i . Also, the authors do not provide empirical data to support a linear relationship between moth visits and plant number. Other literature shows that pollinator visits increase with plant density in complex ways related to the sizes of individual and group floral displays. Other research also finds that herbivores show density-dependent attraction to host plants, although I'm not sure if such functions are usually linear or saturating. Are such data available for any *Datura* species? The number of larvae maturing to adults is similarly assumed to increase linearly with the amount of feeding that larvae do. Given that Eq 3a gives plants a built-in self-limitation, how realistic are these linear functions? Parameter values for larval growth are determined empirically in the supplemental. I wonder, though, whether it is reasonable to assume that larval maturation efficiency is the same on the two plant species, even though their mortality rates differ? What factors cause larval mortality and how do they differ from factors influencing maturation rate?

Equation 3c models adult moth population growth as larval maturation to adult moths ($m_i h_i L_i$), summed over all host plant species, minus natural adult moth death as $d A A$. The summation in this equation accounts for the fact that moths can oviposit and mature larvae on more than one host plant species. Again, the number of larvae maturing to adults is assumed to increase linearly with the amount of

feeding that larvae do.

Important potential complexities: The a constant dA across host plant species assumes that all adult moths are equally fit, regardless of which host plant each fed on as a larva. The model also assumes that the rate at which adult moths visit host plant species is not affected by which host plant they fed on as a larva. How realistic are these assumptions in this system? Is it assumed that since adult life span is so much shorter than larval life span that these differences are trivial? It is also not clear to me how A can be scaled to give the adults recruited per plant of species i when A is the sum of adults recruited from all three potential larval host species.

To pick at nits, the quadratic form in Equation S1.3 should be defined as eigenvalues 2 and 3 (e.g., λ_2 and λ_3) rather than λ_2 , alone. There might be more than one convention for this, but I find that expressing them separately is less confusing. I can't speak to the solution of these two eigenvalues, but assuming that they are correct, then S1.4 follows and S1.5 (Equation 1 in the main text) is correct. I could not follow the model past equation S1.5, which should be evaluated by a more proficient reviewer.

I'd like the authors to provide a little more biological intuition to explain the structure of equation S1.6. This result is important, as it produces Equation 2 in the main text and also explains why the herbivore coexistence spaces (Figure 2a, 2c) do not overlap the mutualistic pollinator coexistence spaces (Figure 2b, 2d). This phenomenon yields a key result, which is that the evolution of plant traits that allow the insect to nectar beneficially (which I interpret to mean the transition to night-blooming flowers, although the authors might be referring here to the evolution of pollinator attractive scent – I'm not sure which and the distinction is important) does not inherently lead to a mutualism. Without understanding the origin of equation S1.6, I feel unable to verify this result.

Reviewer #4 (Remarks to the Author):

Despite that the mutualistic interactions are essential in most ecosystems, the current stage of knowledge about the evolutionary processes behind these interactions is far to be sufficient. So I highly welcome any well-founded hypothesis explaining the origin and evolutionary stability of mutualistic interactions. The paper by Johnson et al. presents such a hypothesis. They not only describe verbally the basic evolutionary process on how antagonism can evolve to mutualism, but they present a formal model and a simple sample community which fits even quantitatively to the model.

I think the Co-opted Antagonist Hypothesis is biologically justified and it is the new invention of the authors. Actually the authors suggest a two-step eco-evolutionary process where first, selection on a focal (plant) species favours the co-option of an antagonist (insect) to perform a potentially beneficial function. The second step is that the focal species evolves a series of phenotypic traits driving the interaction from antagonism to mutualism. The used mathematical tools and model assumptions are adequate, the analysis is correct, although at some points it is too succinct.

Despite my general positive opinion about the manuscript, there are some points which should be clarified before the publication. I listed my comments and suggestions below:

1. Lines 40-42. I suspect that insects are ideal for this two consumer integration since the different food intake in larvae and adult state. How can we explain the evolution of bat pollination? (The discussion mentions the bacterial-phage interaction as an alternative example.)
2. Lines 45-48. It would be nice to have information about the specialization of the central European Lepidopteras mentioned above. Is there any information for that?
3. Line 93. Some explanation for the relation (1) is needed, even if the details are in the supplementary. I suggest to introduce the used variables here, reader should understand the condition without reading the methods part, at least on intuitive level.
4. Eq. (2) is just the reverse of eq (1). Reader may ask again without reading the method and supplementary material why. Some more intuitive explanation may help. I think explanation below the equation is not enough.
5. Lines 149-151. What is the case if multiple plants and insects are coevolved. The model assumes only the evolution of plant species. I think this is the most critical assumption, which definitely should be explained and analysed in the revised version. Discussion mention it as a future step, but the critical question is whether the main conclusions remain valid after including evolution of pollinator.
6. Lines 326-327. Is there any indirect or direct argumentation for this assumption? The authors assume weak competition, but there is no competition between the plant species in the model. Are we sure that the results are robust against this assumption?
7. Line 248 mortality, Line 338. It should be $3bc$.
8. Line 362 It is rational to assume that increasing visiting rate because of pollination increases the oviposition as well, but some explanation is suggested. One more sentence is enough.
9. Line 363. The trade-off between r and h is linear. Does the result depend on this assumption? Concave and convex trade-offs can lead to different evolutionary routes.
10. Line 583. I suggest to denote the eigenvalues with $\lambda_{2,3}$, these are the additional two eigenvalues.
11. Lines 612. Another crucial thing that h' should be at least 30% greater than $r'h$ for the coexistence of the plant and insect. I suggest to mention this fact in the discussion, and to place in a biological context, if it is possible.
12. The mathematical analysis uses the standard adaptive dynamics. Please refer the adequate initial papers of this method.

I note that I am a theoretical biologist, being not familiar in the experimental studies so I cannot evaluate the merits and correctness of this part of the paper.

Response to Reviewers

Below we respond to each comment by the four Referees. We fully reproduce all comments in black italic text, paired with our responses in blue text. References to specific lines in the main text are highlighted.

Reviewer #1

In this manuscript, the authors propose a new hypothesis to explain how a mutualistic interaction can evolve from an antagonistic interaction. They use an eco-evolutionary model, based on an empirical system, to evaluate their Co-opted antagonist hypothesis.

The authors use a system of a insect species and its associated plant species. As an adult, the insect visits plants to nectar feed (and potentially pollinate), while the caterpillars feed on the leaves of the plant.

Hence, the interaction changes from an antagonist to a potential mutualist over the life-time of the insect.

In the model, initially the insects do not pollinate the plants and the interaction is therefore fully antagonistic. In the first step of the model, the plants evolve a trait that allows butterflies to pollinate it. However, this does not lead to a mutualism, since the costs of herbivory are still higher than the benefits of pollination. In the second step of the model, the host plant evolves attraction and defense traits, which then leads to the transition of the system from antagonism to mutualism. The authors show results for different type of plant communities (e.g., an additional nectar source).

In general, I really like the paper. It is well written, and I think of interest for many people. Even though there are many systems where an herbivorous insect pollinates some of its host plants, there are only a few studies investigating the evolutionary causes and consequences of such interactions. A particular strength of this manuscript is that the model is based on and parameterized for a biological system and therefore has a clear basis in biological reality. However, not all assumptions of the system will apply to other systems, and I therefore wonder how general the results of this study are. In particular, I wonder how important the ontogenetic change in interaction type is for the results. I understand that this study is a first step towards understanding the potential of the COA hypothesis, however, I think that the authors should be a bit more careful in generalizing their results.

I find the results in the manuscript complex and I think that the result section needs some rewriting to improve the readability of the manuscript. The authors assume a two-step process, where the host first evolves such that the antagonist can perform a beneficial function. After that, the host evolves to exploit the antagonist such that the interaction becomes mutualistic. It is not completely clear to me why a two-step process is assumed. Would the results be similar in case all three traits evolve simultaneously or not? And if not, how realistic is it to assume that first one trait evolves, and only later the other traits? Some justification and explanation is in my opinion needed for this assumption. In addition, not all model assumptions are clear and some need stronger justification. I have listed details below.

We thank the Reviewer for the compliments and have addressed the listed issues below.

1) The authors explore a wide variety of scenarios, which makes this study really interesting and complete. However, it is therefore also a bit difficult to follow what is happening. As far as I understand, first parameter r (effect of pollination) evolves in a single-plant system. After that, the authors study the co-evolution of attraction & defense in both a single-plant system, a two plant-system, and systems with an additional host plant and/or alternative nectar plant. So, the first step of the COA hypothesis is only studied in a single-plant system, while the second step is studied in a variety of communities. I think the result section could be improved by

a. Stating explicitly that step 1 is only studied in a single-plant system (plus maybe a justification why this is sufficient).

b. Stating explicitly that it is assumed that r is in the ESS of step 1, and does not evolve in step 2.

c. Being more specific of the ancestral trait in case of the additional host/nectar plant. Is an r -value assumed that evolved in the single-plant system? Would the r be different otherwise and would that affect the outcome?

1.1 Response:

We appreciate the Reviewer's point about the difficulty in following what was done in each of the scenarios. Based on this comment and similar comments from other Reviewers, we have revised the model so that all responses (benefits of pollination by the focal insect r_i , attraction v_i , and defense h_i) co-evolve simultaneously. In each scenario, an ancestral insect initially oviposits on, but does not pollinate, the focal plant species. We then model simultaneous evolution of plant traits that allow the focal insect to pollinate it and coevolution of attraction and defense from the coESS of the antagonistic interaction to the new coESS. We have added lines 76-94 to the Results as a "road-map" for readers.

2) *In addition, in the more complex systems it is not very clear if the two plants evolve simultaneously or if only one plant evolves while the other is in its ESS. Does it matter? Would co-evolution of the two plants change the results?*

1.2 Response:

Both plant species coevolve simultaneously (along with the focal insect; see Response 2.0). We depict coevolutionary dynamics of each plant species and the insect with the other plant species held at its final coESS in the two-plant species community in the new Figure 3 strictly for illustrative purposes. We have revised Figure 3 (lines 178-179) to make this clear. Simulations in which first one plant species and then the other coevolve with the insect yield the same coESSs.

3) *Initially, only parameter r evolves and parameters v and h are fixed. In step 2 parameter r is fixed while parameter v and h evolve. I think it would be helpful if this assumption is clarified in the main text to improve readability. Also some justification is needed why the authors assume a 2-step process. Would the results be very different if all parameters can evolve simultaneously? In addition, what happens to parameter r in case a mutualistic interaction has evolved?*

1.3 Response:

Based on this comment and similar comments by other Reviewers, we now allow all three parameters to evolve simultaneously in the revised model. We clarify this on lines 80-94 in the revised manuscript. The model results are similar to those reported in the original manuscript, with similar coESSs to the previous ESSs and the same ecological outcome (transitions from antagonism to mutualism). In the new model, r_i evolves from zero in the ancestral interactions to a non-zero coESS when a mutualism has evolved.

4) *All three trade-off functions (eq6, eq7b, p10) depend on the difference between the mutant trait and another value. It is not clear what the authors mean by these values. Are these the traits of the resident? Or are these the traits of the ancestral species? Some explanation and justification of the trade-offs is needed. In case the value is the trait of the resident, the trade-off disappears in the CSS, is this realistic?*

1.4 Response:

This is no longer an issue as this trade-off has been changed in the new model (see Response 2.0).

5) *In step 1 of the model, only parameter r (effect of pollination) evolves. The authors assume that an increase of pollination leads to a higher herbivory rate (eq 6, p10). What is the biological reason behind this assumption? Why would an increase in r leads to a higher herbivory rate?*

1.5 Response:

We have changed the evolutionary trade-offs in the new model (see Response 2.0), which addresses this issue. In the new model, r_i no longer trades off with a higher herbivory rate.

6) *In step 2, I think it is assumed that r (effect of pollination) has the value it evolved to in step 1. In step 1, however, an increase in r increased the herbivory rate due to the trade-off. It is unclear if this is still true in step 2. Some justification is needed in case the herbivory rate is set back to the value where $r = 0$.*

1.6 Response:

This is no longer an issue as this trade-off has been changed in the new model (see Response 2.0).

7) *In the discussion the authors do discuss the possibility of purging of the partner (L274-L280 and SI fig 4). In the absence of pollination ($r = 0$), I expect that it is possible for the plant to purge its partner as well by evolving v & h to values outside the interaction breakdown. A short explanation why the authors don't take this scenario into account would be helpful.*

1.7 Response:

The original model could not realistically consider the evolution of attraction and defense in the ancestral interaction due to the nature of the original trade-offs. In the revised model, we now model coevolution of the plants and the focal insect in the ancestral interaction so that the initial values of v_i and h_i are now from the coESSs from the ancestral interaction. In the new model, it is possible for the ancestral plants to purge the ancestral focal insect provided that the costs associated with increased defense and/or decreased attraction are not too extreme (see new Figure 5). We do not explicitly consider this scenario further here because the focal insect would no longer exist and the COA mechanism would not be possible.

8) *Related to my previous comment, the authors state in the discussion that the first step of COA reshapes selection acting on attraction and defense trait (L220-221). However, in the ms the authors do not study the evolution of these traits before the first step of COA. It is therefore not clear from the ms how the evolution of these traits is reshaped by the change in r .*

1.8 Response:

This is a great point. We now consider coevolution of the ancestral interactions in the revised manuscript (see Response 1.7), making it possible to show, mathematically, how the plant selection gradients change given evolution of r_i . Evolution of r_i (COA first step) fundamentally changes the plant selection gradient for attraction, as shown by Equation 13 vs. 11a in the revised manuscript (lines 495-524). The selection gradient for plant defense retains the same mathematical form as in the ancestral interactions; however, the COA first step nonetheless alters plant selection on defense by changing trait values and equilibrium densities in Equation 11b (see lines 522-524 in the revised manuscript).

9) *The authors predict that generalist adults feeding at multiple nectar sources may hinder the transition to mutualism by reducing pollination benefits (L47-48, L264). However, the way pollination is modelled, pollination benefits are not affected by alternative host plants, since adult visitation rates are not constrained and not affected by other plants. This predication does, in my opinion, not make much sense since it will, at least in this model, never happen.*

1.9 Response:

We thank the Reviewer for bringing this discrepancy to our attention. We have changed the prediction (lines 48-49) to: “generalist adults feeding at multiple nectar sources may hinder the transition by amplifying oviposition risk”, which is possible in our model.

10) *The authors study the evolutionary transition from antagonism to mutualism in a consumer species that has an ontogenetic shift from antagonism to a neutral or mutualistic interaction. How important is this ontogenetic shift for the results? Does the COA hypothesis also work in the absence of complex life cycles? The authors write (L316-L317) that their results are not specific to their specific system and could apply more broadly. I find this statement a bit bold, given that the studied system is quite specific.*

1.10 Response:

These are excellent points. In response to these comments and helpful comments by other Reviewers, we have derived a far more general model in which the focal partner lacks the stage-structured life cycle of *M. sexta*. The model is described in detail in the revised manuscript (lines 602-627). To derive this more general model and compare its results with those presented in the revised manuscript, we ‘collapsed’ our ecological model (Eq. 3) via timescale separation. We assumed that adult insects are very short-lived relative to the plants and insect larvae so that adult density attains its equilibrium on the timescale of the ecological dynamics. This approach allows us to derive composite parameters for the general model that

encompass parameter estimates from our original model, which allows us to compare coESSs from the general model with appropriately-scaled parameter estimates. Supplementary Table 4 shows that the general model yields coESSs that are broadly similar to those presented in the revised manuscript. We briefly discuss the more general model in the Discussion of the revised manuscript (Lines 328-332).

11) I am not sure if this is relevant, but would it be helpful to show how evolution of plant traits affect the pollinator dynamics? Does the increase in plant density somehow benefit the insect and does this in turn have a positive effect on plant densities? The eco-evolutionary dynamics are hidden in the manuscript, while it might help readers to understand what is going on.

1.11 Response:

We thank the Reviewer for the excellent suggestion. We include plots of plant and insect equilibrium densities and the coevolutionary dynamics over evolutionary time in all of the new Figures. We discuss feedbacks between ecological density and coevolutionary dynamics throughout the revised Results.

Minor comments

In figure 2 (p4) the authors show that the predicted ESS is very close to the ESS based on data (cool!). I wonder if there is some information on the ESS in case of an additional host/nectar plant and if the prediction is equally good.

Response:

Unfortunately, we do not have the data to validate the coESSs of attraction for interactions involving the additional host plant or nectar resource as we did not include these plants in our greenhouse experiments.

The meaning of the white dots is ambiguous. Sometimes the authors refer to them as the ancestral traits (e.g., L120-121), while sometimes it is referred to as the ancestral ESS (e.g., L139). Is it an ESS? If yes, it seems that these results are not in the manuscript.

Response:

In the revised manuscript, all white dots refer to coESSs of ancestral antagonistic interactions.

L338 Should this not be equation 3 instead of 1?

Response:

Yes, we have made this change in the revised manuscript.

L339-L340 ‘Which is scaled such that A gives the adults recruited per plant of species i’. How is this scaled?

Response:

In the revised manuscript, adult density is no longer scaled per plant of species *i*.

*L512 should it not be $P_{\hat{i}} = P_i * q_i$?*

Response:

This typo has been corrected.

L513 According to the text herbivory rate is scaled, but the equation suggests that maturation is scaled.

Response:

We have added a subsection to the Methods (“Model scaling”; lines 371-380) in the revised manuscript to clarify this issue. Both herbivory rate and maturation are scaled in the model.

L513 Time is scaled as well (L548), this should be explained here.

Response:

The model is scaled for purposes of parameterization but is not non-dimensionalized. Thus, time remains unscaled. We clarify this issue in the “Model scaling” section (lines 371-380) of the revised manuscript.

L516 From the text it is not clear if estimated parameters are the scaled or the unscaled values.

Response:

All estimated parameters are the scaled quantities. We clarify this issue in the “Model scaling” section (lines 378-380) of the revised manuscript.

L538 ‘We vary h in ecological models’. What do the authors mean with the ecological models? There are only results from eco-evo models as far as I can see.

Response:

We have removed this sentence from the revised manuscript.

P20/Sup Fig 1: It is unclear what the authors plotted in panels c-f. What are the dark areas? What does this plot tell?

Response:

We have removed this Supplementary Figure from in the revised manuscript.

P22/Sup Fig 3: There is no reference in the text to this figure.

Response:

This figure has been removed from the revised manuscript.

Reviewer #2

*In “Evolutionary transitions from antagonism to mutualism explained by the Co-Opted Antagonist Hypothesis” Johnson et al propose a model of mutualism evolution from antagonism. In this model, a plant initially begins to receive some small benefits (pollination) from a herbivore, but largely suffers defoliation costs. They show that selection on either pollination amount or two traits that separately influence pollination amount in the plant (defense and floral attraction) could drive an evolutionary trajectory that reaches a stable mutualistic state instead of continued antagonism or total breakdown of the interaction. The authors develop an explicit model, which they parameterize with data from a real system (*Manduca sexta* and *Datura* spp), and include a number of relevant extensions.*

This work has one very key strength which I want to emphasize before anything else. While the evolution of breakdown vs. maintenance in mutualisms is highly studied, the evolution of mutualism origin in the first place is under-represented in the literature (though perhaps not as woefully as the authors paint it). Further, explicit mathematical models considering population dynamics (here, e.g. plant density), are generally under-represented for mutualisms even without considering evolution – so this contribution fills two-part literature gap. Also, the text is generally well-written, and really quite exciting to read. Despite the criticisms I levy below, I think this article has a good likelihood of being influential in the field.

The authors' treatment of the real system is no less than excellent: they have considered many aspects of relevant biology here, from ecological constraints on traits to even heterospecific pollen. Yet in both the introduction and discussion, it's clear that this mechanism is likely to be a wider phenomenon than just for Lepidoptera and hosts. It could very much extend the impact of this paper to provide at least one generalized presentation of the model -- as the literature gap it fills extends across mutualism types.

The only true major issue I see is that of insect evolution or co-evolution. Here, evolution (or even behavioral variation-- i.e. insects choosing to oviposit on less defended hosts) in the insect is not considered, except in sudden host shifts. However, it seems confusing to consider only plant evolution when insects are likely to have higher population sizes, and faster generation times – traits that suggest they might do most of the evolution in co-evolutionary scenarios (e.g. Gandon & Michalakis 2002, Bergstrom & Lachmann 2003). Further, the influence of insect evolution could lead to opposite outcomes. As the plant increases herbivory defense and decreases attraction, the obvious insect evolutionary

responses (increased tolerance of secondary compounds and increased sensitivity to olfactory cues for finding hosts) are likely to escalate antagonism (push defense down and further into the antagonistic outcome zone). Indeed, *M. sexta* is unaffected by the extreme chemical defense of *D. wrightii*, which seems certain to be an evolutionary response. On the other hand, from a state where hawkmoths nectar exclusively on non-*Datura* species, a hawkmoth mutant that is able to nectar on the plant it oviposits on may have higher relative fitness through more efficient resource use, mitigating plant fitness costs as a byproduct –e.g. the reverse co-option model.

2.0 Response:

We thank the reviewer for the compliments. Motivated by this comment and similar comments from other Reviewers, we have now developed a coevolutionary model, which we present in the revised manuscript. We will briefly outline the model here, which we discuss in detail in the revised Methods (lines 414-524). In the model, plants evolve traits that allow the focal insect to pollinate it, r_i . Simultaneously, visitation rate v_i and herbivory rate h_i coevolve due to selection on plant trait x^{mut} and insect trait y^{mut} . Evolution of plant traits (Eq. 7; 8a-c) involve trade-offs with the competitive ability of mutant plants (Eq. 8d), while evolution of insect traits (Eq. 9; 10a,b) involve trade-offs with oviposition efficiency (Eq. 10c). We then quantify the selection gradients (Eqs. 11-13 for the plants) and predict coESSs (Figs. 2-4) for both the ancestral interactions and coevolved mutualisms. Given our evolutionary parameter values (lines 570-600), the model predicts similar coESSs as the ESSs in the original manuscript and the same ecological outcomes (i.e., transitions from antagonism to mutualism) as in the original manuscript.

Four minor points:

1) I am confused as to why this model has two separate stages. Seems like in the real world co-option would proceed from increasing floral attraction from 0 or a low value, while leaf attraction would be at some high value. Further, the stages seem to differ in the model only slightly: in the first stage pollination is a trait that mutates on its own, whereas in the second it becomes a function of correlated impacts of two mutant traits (defense and attraction). As a sidenote here, I'd like to point out that the correlated traits aspect of stage 2 allows the inclusion of constraint and is one of the neat features of this model.

2.1 Response:

Based on this comment and those from other Reviewers, all three responses now coevolve simultaneously in the revised manuscript (see Response 1.1). Thus, while the COA Hypothesis involves a two-step eco-coevolutionary process because (step 1) evolution of plant traits that allow the insect to pollinate it is necessary to reset the ecological context of the interaction and reshape selection on (step 2) the coevolution of attraction and defense, all three responses now coevolve simultaneously in the model.

2) The methods are not clear in places. Much of the explanation for the model and math is in the supplemental text, yet most is key to a basic understanding of the model. Despite careful reading, I am left with a few confusions about what was actually done and what wasn't, though probably only because it exists chopped up into so many different places. Because of this, it is hard for the reader to distinguish ecology only or eco-evo models, figures, and results.

2.2 Response:

We appreciate the Reviewer's point about the difficulty in following the explanations of the model. In the revised manuscript, we have consolidated all sections describing the model to the Methods section of the main text. We have also clearly labeled subsections of the Methods and worked to improve clarity throughout the Methods section of the revised manuscript.

3) This model is somewhat similar to two existing ideas in the literature. For one (evolution of mutualism from vertically or semi-vertically transmitted parasites/antagonists), the authors distinguish their model, but perhaps underemphasize previous work – e.g. suggestion of evolution of ant plant defense from herbivory (Nelson et al 2018) or of ant-hemipteran defense from ant-hemipteran predation (Stadler & Dixon 1999). Yet for another, even more similar idea (evolution of mutualism from herbivory via

tolerance – de Mazancourt et al 2001, 2005) the authors do not. I note that a generalized case of the authors' model as I have suggested could provide a unifying framework across these systems.

2.3 Response:

We thank the Reviewer for the references. We have followed the Reviewer's suggestion and comments by other Reviewers to develop a more general eco-coevolutionary model (see Response 1.10), which we present in detail in the Methods (lines 602-627) of the revised manuscript. We discuss how this general model could provide a unifying framework as suggested by the Reviewer and include the suggested references in the Discussion (lines 328-332) of the revised manuscript.

4) As a slightly different mechanism, could the timing of flowering and herbivory actually induce the plant fitness benefit of nectaring by the herbivore? An early herbivore that also nectars could reduce fitness benefits of late pollinators because the eating/developing larva destroy later developing leaf and flower tissue. Related to my comment for line 364.

2.4 Response:

This is an interesting point. We agree that an early herbivore that pollinates might reduce the fitness benefits of late pollinators by damaging plants and making them less attractive to other pollinators. We highlight this possibility in the revised manuscript (lines 306-307).

Line by line notes, mostly examples of points above:

Just to quickly note: The readme file for the code looks good, and notebooks seem more than adequate for repeating analyses. I have not checked them in Wolfram Mathematica, as I am not familiar, nor do I currently have access to a machine I can install it on (COVID-19 related)

Response:

We recognize that Mathematica is not freely available and appreciate that the Reviewer looked through the notebooks. We have worked diligently to make the codes clear and well-annotated.

Line 82: Why "Given" when it is modeled?

Response:

We use this qualifier as an ancestral plant species in the model is not, in principle, guaranteed to evolve traits that allow the focal insect to pollinate it. When evolutionary constraints are too high, for example, the ancestral antagonism is retained (Figure 5).

Line 98: It took me 3 read-throughs to realize that "Co-option of a herbivore as a pollinator" models are not in Figure 2, even though the paragraph cites Figure 2 panels. Also, this section seems to mostly discuss trait models, even though this section isn't ostensibly about trait models.

Response:

We apologize for the confusion. Because all three responses evolve simultaneously in the new model (Response 1.1), there is no longer a separate "co-option of a herbivore as a pollinator" model. In the "COA step 1" section of the revised Results (lines 104-123), we focus on the ecological conditions for mutualism to arise (Eq. 2) and highlight the conflict of interest between plants and the insect that it captures. We then show that co-option of the herbivore as a pollinator does not, in and of itself, lead to a mutualism because the phenotype of any viable ancestral interactions always remains in the antagonistic region of Figure 2. The integration of the antagonist as a pollinator, however, changes the ecology of the interaction and reshapes selection acting on attraction and defense (COA second step).

Line 125: For panels g and h, I don't know what ESS of "other" species means when the ESS clearly depend on each other (shifted from e and f) – does this mean that one species is held at its single-species ESS? And only one plant evolves in the two plant community?

Response:

The plant species coevolve simultaneously in the two-plant species community. We depict coevolutionary dynamics for each plant species (and the insect) while holding the other plant species at its final coESS in the two-plant species community in the new Figure 3 strictly for illustrative purposes. We have revised the caption of Figure 3 (lines 178-179) to make this point clear.

Also Figure 2: Given that the model is parameterized with data from the experiment, is it really exciting that the ESS matches traits measured in that same experiment? And, if that is exciting, why aren't the measured traits also reported in Figure 3?

Response:

The model is independently validated with data that were not used to parameterize it. Thus, our data do provide a strong validation of the model predictions and is one of the novel and exciting features of our modeling framework. Unfortunately, we do not have data to validate attraction for interactions involving the alternative host plant or nectar resource as we did not include these plants in our experiments.

Line 126: The concept of isocline isn't explained, unclear what it means, especially in contrast to the arrows in the figures.

Response:

We have added the subsection “Model analyses” to the Methods of the revised manuscript, in which we now discuss the evolutionary isoclines (lines 487-488). Note that the isoclines have been removed from the Figures for illustrative purposes in the revised manuscript.

Line 140: Just wondering generally if there's any evidence to support ancestral antagonism in this system.

Response:

This is an interesting question. We do not have a direct answer, but *Manduca* species as a whole are specialists on host plants in the Solanaceae, whereas Solanaceous species have a very wide range of pollinators. That suggests that the ancestral *Manduca* was probably feeding on leaves of an ancestral *Datura* or *Datura* relative. We do not know what it was using as a nectaring plant.

Line 146: Is it really independent? Seems the parameterization came from the same experiment.

Response:

Visitation rate (v_i) and herbivory rate (h_i) were not parameterized in the model, but allowed to evolve. Thus, our experimental data used to validate predictions for v_i and h_i at the coESSs is independent from the data used to parameterize the model. We discuss model validation in the Results (lines 153-163) and have added distinct subsections “Model parameterization” and “Model validation” to the Methods of the revised manuscript (lines 526-568) to make this point clear.

Line 147: Extinction how? Do stochastic forces push it across into the trait space where the interaction breaks down? It should be specified.

Response:

The coESS of the two-plant species community (Fig. 3) is in the interaction breakdown region (insect extinction) of the parameter space for the one-plant species community (Fig. 2). Thus, if evolution had led to this coESS in the one-plant species community, it would have driven the evolutionary purging of the focal insect. We clarify this issue in the revised version of the manuscript (lines 155-157).

Line 161: When adding a host or a nectar source, why do we assume amelioration of herbivory? Couldn't the reverse be true? (i.e. apparent competition – two hosts support greater than additive larval density and therefore more costs?) In contrast, line 173 I understand well and is a very clear point.

Response:

To be clear: we predicted the amelioration of herbivory, but it was not assumed in the model. Apparent competition is indeed possible in the model, so amelioration of herbivory is not guaranteed in the model.

The Reviewer's point, however, is well taken, so we removed the sentence causing confusion (formerly line 161) in favor of the explanation that the Reviewer found clear (formerly line 173).

Line 168: Should this be "attraction"? I find the distinction between "attraction" and "visitation" to be confusing generally.

Response:

This sentence has been removed from the revised manuscript. Following the Reviewer's comment on the distinction between "attraction" and "visitation", we have worked to avoid this confusion in the revised manuscript by generally using the term "attraction" and specifying "visitation rate" only when necessary to refer to the specific model parameter v_i .

Line 169: When the insect was initially generalized, where is the starting state on the traits? And why?

Response:

Following this comment and similar comments from other Reviewers, we now start each scenario with an ancestral insect that initially oviposits on, but does not pollinate, the focal plant species. Given our new model, the starting state of each scenario in the revised manuscript (white points in Figs. 2-4) is now given by the coESS of the ancestral antagonistic interaction.

Line 181: Took me a long time to get exactly what happens here: first evolution to ESS in a single species (which authors have argued is unstable, line 147); then when porting those plant trait coordinates to the new ecological context of a host shift (Fig 3 upper), the interaction is again antagonism, but eco-evo dynamics then push it back to mutualism again (Fig 3 lower). Could use more explanation.

Response:

We agree and based on this comment and similar comments from other Reviewers, we have greatly revised how we present each of the scenarios (see Response 1.1).

*Figure 3: Is *D. discolor* still in the models? If so, as an evolving or non-evolving host? Not clear. Here is also where I noticed that Supplementary Table 3 suggests there are models with all four plants, yet no results/discussion for them?*

Response:

In the revised manuscript, the results for *D. discolor* with an alternative host plant or nectar source are now presented in Figure 4c,d. We have revised Supplementary Table 3 accordingly as we do not discuss the models with all four plants in the manuscript.

In Figure 2 and 3, I found it hard on first couple read-throughs to identify ecology only and evolutionary scenarios. I think it's because there are "initial values" on the ecology only models that just map the interaction outcomes to regions of trait-space - e.g. there are no evolving values in those panels, correct?

Response:

Following this comment and similar comments from other Reviewers, we have simplified the Figures in the revised manuscript by removing many of the subpanels. Technically, there are no "ecology only" results shown in the Figures of the revised manuscript because evolution now determines both the interaction breakdown and transition boundaries as well as the coESS of both the ancestral interaction and the final interaction. In the revised manuscript, all white and black points in the Figures now refer to the coESSs of the ancestral interactions and coevolved mutualisms, respectively.

*Line 248: These arguments are confusing. How can *D. wrightii* be both better defended (defined as less eaten which is tied to less larval development) and a better host (more larval development) within the confines of the model? To me, this suggests that co-evolution matters.*

Response:

We entirely agree that this counterintuitive result suggests that coevolution matters. Intriguingly, our new coevolutionary model reinforces this result, which is also supported to a degree by our experimental data. While herbivory rate is lower on *D. wrightii* (leading to lower larval maturation into adults; $m_i h_i L_i$ in our model), *M. sexta* also enjoys greater maturation efficiency ($m_w > m_d$) and lower larval mortality ($d_w < d_d$) on *D. wrightii* in the new model. As a result, *M. sexta* density is actually greater on *D. wrightii* than on *D. discolor* in the model (see Fig. 2e,f).

From an empirical perspective, how well defended *D. wrightii* is compared to *D. discolor* remains a fairly open question, but it seems likely that *D. wrightii* is better defended. More generally, it is somewhat unclear whether *M. sexta*'s lower herbivory rate on *D. wrightii* than on *D. discolor* in our experiments is due to plant defense or plant quality. There is some evidence that *M. sexta* herbivory might be greater on *D. discolor* due to compensatory feeding, suggesting that *D. discolor* is a lower-quality food source for *M. sexta* than is *D. wrightii*. We incorporate this knowledge of the system, to a degree, into the model via lower maturation efficiency and higher larval mortality on *D. discolor* than on *D. wrightii*, as discussed in the paragraph above and on lines 252-253 in the revised manuscript.

Line 250: This paragraph seems to fly in the face of the initial justification (that this model is great because it's not vertical transmission) Further, "vertical" here is not that more attractive plants interact with better pollinators, they just interact with all pollinators more...Are these really the same idea as vertical transmission and partner fidelity as used in the literature?

Response:

While the Reviewer's points are well taken, we disagree that this paragraph flies in the face of a key justification for the model. First, to clarify: more attractive plants in our model interact more with the focal insect, not all pollinators. More broadly, the point of this paragraph (lines 254-261) is that COA extends the ideas of virulence theory, while resolving some of its difficulties. For example, host fidelity in virulence theory is achieved by vertical transmission, but this mechanism is not general to other systems, such as pollination mutualisms. Thus, a challenge for the evolution of mutualism in many natural systems is how to achieve some form of partner fidelity without strict vertical transmission. COA resolves this challenge via evolutionary co-option of the focal partner. We therefore argue that our framework relates to these ideas in the literature by resolving the key challenge of partner fidelity without strict vertical transmission for the evolution of mutualism.

Line 268: This paragraph is a great contribution. Plus this fits with other literature (see Chomicki et al 2017; Werner et al 2018) where breakdown is generally loss. And is a key example of why generalizing the model & introduction would be very useful to the mutualism research community writ large.

Response:

We thank the Reviewer for the compliment and the references, which we discuss in the context of mutualism breakdown (lines 276-277). In response to this comment and comments of other Reviewers, (Response 1.10) we have derived a general model in the revised manuscript (lines 328-332; 602-627).

Line 277: P_i was not defined previously, thus its not clear what this means.

Response:

These references to P_i have been removed from the revised manuscript.

Line 281: This paragraph is about how often the fitness benefits of pollination by an antagonist would be high enough. Is there anything known on how often host plants where herbivores also nectar are pollen limited?

Response:

This is an interesting question. In general, we do not know how often plants that interact with herbivorous pollinators are pollen limited. It seems likely that *Datura* plants in nature are often pollen limited because they have the ability to readily autonomously self-pollinate, perhaps as an adaptation for pollen limitation.

I didn't fully understand lines 295-303, but I think probably only because I don't understand the example. I think maybe some detail is missing? Or maybe the wording has me confused.

Response:

We apologize for the confusion. We have removed this example for clarity in the revised manuscript.

Line 342: What is the half saturation constant for? I intuit from recalling predator feeding equations, that it could account for re-visitation of already visited plants? But that feels wrong given later text...

Response:

H is the saturation constant for pollination, reflecting that pollination benefits (viable seed set of insect-pollinated flowers) saturate rapidly with increasing floral visits (Supplementary Data 1).

Line 364 and 378-385. This is great consideration of trade-offs/epistatic links between attraction/herbivory and defense/repulsion and how they influence the evolution of mutualism, but aren't there other relevant mechanisms that would do the same thing – e.g. if single chemicals are both defensive and pollinator repulsive (Kessler 2011)? Interestingly, it seems to me that such a mechanism could induce reliance on hawkmoths as pollinators in the first place – if herbivory in any way makes plants less attractive to other pollinators. I'm also curious as to how this trade-off affects the relationship between attraction and realized defense inherent defense traits (after accounting for D_v , D_r) versus v and h ?

Response:

The Reviewer makes an interesting point about other mechanisms linking attraction and defense, such as chemicals that are both defensive and pollinator repulsive. In the new coevolutionary model, plant traits driving attraction and defense could, in principle, capture the Reviewer's mechanism if we set $X^V = X^H$ (Eqs. 7-10). We model plant traits affecting attraction and defense separately in the model for generality.

Line 485: How can there be cross-pollination if there is only one individual per species?

Response:

There were two individuals of each plant species in the arrays in our greenhouse experiment. We clarify this point in the revised manuscript (line 737-740).

Line 488: I was excited to see that the authors considered the possible role / cost / influence of avoiding heterospecific pollen. That really demonstrates a thorough thinking about the system. However this sentence confused me. The words on the page imply that good viable seeds are formed from the transfer of only heterospecific pollen (really?). Yet, I want to understand that adding heterospecific pollen doesn't interfere with the production of seeds from conspecific pollen. Which is it?

Response:

Seed set for both *Datura* species is greater for insect-visited flowers than for autonomously self-pollinated flowers. Transfer of heterospecific pollen slightly *reduced* seed set for both species; i.e., there was a trend towards lower seed set in flowers receiving heterospecific pollen (visited by insects that had previously visited heterospecific flowers) than flowers receiving only conspecific pollen (visited by insects that had not visited heterospecific flowers), but the difference was not statistically significant (lines 757-762).

Line 524: For which models exactly are v_w and v_d set to these values? I'm confused.

Response:

In the new model, these parameters evolve in all models, and are validated by our empirical data, as we now discuss in the "Model validation" section of the revised manuscript (lines 513-528).

Line 531: Does the $h_w = 1$ and $h_d = 2$ relationship assume plant species and individuals are equal sizes? since herbivory subtracts from density which is individuals? or is density foliage based?

Response:

This is a good point. We made cursory estimates of *D. wrightii* and *D. discolor* leaf biomass consumed by *M. sexta* larvae in our experiment. We clarify this in the revised manuscript (lines 565-566).

Line 560: Sorry if I'm being dense, but I don't understand the "double" and "half" explanations. If 80% of larvae survived on P. parviflora, that means 20% died, and if survivorship on P. parviflora is twice that of D. wrightii that would be 40% survive on D. wrightii (and 60% die). But 60% is not double 20%.

Response:

This has been resolved in the estimation of larval mortality (lines 546-547) in the revised manuscript.

Reviewer #3

- What are the noteworthy results?

This exciting and densely packed manuscript introduces a novel mutualism theory. It proposes that mutualisms might arise from parasitism via the evolution of greater host defense rather than reduced parasite virulence.

*I really like the fact that the parameters of the model can be estimated from empirical data, which makes it unfortunate that these are relegated to the supplementary material. Success of the model depends on both the ecological context and the specific shapes of the trade-off curves. These dependencies are informed by empirical observations of a specific system, studied by two of the co-authors, comprising the hawkmoth *Manduca sexta*, and four plants with which it interacts in the Sonoran Desert.*

3.1 Response:

We thank the Reviewer for the complements. In response to the Reviewer's comment and similar comments by other Reviewers, we have moved the discussion of empirical data to the "Model parameterization" and "Model validation" subsections of the revised manuscript (lines 526-568).

- Will the work be of significance to the field and related fields? How does it compare to the established literature? If the work is not original, please provide relevant references.

The evolution of mutualism, and transitions between mutualism and pathogenicity, present major challenges to evolutionary biology. In addition to proposing a new way of thinking about the evolution of mutualism, the manuscript adds important ecological realism to a literature that has been dominated by simplistic evolutionary models of mutualism. I find especially compelling the insight that in a two-plant community, an alternative, less-defended, host plant can support a pollinating herbivore during the evolution of mutualism in a more highly defended plant.

3.2 Response:

We thank the Reviewer for the complements, particularly about the alternative host plant model.

- Does the work support the conclusions and claims, or is additional evidence needed?

The work seems to support the conclusions and claims, but could benefit from a clearer, more detailed description. For example, in the introduction, the authors mention night-blooming flowers as a trait that could attract and co-opt an insect herbivore to perform pollination services. Later, they describe modeling the evolution of floral scent as an attractive trait. Yet, they also assume that night-blooming has evolved and is a pre-requisite for the evolution of the pollinating mutualism. It would be helpful to further develop how night-blooming and scent-production interact in the model and in their thinking. For example, how might night-blooming arise and what selection pressures would maintain this trait? Also, of the plants they studied empirically, which bloom at night; which bloom during the day? Finally, is this the trait whose ecological effect is modeled in Equation S1.16?

3.3 Response:

We agree with the Reviewer that using the example of night-blooming as a trait that could co-opt the insect creates confusion given that we also model the evolution of attraction. To avoid this confusion, we use flower size in the revised manuscript as an example of a plant trait that could allow the focal insect to

pollinate it, and thus be co-opted as a pollinator (lines 40-41). (For reference, both *Datura* species flower mainly at night, but can be open for part of the morning). Also, there is no Equation S1.16 in the original manuscript (perhaps this question was about Equation S1.6?). Regardless, the equations presented in Supplementary Equations 1 in the original manuscript have been heavily revised and included in the revised Methods section. Finally, to be clear, evolution of plant traits that allow the focal insect to pollinate it (r_i) is not the same as evolution of attraction (v_i). We have added lines 76-94 in the revised Results to clarify this issue and to provide a ‘road map’ for readers.

Why does the interaction transition boundary in the ecological model not shift when the plant evolves night-blooming (e.g., Figure 2a and 2c versus 2b and 2d)? Night-blooming increases visitation rate, which comes with a pollination/oviposition trade-off. Is the trade-off scaled so that these boundaries don't shift? I really need more explanation of the statement in lines 108 – 110.

3.4 Response:

In the model, the evolution of pollination benefits (r_i) expands the mutualistic regions of Figure 2 by reducing the interaction breakdown boundary (solid black lines in Figure 2). Thus, it is the interaction breakdown boundary, not the interaction transition boundary, that shifts as the plant evolves traits that allow the focal insect to pollinate it. Equation 2 is the interaction transition boundary upon evolutionary co-option of the focal insect, which is the exact reverse of Equation 1. Thus, evolution of r_i (COA step 1), in and of itself, cannot lead to mutualism. We have revised both the Results (lines 125-135) and Figure legends (lines 141-144) to clarify these issues.

Why is the subsequent evolution of floral attractant different? How has the “resetting” of the “ecological context” caused this difference? Floral attractant, too, increases visitation rate, which involves the same pollination/oviposition trade-off. Is the assumption of a defense that entails a resource allocation cost, but is not a perfect trade-off, the thing that allows the mutualism to evolve?

3.5 Response:

We apologize for the confusion. To be clear, evolution of plant traits that allow the focal insect to pollinate it (r_i) is not the same as evolution of attraction (v_i). We have added lines 76-94 in the revised Results to clarify this issue. We also show, in greater mathematical detail, how evolution of plant traits that allow the insect to pollinate it reshapes selection on attraction and defense, from the initial antagonistic interaction to the final coESS in the revised Methods (lines 495-524). Finally, while plant trade-offs have changed in the new model, defense given a resource allocation cost is not, in and of itself, what allowed the mutualism to evolve in the original model, but rather the joint evolution of attraction and defense following co-option.

The first step in the eco-evo model is that the plant shifts flowering from day to night, which increases visitation, which increases pollination and oviposition/herbivory (trade-off). In the second step, the plant increases floral volatiles, which further increases visitation, and therefore pollination and oviposition/herbivory. However, it is also assumed to increase defense compounds, which decrease herbivory, but not pollination. But then, the evolutionary model adds a genetic correlation between defense and fecundity – i.e., an investment cost of defense. Given its relevance to the initial hypothesis (evolution of greater host defense rather than reduced pathogen virulence), this trait is the most important one in the model. It is not just that plants have “co-opted” the pollinator, but they have also “tamed” it by evolving a defense, which is a way to get pollinated without suffering the cost of herbivory that comes with oviposition. Thus, the structure of the defense cost is crucial to the success of the model. The issue of defense also comes up in one of the bacterial examples the authors provide (lines 299 – 300). I'd like to see more discussion of the defense aspect of the model.

3.6 Response:

The Reviewer makes an excellent point about the plants “taming” the focal insect via defense evolution as well as co-opting it. Given this comment and similar comments by other Reviewers, we now model the

coevolution of defense with both plant defensive traits and insect traits to overcome plant defenses (see Response 2.0). We discuss coevolution of defense in greater detail throughout the revised manuscript.

- Are there any flaws in the data analysis, interpretation and conclusions? - Do these prohibit publication or require revision?

I don't think there are flaws in the data analysis, interpretation and conclusions, but I would like to see additional description of the way that defense evolution is incorporated into the model.

Response:

We think that the new coevolutionary model greatly improves how defense evolution is modeled (see Response 2.0). We discuss coevolution of defense in greater detail throughout the revised manuscript.

- Is the methodology sound? Does the work meet the expected standards in your field?

I believe that the methodology is sound and meets the expected standards in my field, but due to my own limitations, I was unable to fully evaluate the mathematical model.

Response:

We appreciate the Reviewer's effort to evaluate the mathematical model and have taken the Reviewer's comments into consideration when developing the new coevolutionary model.

- Is there enough detail provided in the methods for the work to be reproduced?

Yes, if the modeling is a bit more clearly presented. The modeling effort provides the ecological context in equations 3, which rather confusingly (and driven by the journal's manuscript structure of presenting methods after results) come after the resulting conditions for herbivore persistence (Equation 1), which are visualized in Figure 2.

I had trouble connecting the ecological model to its historical antecedents and so modified its presentation for my own edification. (Editor and/or authors can skip this or use it to determine if I really understood the model. I've put my specific questions/concerns in bold face.)

*The ecological model expands a basic Lotka-Volterra predator-prey model to include predation by the larval insect live stage and mutualism by the adult insect life stage. The insect feeds on three different plant species (density of the i th plant species symbolized as P_i) as a larva (density on each plant species symbolized as L_i). The larvae from all the plant hosts combine to form a unified adult insect population (which grows at a rate of A insects per plant). In the equations below, there are four terms without an index. They are A (explained above), dA , which is the uniform adult insect mortality rate, H , which is a half-saturation constant (of what, I wasn't sure until I sorted out the function), and ϵ , which accounts for an extra proportion of eggs that insects can produce if they feed on nectar from *Agave palmeri*, a plant on which they don't oviposit.*

*Equation 3a models population growth of four plant species (Did they, as stated in lines 337 – 338, actually model population growth of *Agave palmeri*? If so, does visitation affect its population growth rate?) with self-constrained increase as $P_i(1 - P_i)$, saturating increase due to mutualistic insect pollination (on the two *Datura* species) as $P_i [r_i v_i A / (H + v_i A)]$, and linear decrease due to predation (i.e., herbivory by larvae on *Proboscidea parviflora* and the two *Datura* species) as $h_i L_i P_i$. The per capita increase of plant density due to pollination by an adult insect is symbolized as r_i . The rate at which adult female insects visit each plant species is symbolized as v_i . The rate of plant population increase due to pollination by adult insect visitation is modeled as a saturating Michaelis-Menton function (e.g., $dP/dt = V_{max} S / [K_M + S]$, in which $r_i = V_{max}$, $v_i A = S$, and $H = K_M$). In this case, I think that H is the number of adult insects at which the plant fitness benefit from pollination is half of the total possible benefit provided by infinitely many adult insects. The authors show in the supplemental that for both *Datura* species, pollination benefits saturate with only a single insect visit, which provides an empirical argument*

for why this value need not vary among plant species. The reduction in plant population growth due to herbivory per larva is depicted as a linear function with slope h_i .

Response:

We appreciate the Reviewer's efforts to fully understand the model, which we describe in greater detail in the revised Methods (lines 346-369). To improve clarity, we have also simplified Equations 1 and 2 in the revised manuscript, which we now describe more fully in the main text (lines 96-110). Note that *M. sexta* visitation does not affect population growth of *A. palmeri* as *M. sexta* does not pollinate nor consume it.

*Equation 3b models larval population growth with plant population increase due to adult insects ovipositing onto plants as $P_i A_i$. Decrease occurs via larval maturation to the adult stage, $m_i h_i L_i$, and natural larval death, $d_i L_i$. An adult female that visits a plant oviposits e_i eggs, but can increase her egg production by a fraction ϵ if she has nectared-at *Agave palmeri*, a plant on which she will not oviposit. This value is roughly parameterized from empirical data. In the model, oviposition follows a linear function of insect visits, which reflects observations. However, in the supplemental, e_i is defined as the oviposition efficiency, which is the relationship between oviposition rate and floral visitation rate, v_i . This reader is still a bit confused about the meaning of e_i . Also, the authors do not provide empirical data to support a linear relationship between insect visits and plant number. Other literature shows that pollinator visits increase with plant density in complex ways related to the sizes of individual and group floral displays. Other research also finds that herbivores show density-dependent attraction to host plants, although I'm not sure if such functions are usually linear or saturating. Are such data available for any *Datura* species? The number of larvae maturing to adults is similarly assumed to increase linearly with the amount of feeding that larvae do. Given that Eq 3a gives plants a built in self-limitation, how realistic are these linear functions? Parameter values for larval growth are determined empirically in the supplemental. I wonder, though, whether it is reasonable to assume that larval maturation efficiency is the same on the two plant species, even though their mortality rates differ? What factors cause larval mortality and how do they differ from factors influencing maturation rate?*

Response:

The Reviewer makes several good points. We define oviposition efficiency in the Methods section directly as the number of eggs that female *M. sexta* lay per floral visit. In lines (364-365) we reference empirical support for this linear function, which we explain in Supplementary Data 1 (lines 766-773). The Reviewer is correct that we assume that the number of larvae maturing to adults increases linearly with herbivory rate, which we do for tractability and due to data limitations. When developing the model, we explored incorporating insect self-limitation into the model; however, it greatly complicates the model (perhaps rendering the model intractable) and is extremely difficult to quantify empirically, so we omit insect self-limitation for tractability. Finally, motivated by the Reviewer's comments, we have changed how we estimate maturation efficiency and larval mortality in the model (lines 534-540).

Equation 3c models adult insect population growth as larval maturation to adult insects ($m_i h_i L_i$), summed over all host plant species, minus natural adult insect death as d_A . The summation in this equation accounts for the fact that insects can oviposit and mature larvae on more than one host plant species. Again, the number of larvae maturing to adults is assumed to increase linearly with the amount of feeding that larvae do.

Important potential complexities: The a constant d_A across host plant species assumes that all adult insects are equally fit, regardless of which host plant each fed on as a larva. The model also assumes that the rate at which adult insects visit host plant species is not affected by which host plant they fed on as a larva. How realistic are these assumptions in this system? Is it assumed that since adult life span is so much shorter than larval life span that these differences are trivial? It is also not clear to me how A can be scaled to give the adults recruited per plant of species i when A is the sum of adults recruited from all three potential larval host species.

Response:

These are good points. We assume that adult mortality is the same across host plant species and does not affect visitation/oviposition preference purely for simplicity and due to data limitations. We do not know how realistic these assumptions are empirically, but if they were relaxed in the model, we predict that it would only increase *M. sexta*'s preference for *D. wrightii* (assuming that *D. wrightii* is a higher-quality host plant, and this difference translates into more fit adults). Finally, adult density is no longer scaled per plant of species *i* in the revised manuscript to avoid the issue identified by the Reviewer.

To pick at nits, the quadratic form in Equation S1.3 should be defined as eigenvalues 2 and 3 (e.g., λ_2 and λ_3) rather than λ_2 , alone. There might be more than one convention for this, but I find that expressing them separately is less confusing. I can't speak to the solution of these two eigenvalues, but assuming that they are correct, then S1.4 follows and S1.5 (Equation 1 in the main text) is correct. I could not follow the model past equation S1.5, which should be evaluated by a more proficient reviewer.

I'd like the authors to provide a little more biological intuition to explain the structure of equation S1.6. This result is important, as it produces Equation 2 in the main text and also explains why the herbivore coexistence spaces (Figure 2a, 2c) do not overlap the mutualistic pollinator coexistence spaces (Figure 2b, 2d). This phenomenon yields a key result, which is that the evolution of plant traits that allow the insect to nectar beneficially (which I interpret to mean the transition to night-blooming flowers, although the authors might be referring here to the evolution of pollinator attractive scent – I'm not sure which and the distinction is important) does not inherently lead to a mutualism. Without understanding the origin of equation S1.6, I feel unable to verify this result.

Response:

In response to this comment and comments from other Reviewers, we have moved the mathematics in Supplementary Equations 1 and 2 of the original manuscript into more clearly delineated subsections of the revised Methods (lines 382-412). The conditions for mutualism (Equation S1.6 in the original manuscript) is derived by setting the equilibrium plant density, P_i^* , greater than one (i.e., the threshold above which the interaction is mutualistic for the plant) and rearranging. We show the algebra of this derivation in greater detail in the revised manuscript (lines 400-404).

Reviewer #4

Despite that the mutualistic interactions are essential in most ecosystems, the current stage of knowledge about the evolutionary processes behind these interactions is far to be sufficient. So I highly welcome any well-founded hypothesis explaining the origin and evolutionary stability of mutualistic interactions. The paper by Johnson et al. presents such a hypothesis. They not only describe verbally the basic evolutionary process on how antagonism can evolve to mutualism, but they present a formal model and a simple sample community which fits even quantitatively to the model.

I think the Co-opted Antagonist Hypothesis is biologically justified and it is the new invention of the authors. Actually the authors suggest a two-step eco-evolutionary process where first, selection on a focal (plant) species favours the co-option of an antagonist (insect) to perform a potentially beneficial function. The second step is that the focal species evolves a series of phenotypic traits driving the interaction from antagonism to mutualism. The used mathematical tools and model assumptions are adequate, the analysis is correct, although at some points it is too succinct.

Despite my general positive opinion about the manuscript, there are some points which should be clarified before the publication. I listed my comments and suggestions below:

1. Lines 40-42. I suspect that insects are ideal for this two consumer integration since the different food intake in larvae and adult state. How can we explain the evolution of bat pollination? (The discussion mentions the bacterial-phage interaction as an alternative example.)

4.1 Response:

This is a great point, which we think calls for a more general model. We derived a more general model in which the focal partner lacks complex stage-structured of *M. sexta* (see Response 1.10). The more general model is described in detail in the revised Methods (lines 602-627) and we briefly discuss its implications in the Discussion of the revised manuscript (Lines 328-332).

2. Lines 45-48. It would be nice to have information about the specialization of the central European Lepidopteras mentioned above. Is there any information for that?

4.2 Response:

This is an interesting question. Some information on the specialization of the Lepidoptera in this system is given in an appendix of their paper (Altermatt & Pearce 2011). While a full analysis of this data is beyond the scope of our paper, our *M. sexta* system appears to be far more specialized than most Lepidoptera in that system, a point that we make more generally in lines (70-73) of the revised manuscript.

3. Line 93. Some explanation for the relation (1) is needed, even if the details are in the supplementary. I suggest to introduce the used variables here, reader should understand the condition without reading the methods part, at least on intuitive level.

4.3 Response:

This is a great point. We have simplified Equation 1 into two more intuitive quantities, insect lifetime fecundity and larval success, which we now discuss in the explanation of Equation 1 in the revised manuscript (lines 98-99).

4. Eq. (2) is just the reverse of eq (1). Reader may ask again without reading the method and supplementary material why. Some more intuitive explanation may help. I think explanation below the equation is not enough.

4.4 Response:

We agree and have made the edits discussed in Response 4.3 as well as discuss how Equation 2 captures a conflict of interest between plants and the insect (lines 109-110) and its consequences for evolutionary transitions to mutualism (lines 111-117) in the revised manuscript.

5. Lines 149-151. What is the case if multiple plants and insects are coevolved. The model assumes only the evolution of plant species. I think this is the most critical assumption, which definitely should be explained and analysed in the revised version. Discussion mention it as a future step, but the critical question is whether the main conclusions remain valid after including evolution of pollinator.

4.5 Response:

This is a great point. In response to this comment and similar comments by other Reviewers, we have developed a new coevolutionary model (see Response 2.0), which we discussed in detail in both the Methods (lines 414-524) and throughout the main text of the revised manuscript.

6. Lines 326-327. Is there any indirect or direct argumentation for this assumption? The authors assume weak competition, but there is no competition between the plant species in the model. Are we sure that the results are robust against this assumption?

4.6 Response:

The assumption is that the plants do not compete interspecifically; intraspecific competition is captured in the model via plant self-limitation. When developing the model, we explored the effects of interspecific competition; however, it greatly complicates the model (perhaps rendering the coevolutionary model intractable). Competition coefficients are also difficult to quantify empirically, and no such data exists for the *M. sexta* system. Thus, we make the simplifying assumption that interspecific competition is weak for tractability. This assumption also allows us to isolate the effects of the focal insect on plant ecological dynamics from those of exploitative competition (note, apparent competition is possible in our model).

7. Line 248 mortality, Line 338. It should be 3bc.

4.7 Response:

An amusing typo, which we have now corrected.

8. Line 362 It is rational to assume that increasing visiting rate because of pollination increases the oviposition as well, but some explanation is suggested. One more sentence is enough.

4.8 Response:

We have changed the nature of the trade-offs in the new coevolutionary model, so that this is no longer an issue in the revised manuscript.

9. Line 363. The trade-off between r and h is linear. Does the result depend on this assumption? Concave and convex trade-offs can lead to different evolutionary routes.

4.9 Response:

We have changed the trade-offs in the new coevolutionary model in the revised manuscript. In the new model, plant trade-offs (which now involve mutant plant competitive ability) are still linear functions because concave and convex trade-off shapes cause selection on plant trait x^{mut} (for any general trait x) to either go to infinity or zero depending on the values of the evolutionary coefficients.

10. Line 583. I suggest to denote the eigenvalues with $\lambda_{2,3}$, these are the additional two eigenvalues.

4.10 Response:

This has been corrected.

11. Lines 612. Another crucial thing that h' should be at least 30% greater than $r'h$ for the coexistence of the plant and insect. I suggest to mention this fact in the discussion, and to place in a biological context, if it is possible.

4.11 Response:

This result no longer holds in the revised manuscript due to changes in the new coevolutionary model.

12. The mathematical analysis uses the standard adaptive dynamics. Please refer the adequate initial papers of this method.

4.12 Response:

We have added these references in the revised version of the text.

I note that I am a theoretical biologist, being not familiar in the experimental studies so I cannot evaluate the merits and correctness of this part of the paper.

REVIEWER COMMENTS

Reviewer #1 (Remarks to the Author):

In the revised manuscript, the authors have addressed most of my initial concerns. The model formulation and trade-off functions are now very clear, and also the result section has improved considerably. I'm especially happy to see the addition of fig 5 and the dynamics of the traits and species in the other figures.

Although the ms is now much easier to follow, as a reader I needed to work quite hard to understand what is going on. I therefore think the authors need to improve their result section, to ensure that people can understand what is going on in the model. Below, I have described in more detail what I thought is unclear in the result section. All in all, I really like the paper and I hope the authors will be able to address these points.

1) The eco-evo dynamics start from a situation where $r_i = 0$, while the other traits are in their ESS (white dots). It would be helpful to state this clearly in the result section, because this assumption is now hidden in the model description.

2) The authors initially state that mutualism is not possible because a viable phenotype of the ancestral insects remains in the antagonistic regions (L133-135). This is explained by the finding that eq 2 is the reverse of eq 1. It is not very clear what happens exactly that causes eq 1 to change. Is it that the plant densities are above 1? A short explanation would be useful.

3) The authors describe the COA hypothesis as a two-way process. I find this a bit confusing given that all traits change together, and it is therefore a continuous process. The new figures give a good intuitive understanding of what is going on. However, if I only read the text, I almost get the impression that initially r_i evolves, and subsequently the other traits, which is not the case. It would help me if the authors describe more clearly that in step 1 r_i evolves, together with the other traits but that initially there is no net mutualism (although plant densities increase). Due to the change in traits, over evolutionary time (step 2) the coESS evolves to support a mutualistic interaction. Maybe it would help to create a video for the supplementary info to show how the green area arises together with the trajectory of the coESS over evolutionary time?

4) Related to my previous point, in the discussion (L235-L46) the results are not so clearly described, again the suggestion is given that first r_i evolves and that only later the other traits follow. Also, L237-238 'to perform a potentially beneficial function', I think this sentence is slightly misleading. It is already beneficial, since it increases plant densities. If it would be disadvantageous, it wouldn't evolve in the first place.

5) I don't understand figure 4A. The ancestral co-ESS seems to be in the region of extinction. How is this possible and what does it imply?

Minor points/typos

L109 'Coevolution of all three traits', I think it should be 5 traits.

L 523 'We constrain xi to be positive', should be non-negative since it can have values of 0.

Sup table 4: the line numbers are currently in the table instead of next to it.

Reviewer #2 (Remarks to the Author):

Dear authors and editors,

I was the original reviewer 2. I am now even more convinced that this manuscript will advance the field substantially (see my previous review for why).

The authors have made exceptional improvements, and have demonstrated thoroughness in responding to the reviews. I am satisfied with the response to every single one of my comments (and the comments in response to other reviewers). I cannot praise the changes enough (evolution of all three traits, plus coevolution of insects, plus a general model – this is a lot of work!). I found the new version of the manuscript an exciting read.

As there is a large amount of new content however, I have one new comment, and a few minor ones listed below.

First, the only important comment, is that a quick biological or intuitive justification for some new results in 4-5 places would be helpful. Specifically...

Line 115: I'm a plant biologist, so I intuit where costs of producing attractants, defenses come from, but I'm not an insect biologist. A 0 cost of being attracted in insects totally changes results (Fig 5) making the assumption of costs of being attracted important. Lines 410-411 start this, but aren't quite clear.

Line 137: Why does the zone of persistence (Fig 2) expand for the insect (relevant also to line 195)? & relatedly on line 154 – why do h and v decrease, and insect density increases, when r increases?– are there reduced costs of sensitivity and tolerance traits as plant evolves less defense chemicals & insect evolution responds? It might help to see a supplementary figure on dynamics of xiv, yiv, xih, yih, or expansion of the persistence zone, although I support the choice of only including the realized visitation and herbivory rates in main text figures as this is much less complicated.

Line 188: The reason for this becomes clear towards the end of the methods, but a quick intuition for why here, if possible, would be ideal.

Typos, minor compliments, & formatting suggestions-

Figure 1: Text in the key is a bit small. Maybe hmax should be defined? or maybe note that axis is reciprocal of herbivory in legend? We otherwise don't see hmax until later. Notably, a & b are hmax-h,

but c and d present h, it might be more intuitive if a and b had a flipped x (or c&d presented also hmax-h) – likewise for Figures 3&4.

Line 178: Cool insight.

Figure 2: Why is the trajectory of insect density different in c and d? I thought only panels a and b were where one species varies and the other is held at the coESS? I suspect this is a legend issue (e.g. insect densities partitioned by plant host?)

Line 308 & thereabouts: I like this paragraph. Useful insight.

Line 331 & paragraph following: Also a great addition.

**Some of the new math escaped my understanding in detail, so (editors & authors) please take the following comments with a grain of salt.

Line 390: does this affect equation 3b too?

Lines 427-430: I didn't quite understand where this section came from, but I think I'm just unfamiliar with these mathematical methods.

Line 433: The simulations are mentioned a few times, but readers aren't pointed towards where to find them (code, I assume?)

Lines 516-527: it'd be nice to have a biological interpretation of why x_{IV} should be constrained positive in ancestor, but allowed to go negative above....but, as I think this model is just meant to contrast with the full model, not sure this is really necessary

Line 525: "here" but if in supplementary code, good to mention where reader can find

Table S3: Why are empirically estimated visitation rates for Dw and Dd only included for comparison with coESS in the two-datura species scenario only?

Reviewer #3 (Remarks to the Author):

Lines 97 – 99, This sentence does not clear state what organisms occur in each of the four scenarios, or even what are the four scenarios. Currently, I count: 1. insect + D. w., 2. insect + D. d., 3. D. w. + D. d., 4. D. w. + alt host, 5. D. d. + alt host, 6. D. w. + alt nectar source, 7. D. w. + alt nectar source, where none of the latter five scenarios involves an insect.

Line 102 – 103, not clear what is meant by "two plant-insect responses."

Line 109, it would be best to avoid "natural selection acting on." Natural selection is the differential fitness of organisms; it is not an entity that acts on anything. This issue comes up several times in the manuscript.

Lines 111 – 112, The sentence starting with "Evolution of pollination" is difficult to parse.

Lines 113 - 115, Competitive ability is a plant trait. Also, "in the model" at the end of the sentence is redundant with the beginning of the sentence.

Line 162, Why is "COA step 1" in parentheses?

Lines 163 – 165, This phrasing is a big improvement over the previous draft, as it clarifies the distinction between the interaction transition boundary and the interaction breakdown boundary.

Lines 177 – 178 and figures 2 - 5, It would be nice to see all the predictions on the same figure, although, as was probably also true for the authors, I can't imagine a way to do that without making a figure that is too dense.

Line 198, Why are the blue crossbars included only in Figure 3, but not the other figures?

Line 208, it would be nice occasionally include "larval" with "host plant" to make clear what "host plant" means, as one could also think of nectar production as a "host" trait.

Line 212, It is not clear what is meant by "around"

Lines 213 – 214, This sentence should come before the previous sentence, as it is a more general statement, whereas the previous sentence is one of several examples, the others of which follow in this paragraph.

Lines 215 – 216, This sentence could be clearer with active voice. For example, substitute in "Conversely, the presence of the alternative nectar source narrows the parameter space available for mutualism because ..."

Line 218, substitute "increases" for "leads to increased"

Lines 221 – 223 This sentence is difficult to parse.

Lines 245 - 246, Natural selection doesn't act "on the second step." This sentence needs restating.

Lines 286 – 288, This sentence doesn't make sense. Some solutions could be, "subsidizing the cost of herbivory experienced by the focal plant" OR "shifting the cost of herbivory from the focal plant".

Line 289, Maybe substitute "its consequent" for "buoying"

Lines 291, substitute "resource axis along" for "resource axis in"

Line 304, Evolution is not an entity that can modify traits to do things. Maybe substitute "traits whose evolution can co-opt" rather than "traits that evolution can modify to co-opt"

Lines 305 – 306, This sentence is really ugly. Again, natural selection is differential fitness; it does not "act."

Lines 307 – 308, Maybe substitute “, which could coevolve toward greater mutualistic benefit at lower trophic cost” instead of “, for coevolution to drive”

Line 309, The insect is the antagonist, right? Because at the start of this thread (First, focal species ... co-opt the antagonis.” So, the plant is the focal species. Could the insect be the focal species? Perhaps I’ve gotten balled up in the language.

Lines 309 – 311, Maybe substitute, “Third, if fitness costs were too extreme, the antagonism would either persist or disappear via loss of the antagonist.”

Lines 311 – 312, Maybe substitute, “Finally, the focal species must experience stronger selection from the antagonist than it does from other species in the community.” Or, perhaps I’ve misinterpreted the meaning of this somewhat vague final statement.

Line 319, Here, “focal” is used to describe the insect, which adds to confusion created in the previous paragraph, where the role of the focal species is not identified.

Lines 321 – 330, This paragraph reads too much like a brain dump. It needs better organization. For example, maybe organize the patterns of selection around the patterns of trade-offs or lack thereof.

Lines 335 – 339, This is a really nice point and well-stated.

Lines 349 – 353, I have the feeling that this general model gets kind of tacked on at the end. Could it be better integrated into the introduction?

Methods section, It’s really helpful to have the entire model development in one place, including the model scaling (lines 392 – 401).

Line 575, “insect trade-offs” is too vague, please describe these as trait trade-offs and maybe even mention which are the insect traits that are negatively correlated.

Line 599, This is an important point.

Lines 608 – 610, “provided that” is kind of a vague transition in this sentence.

Lines 610 – 611, This is a much cleaner way of stating a result.

Lines 623 – 648, The general model is a really nice addition and should be better highlighted or presaged in the introduction.

Supplementary Table 1, I was unable to distinguish bold faced from normal text parameters, although I could see that the parameter values were bold-faced.

Supplementary Table 3, A bit more line spacing would make the trait initial and final trait values, with both super- and sub-scripts, much more readable.

Supplementary table 4 has a problem with the line numbering.

Some of my comments reflect my opinions about writing style. Therefore, I don't expect the authors to take all my suggestions. In some cases, though, I've highlighted real problems with clarity and I hope that the authors will solve those problems.

Ellen L. Simms

Reviewer #4 (Remarks to the Author):

Below I attached my reactions to the responses of the authors to my previous report.

Numbers denote the same order as they were posed in the previous report. Missing points means that I completely accepted the answer and the modification in the text.

1. I am happy with the new generalized model, but I think it still contains those terms (and biological factors) which are still basically important for the COA hypotheses described by the authors. Namely that there is a term and thus a biological process which decreases the reproductive success of the host and a separate term and thus biological process which potentially increases the reproductive success of the host. These terms and are typically present in herbivorous insects but not so obvious in case of for example microbial mutualism or in case of bat pollination. I think the new inserted text still doesn't refer to this point.

The other more technical problem connected with the general model as it is introduced. First, the authors assume that adult dynamics is much faster than any other dynamics which leads to a simpler dynamical system. This is fine and adequate in case of insects. But this step seems to be not a generalisation per se. Then, based on this assumption they suggest that this model can be a generalization of the more specific model studied in the paper. The comparison the parameters of the specific model with the general one is rather disturbing for the reader, since for example the indices don't fit to each other. I think the reader can be lost easily why a simple index i becomes indices i, j , etc. So I suggest to introduce the general model first, and mention that the time separation leads to the studied model into this general model (where there is no two life stage of the symbiont), and to mention that the general parameters which uses two indices (and have general meaning) is simplified to parameters with one index in the specific model used by the paper.

6. I understand that taking plant competition into consideration makes the analysis intractable or at least much more complicated. But my question focused on the robustness of the result, and I think this point is still open. One cannot exclude that even weak interspecific competition can modify the main conclusion. I would suggest to add at least some intuitive argument why this is not the case, or at least

to mention in the discussion this weak point of the study.

9. The fact that nonlinear trade-offs modify the results significantly is again a warning signal for the problem of robustness of the model. Again I suggest at least to analyse this problem in the discussion. I would be very happy if the authors could convince me that the observation that μ goes to infinity or zero at any non-linear trade-offs doesn't mean that the model and thus the results are non-robust.

Comment: The referred lines in the response were almost always incorrect which made the reading more time consuming.

Response to Reviewers

Below we respond to each comment by the four Referees. We fully reproduce all comments in black italic text, paired with our responses in blue text. References to specific lines in the main text are highlighted.

Reviewer #1

In the revised manuscript, the authors have addressed most of my initial concerns. The model formulation and trade-off functions are now very clear, and also the result section has improved considerably. I'm especially happy to see the addition of fig 5 and the dynamics of the traits and species in the other figures.

Although the ms is now much easier to follow, as a reader I needed to work quite hard to understand what is going on. I therefore think the authors need to improve their result section, to ensure that people can understand what is going on in the model. Below, I have described in more detail what I thought is unclear in the result section. All in all, I really like the paper and I hope the authors will be able to address these points.

1) The eco-evo dynamics start from a situation where $r_i = 0$, while the other traits are in their ESS (white dots). It would be helpful to state this clearly in the result section, because this assumption is now hidden in the model description.

We have included this condition before Equation 1. We have also changed the notation for pollination benefits from r_i to b_i to avoid confusion as r_i commonly refers to an intrinsic per capita growth rate.

2) The authors initially state that mutualism is not possible because a viable phenotype of the ancestral insects remains in the antagonistic regions (L133-135). This is explained by the finding that eq 2 is the reverse of eq 1. It is not very clear what happens exactly that causes eq 1 to change. Is it that the plant densities are above 1? A short explanation would be useful.

Motivated by this comment, we have included an approximation of the interaction breakdown boundary (Eq. 2b) under the assumption that the saturation constant for pollination, H , is relatively low (Methods). Evolution of traits that allow the plant to co-opt the antagonist as a pollinator and subsequently increase the pollination benefits (b_i) creates the region in which mutualism occurs by buoying plant density and relaxing the interaction breakdown boundary. That is, through its effects on plant per capita population growth rate, the evolution of pollination benefits ($b_i > 0$) allows the insect to persist under lower visitation rate (v_i) and herbivory rate (h_i) than it could as a pure antagonist. Within this new parameter region, the interaction is net mutualistic for the plant (delineated by the interaction transition boundary, Eq. 2a, and the new interaction breakdown boundary, Eq. 2b) because the plant is able to attain densities greater than one. We have included a brief explanation in the Results (lines 109-111).

3) The authors describe the COA hypothesis as a two-way process. I find this a bit confusing given that all traits change together, and it is therefore a continuous process. The new figures give a good intuitive understanding of what is going on. However, if I only read the text, I almost get the impression that initially r_i evolves, and subsequently the other traits, which is not the case. It would help me if the authors describe more clearly that in step 1 r_i evolves, together with the other traits but that initially there is no net mutualism (although plant densities increase). Due to the change in traits, over evolutionary time (step 2) the coESS evolves to support a mutualistic interaction. Maybe it would help to create a video for the supplementary info to show how the green area arises together with the trajectory of the coESS over evolutionary time?

In the revised manuscript, we have reduced the emphasis on the two steps of the COA to avoid confusion that b_i (formerly r_i) evolves prior to the other responses, which is not the case. We highlight via Equations 2a and 2b that b_i affects the interaction breakdown boundary directly (allowing the insect to persist over a

larger parameter range), but that the transition to mutualism requires coevolution of all three responses. Following the Reviewer's excellent suggestion, we have created Supplementary Video 1 for Figure 2a.

4) *Related to my previous point, in the discussion (L235-L46) the results are not so clearly described, again the suggestion is given that first r_i evolves and that only later the other traits follow. Also, L237-238 'to perform a potentially beneficial function', I think this sentence is slightly misleading. It is already beneficial, since it increases plant densities. If it would be disadvantageous, it wouldn't evolve in the first place.*

We have revised the Discussion to highlight that it is the coevolution of all three responses – pollination benefits, attraction, and defense, that drives the evolutionary transition from antagonism to mutualism (lines 218-221). We avoid the phrase “to perform a potentially beneficial function”.

5) *I don't understand figure 4A. The ancestral co-ESS seems to be in the region of extinction. How is this possible and what does it imply?*

We failed to plot the interaction transition boundaries for the ancestral interaction. We thank the Reviewer for identifying this omission. In the revised manuscript, we have plotted the interaction breakdown boundaries for the ancestral interaction (dashed gray lines) in all Figures. As can be seen in the updated Figure 4, the ancestral coESS is well within the region of persistence. Unlike in the other scenarios, coevolution actually expands the green mutualistic region over the ancestral coESS in the presence of an alternative larval host plant, causing the confusion identified by the Reviewer.

Minor points/typos

L109 'Coevolution of all three traits', I think it should be 5 traits.

This line has been removed from the revised manuscript.

L 523 'We constrain x_i to be positive', should be non-negative since it can have values of 0.

This issue has been corrected.

Sup table 4: the line numbers are currently in the table instead of next to it.

This issue has been corrected.

Reviewer #2:

I was the original reviewer 2. I am now even more convinced that this manuscript will advance the field substantially (see my previous review for why).

The authors have made exceptional improvements, and have demonstrated thoroughness in responding to the reviews. I am satisfied with the response to every single one of my comments (and the comments in response to other reviewers). I cannot praise the changes enough (evolution of all three traits, plus coevolution of insects, plus a general model – this is a lot of work!). I found the new version of the manuscript an exciting read.

As there is a large amount of new content, I have one new comment, and a few minor ones listed below. First, the only important comment, is that a quick biological or intuitive justification for some new results in 4-5 places would be helpful. Specifically...

Line 115: I'm a plant biologist, so I intuit where costs of producing attractants, defenses come from, but I'm not an insect biologist. A 0 cost of being attracted in insects totally changes results (Fig 5) making the assumption of costs of being attracted important. Lines 410-411 start this, but aren't quite clear.

We briefly discuss the plant and insect trade-offs in greater detail in the revised manuscript following this helpful suggestion (lines 90-92). Insect traits trade off with oviposition so that mutant insects with high sensitivity to olfactory cues or high tolerance to plant defenses, for example, produce fewer eggs.

Line 137: Why does the zone of persistence (Fig 2) expand for the insect (relevant also to line 195)? & relatedly on line 154 – why do h and v decrease, and insect density increases, when r increases?– are there reduced costs of sensitivity and tolerance traits as plant evolves less defense chemicals & insect evolution responds? It might help to see a supplementary figure on dynamics of x_{iv} , y_{iv} , x_{ih} , y_{ih} , or expansion of the persistence zone, although I support the choice of only including the realized visitation and herbivory rates in main text figures as this is much less complicated.

The region in which the insect persists increases as pollination benefits, b_i (formerly r_i), increase because increasing plant density relaxes the interaction breakdown boundary, a point we highlight in more detail in the revised manuscript (lines 106-118). As plant density increases, both v_i and h_i decrease because the evolving mutualism increases insect fitness even as v_i and h_i decrease, allowing the insect to evolve lower sensitivity and tolerance traits in favor of greater oviposition. Following the Reviewer's suggestion, we have also added Supplementary Figure 1, which plots coevolution of the trait underlying Figure 2e,f.

Line 188: The reason for this becomes clear towards the end of the methods, but a quick intuition for why here, if possible, would be ideal.

We briefly discuss the different evolutionary strategies by the *Datura* species (lines 167-170).

Typos, minor compliments, & formatting suggestions-

Figure 1: Text in the key is a bit small. Maybe h_{max} should be defined? or maybe note that axis is reciprocal of herbivory in legend? We otherwise don't see h_{max} until later. Notably, a & b are $h_{max}-h$, but c and d present h , it might be more intuitive if a and b had a flipped x (or c & d presented also $h_{max}-h$) – likewise for Figures 3&4.

We have increased the text size in the keys and included h_{max} in the legends of all Figures.

Line 178: Cool insight.

Thank you.

Figure 2: Why is the trajectory of insect density different in c and d ? I thought only panels a and b were where one species varies and the other is held at the coESS? I suspect this is a legend issue (e.g. insect densities partitioned by plant host?)

Figure 2 is for each of the one-plant species communities, so insect density follows different trajectories because they are involved in different pairwise interactions: the left panels show the pairwise *D. wrightii* – *M. sexta* interaction and the right panels show the pairwise *D. discolor* – *M. sexta* interaction. To aid with interpretation, we have plotted larval densities per host plant in all Figures in the revised manuscript.

Line 308 & thereabouts: I like this paragraph. Useful insight.

Line 331 & paragraph following: Also a great addition.

We thank the Reviewer for the compliments.

***Some of the new math escaped my understanding in detail, so (editors & authors) please take the following comments with a grain of salt.*

Line 390: does this affect equation 3b too?

Pupae survival, ρ_i , does not affect larval dynamics (Eq. 3b) because pupae have already left the larval stage of the model (via the second term in Eq. 3b) and thus their survival only impacts adult dynamics.

Lines 427-430: I didn't quite understand where this section came from, but I think I'm just unfamiliar with these mathematical methods.

The interaction breakdown boundary in a one-plant species community was determined by the conditions for the equilibrium to be feasible (real), indicating that the insect can persist. We discuss this section in greater detail as we introduce the interaction breakdown boundary (Eq. 2b) in the revised manuscript.

Line 433: The simulations are mentioned a few times, but readers aren't pointed towards where to find them (code, I assume?)

We thank the Reviewer for catching this omission, we reference our online codes in the revised version.

Lines 516-527: it'd be nice to have a biological interpretation of why x_i^V should be constrained positive in ancestor, but allowed to go negative above....but, as I think this model is just meant to contrast with the full model, not sure this is really necessary

We constrain x_i^V to be non-negative in the ancestral interaction because otherwise x_i^V always goes to $-\infty$, the visitation rate goes to zero, and the plant always purges the insect. Mathematically, this is problematic because there is no stable coESS for x_i^V . We discuss this issue in the revised manuscript (line 565-567).

Line 525: "here" but if in supplementary code, good to mention where reader can find

We have referenced our online codes in the revised version.

Table S3: Why are empirically estimated visitation rates for Dw and Dd only included for comparison with coESS in the two-datura species scenario only?

We only have estimates of the visitation rates for *D. wrightii* and *D. discolor* in the two-*Datura* species scenarios because we do not have sufficient data for the scenarios in which *M. sexta* visited only *D. wrightii* or only *D. discolor*. We have clarified this point in the revised manuscript (lines 146-147).

Reviewer #3:

Lines 97 – 99, This sentence does not clear state what organisms occur in each of the four scenarios, or even what are the four scenarios. Currently, I count: 1. insect + D. w., 2. insect + D. d., 3. D. w. + D. d., 4. D. w. + alt host, 5. D. d. + alt host, 6. D. w. + alt nectar source, 7. D. w. + alt nectar source, where none of the latter five scenarios involves an insect.

We revised the sentence to clearly state what organisms occur in each of the four scenarios (lines 78-81).

Line 102 – 103, not clear what is meant by “two plant-insect responses.”

We have revised the sentence to specify that we are modeling plant and insect traits driving coevolution of attraction and defense (lines 82-84).

Line 109, it would be best to avoid “natural selection acting on.” Natural selection is the differential fitness of organisms; it is not an entity that acts on anything. This issue comes up several times in the manuscript.

We agree and have revised this, and other, lines in the manuscript accordingly.

Lines 111 – 112, The sentence starting with “Evolution of pollination” is difficult to parse.

We have removed this line in the revised manuscript.

Lines 113 - 115, Competitive ability is a plant trait. Also, “in the model” at the end of the sentence is redundant with the beginning of the sentence.

We have revised this line in the revised manuscript.

Line 162, Why is “COA step 1” in parentheses?

We have removed this phrase in the revised manuscript.

Lines 163 – 165, This phrasing is a big improvement over the previous draft, as it clarifies the distinction between the interaction transition boundary and the interaction breakdown boundary.

We thank the Reviewer for the compliment.

Lines 177 – 178 and Figures 2 - 5, It would be nice to see all the predictions on the same figure, although, as was probably also true for the authors, I can’t imagine a way to do that without making a figure that is too dense.

We considered how to condense the number of Figures as well, but concluded that there was far too much information being presented in each Figure that could be reasonably presented in one Figure.

Line 198, Why are the blue crossbars included only in Figure 3, but not the other figures?

We only have estimates of the visitation rates for *D. wrightii* and *D. discolor* in the two-*Datura* species scenarios because we do not have sufficient empirical data for the scenarios in which *M. sexta* visited only *D. wrightii* or only *D. discolor*. We clarify this point in the revised manuscript (lines 146-147).

Line 208, it would be nice occasionally include “larval” with “host plant” to make clear what “host plant” means, as one could also think of nectar production as a “host” trait.

Good point, we have done so throughout the manuscript.

Line 212, It is not clear what is meant by “around”

We have removed this sentence from the revised manuscript.

Lines 213 – 214, This sentence should come before the previous sentence, as it is a more general statement, whereas the previous sentence is one of several examples, the others of which follow in this paragraph.

We have reordered these sentences in the revised manuscript.

Lines 215 – 216, This sentence could be clearer with active voice. For example, substitute in “Conversely, the presence of the alternative nectar source narrows the parameter space available for mutualism because ...”

We have made the suggested changes.

Line 218, substitute “increases” for “leads to increased”

We have made the suggested edit.

Lines 221 – 223 This sentence is difficult to parse.

We have revised this sentence for clarity.

Lines 245 - 246, Natural selection doesn't act “on the second step.” This sentence needs restating.

True. We have revised this sentence.

Lines 286 – 288, This sentence doesn't make sense. Some solutions could be, “subsidizing the cost of herbivory experienced by the focal plant” OR “shifting the cost of herbivory from the focal plant”.

We have made the suggested edit.

Line 289, Maybe substitute “its consequent” for “buoying”

We have made the suggested change.

Lines 291, substitute “resource axis along” for “resource axis in”

We have made the suggested change.

Line 304, Evolution is not an entity that can modify traits to do things. Maybe substitute “traits whose evolution can co-opt” rather than “traits that evolution can modify to co-opt”

Good point, we have made the suggested substitution.

Lines 305 – 306, This sentence is really ugly. Again, natural selection is differential fitness; it does not “act.”

Noted, we have revised this sentence to improve clarity.

Lines 307 – 308, Maybe substitute “, which could coevolve toward greater mutualistic benefit at lower trophic cost” instead of “, for coevolution to drive”

We thank the Reviewer for the suggestion, which we have made in the revised manuscript.

Line 309, The insect is the antagonist, right? Because at the start of this thread (First, focal species ... co-opt the antagonist.” So, the plant is the focal species. Could the insect be the focal species? Perhaps I’ve gotten balled up in the language.

We have revised the paragraph to avoid confusion about which species is the ‘focal’ (lines 277-286).

Lines 309 – 311, Maybe substitute, “Third, if fitness costs were too extreme, the antagonism would either persist or disappear via loss of the antagonist.”

We thank the Reviewer for the suggestion, which we have incorporated into the revised manuscript.

Lines 311 – 312, Maybe substitute, “Finally, the focal species must experience stronger selection from the antagonist than it does from other species in the community.” Or, perhaps I’ve misinterpreted the meaning of this somewhat vague final statement.

Correct, we have made the suggested edits to improve the clarity of the final statement. We thank the Reviewer for all of the helpful comments on this paragraph of the Discussion.

Line 319, Here, “focal” is used to describe the insect, which adds to confusion created in the previous paragraph, where the role of the focal species is not identified.

We have revised the previous paragraph to avoid confusion about the ‘focal’ species (lines 277-286).

Lines 321 – 330, This paragraph reads too much like a brain dump. It needs better organization. For example, maybe organize the patterns of selection around the patterns of trade-offs or lack thereof.

We have reorganized this paragraph in response to the Reviewer’s suggestions as well as incorporated a brief discussion why competition could affect coevolution (lines 296-304) in response to Reviewer 4.

Lines 335 – 339, This is a really nice point and well-stated.

Thank you.

Lines 349 – 353, I have the feeling that this general model gets kind of tacked on at the end. Could it be better integrated into the introduction?

In response to this comment and similar comments from the other Reviewers, we have better integrated the general model into the Introduction of the revised manuscript.

Methods section, It’s really helpful to have the entire model development in one place, including the model scaling (lines 392 – 401).

We thank all the Reviewers for the helpful suggestion of combining the Methods sections in the last review, which we agree has improved the manuscript.

Line 575, “insect trade-offs” is too vague, please describe these as trait trade-offs and maybe even mention which are the insect traits that are negatively correlated.

We have revised this sentence in the revised manuscript.

Line 599, This is an important point.

True, the costs associated with plant and insect traits largely determine the coevolutionary outcomes given the way that we parameterized the coevolutionary model.

Lines 608 – 610, “provided that” is kind of a vague transition in this sentence.

Lines 610 – 611, This is a much cleaner way of stating a result.

We have edited this sentence in the revised manuscript in light of these comments.

Lines 623 – 648, The general model is a really nice addition and should be better highlighted or presaged in the introduction.

In response to this comment and similar comments from the other Reviewers, we have better integrated the general model into the Introduction and Results of the revised manuscript.

Supplementary Table 1, I was unable to distinguish bold faced from normal text parameters, although I could see that the parameter values were bold-faced.

We have slightly altered Supplementary Table 1 to highlight the parameters that evolve in the model.

Supplementary Table 3, A bit more line spacing would make the trait initial and final trait values, with both super- and sub-scripts, much more readable.

We have followed the Reviewer’s suggestion in Supplementary Table 3.

Supplementary table 4 has a problem with the line numbering.

We have removed Supplementary Table 4 from the revised manuscript.

Some of my comments reflect my opinions about writing style. Therefore, I don't expect the authors to take all my suggestions. In some cases, though, I've highlighted real problems with clarity and I hope that the authors will solve those problems.

We greatly appreciate all of the Reviewer’s comments about writing style.

Ellen L. Simms

Reviewer #4:

Below I attached my reactions to the responses of the authors to my previous report.

Numbers denote the same order as they were posed in the previous report. Missing points means that I completely accepted the answer and the modification in the text.

1. I am happy with the new generalized model, but I think it still contains those terms (and biological factors) which are still basically important for the COA hypotheses described by the authors. Namely that there is a term and thus a biological process which decreases the reproductive success of the host and a separate term and thus biological process which potentially increases the reproductive success of the host. These terms and are typically present in herbivorous insects but not so obvious in case of for example microbial mutualism or in case of bat pollination. I think the new inserted text still doesn't refer to this point.

The other more technical problem connected with the general model as it is introduced. First, the authors assume that adult dynamics is much faster than any other dynamics which leads to a simpler dynamical system. This is fine and adequate in case of insects. But this step seems to be not a generalisation per se. Then, based on this assumption they suggest that this model can be a generalization of the more specific model studied in the paper. The comparison the parameters of the specific model with the general one is rather disturbing for the reader, since for example the indices don't fit to each other. I think the reader can be lost easily why a simple index i becomes indices i, j , etc. So I suggest to introduce the general model first, and mention that the time separation leads to the studied model into this general model (where there is no two life stage of the symbiont), and to mention that the general parameters which uses two indices (and have general meaning) is simplified to parameters with one index in the specific model used by the paper.

Following the Reviewer's suggestion, we have reorganized the paper to present the general model first, from which we derive the pollination-herbivory model. In response to this comment and comments from the other Reviewers, we integrate the general model into the Introduction. We have made several changes to the general model in response to helpful comments from the Reviewer. Specifically, as we present the general model first, we no longer derive it via timescale separation (and so have removed Supplementary Table 4). These revisions avoid confusion about both the timescale separation and model indices as noted by the Reviewer, and allow the model to incorporate general functions of the mutualistic and antagonistic interactions. Now one could, in principle, apply the general framework to any interaction, from microbial mutualisms to bat pollination, by specifying these functions according to the biology of the study system.

To derive a more explicit eco-coevolutionary model, we simplify the general model (Eq. 1) to model a single host and partner species and specify the functions of the mutualistic and antagonistic interactions in terms of host and partner traits (Eq. 2). Here we note the assumption that the mutualistic and antagonistic interactions have separate and additive effects on host population growth. We specify functions for how host and partner traits affect species' population growth (Eq. 3) and present the eco-coevolutionary model for the generalized host-partner interaction (Eqs. 4-5). Lastly, we briefly discuss key results of the general model in terms of the shape of the trade-offs (see our Response 9 below). We thank the Reviewer for the helpful comments, which we think greatly improves the general model (see Methods lines 338-417).

6. I understand that taking plant competition into consideration makes the analysis intractable or at least much more complicated. But my question focused on the robustness of the result, and I think this point is still open. One cannot exclude that even weak interspecific competition can modify the main conclusion. I would suggest to add at least some intuitive argument why this is not the case, or at least to mention in the discussion this weak point of the study.

We have briefly discussed competition as a future direction in the Discussion (lines 302-304). We agree that even weak interspecific competition can modify coevolutionary dynamics. We make this assumption for tractability as well as due to data limitations (there are no data of which we are aware on interspecific competition between the *Datura* species). Our argument for why weak interspecific competition may not prevent the evolution of mutualism is as follows. For the ancestral community to be viable, interspecific competition must be weak or else the combined effects of exploitative *and* apparent competition (via the focal antagonist) would drive one of the ancestral plants to extinction. Interspecific competition would also lower the equilibrium densities of the ancestral plants (and therefore the thresholds for mutualism evolution). While interspecific competition would modify eco-coevolutionary dynamics, it may not prevent evolution of mutualism because mutant and resident plants would be equally sensitive to interspecific competition (recall that the trade-offs affect *intraspecific* competition). Mathematically, invasion fitness of mutant i given interspecific competition is:

$$IGR_{i,mut} = (1 - \alpha_{ij}P_j^* - q[x_i^{mut}, x_i]P_i^*) + f(\text{mutualism}) - f(\text{antagonism})$$

where $f(\text{mutualism})$ and $f(\text{antagonism})$ are the functions for the mutualistic and antagonistic interactions, respectively (Eq. 9a). The selection gradient for any general trait x_i is then: $\frac{\partial IGR_{i,mut}}{\partial x_i^{mut}} \Big|_{x_i^{mut}=x_i}$, which will

not depend upon the $-a_{ij}P_j^*$ term describing interspecific competition. Thus, the effects of interspecific competition on coevolutionary dynamics will be driven mainly by changes in the equilibrium density of the resident plants, P_i^* . As discussed previously, interspecific competition will reduce equilibrium plant density in both the ancestral and coevolved system. As long as interspecific competition does not harm plants in the coevolved system significantly more than in the ancestral system, interspecific competition is unlikely to prevent evolution of mutualism when it would otherwise arise. How mutualism, antagonism, and competition all jointly influence coevolutionary transitions from antagonism to mutualism, more generally, is an interesting question for future studies, but is beyond the scope of the current manuscript.

9. The fact that nonlinear trade-offs modify the results significantly is again a warning signal for the problem of robustness of the model. Again I suggest at least to analyse this problem in the discussion. I would be very happy if the authors could convince me that the observation that $\$x^{\{mut\}}\$$ goes to infinity or zero at any non-linear trade-offs doesn't mean that the model and thus the results are non-robust.

We agree that it is important to show that the model is robust to trade-off shape. In our previous response, nonlinear trade-offs drove evolutionary purging of the insect for that particular model parameterization, which led to imaginary values of the ecological equilibrium, causing coevolutionary dynamics to either go to infinity or zero. We have revised the codes for this manuscript (Johnson_et_al_Codes.zip) so that when evolutionary purging occurs, the model now reports it and stops the coevolutionary dynamics.

Critically, this is not what happens in general and mutualism can readily evolve in the model with nonlinear trade-offs. In the revised manuscript, we investigate the effects of linear, convex, and concave trade-offs in the general model. As shown in Supplementary Figure 2, mutualism can evolve via the COA for all trade-off shapes. The coevolution of mutualism requires that the costs associated with host traits underlying attraction (c_H^I) are below a threshold above which there is evolutionary purging of the partner, and that the costs associated with partner traits underlying mutualism (c_F^I) and antagonism (c_F^A) are above a threshold below which there is evolutionary purging or the antagonism persists despite co-option. These results highlight that while the shape of the trade-offs plays a critical role in coevolutionary dynamics, the coevolution of mutualism can occur regardless of the shape of the trade-off (see Methods lines 406-417).

REVIEWERS' COMMENTS

Reviewer #1 (Remarks to the Author):

I have only two minor comments (plus a few suggestions).

L94-L118 & L454-L483: This seems to be a complicated way to explain that an allee effect arises as soon as $b > 0$. I find the current explanation difficult to follow and I urge the authors to reformulate this part of the ms to clarify their results. It is currently still unclear why the interaction breakdown boundary changes as soon as $b > 0$, since eq 1 does not change for $b > 0$. I think the reason that it is confusing, is that initially the interaction breakdown is calculated by calculating the point where the insect can invade while rare (eq 1), while in case of $b > 0$, the interaction boundary is calculated by finding the condition for an internal equilibrium (eq 2b). A bifurcation graph might maybe help? E.g., showing the plant & insect densities as a function of the defense ($h_{max} - h_i$) both in the absence and in the presence of pollination benefits. Also, it might be worth to mention that in case the insect disappears (for whatever reason), it might be difficult for the insect to invade again due to the bistability.

How exactly are the alternative larval host plant and alternative nectar plant modelled? The equations (6a,b,c), don't specify what happens in case a plant is only a nectar plant/only a host plant. Based on the tables in the sup mat, I guess that for the alternative host plant it is assumed that pollination benefits can't evolve, please specify this somewhere in the text. Is an additional nectar plant only increasing parameter epsilon, or are the dynamics of this plant also modelled? In the first case, please state this somewhere, since the text (e.g., L429-430) suggest that the alternative nectar plant is explicitly taken into account. In the second case, the parameters of this plant are missing in the manuscript.

Minor issues/typos

Thank you for including the video, it would be nice to make videos for all figures.

I like the new method, where the authors start with explaining the general model. However, notation changes in the *M. sexta* model, e.g., I is changed to V , A to H . This makes it a bit confusing and I think it would be better to keep the notation as similar as possible between the general and specific model.

Why do eq 3c and 8c differ from each other? I guess the minus before parameter A' should not be there.

The authors changed r_i to b_i , but at some places in the text this hasn't happen yet (eq 7a for example, also in supplementary table 3).

L461: should it not be > 1 ?

In the supplementary table 3, the values of the coESS seems to be slightly off. E.g., for the final v_d in d. wright only, the value is 5.4, but if I fill in all parameters in the equations, I get 5.3.

Supplementary table 1: the superscript should be 'a', not 'c'.

Reviewer #2 (Remarks to the Author):

Dear authors and editors,

I was Reviewer 2. I like the new version of the manuscript. The improvements are excellent. The changes and comments in the rebuttal letter I feel adequately address my own and other reviewers' comments. I am looking forward to seeing this in print soon.

I think there may be one typo, or perhaps I misunderstood. I do not need to see any change the authors make if the authors agree there is a typo. I also do not need any explanation of how I have misunderstood, if that's the case.

Line 154 "Comparing the predicted coESS with independent data provides a strong validation of the model – at least of the selection gradients on attraction and defense around the contemporary state." This line seems to imply that selection gradients were measured empirically, but having read the whole MS, my understanding is that it is the coESS and empirically measured current trait values that are compared?

Sincerely,
R2

Reviewer #4 (Remarks to the Author):

The modifications in the structure of the paper made it more readable. Most importantly the authors additionally presented results and mathematical argumentation convinced me that the model serves robust results and thus their conclusions are reliable.

All in all I think that the authors considered all of my critical points accordingly.

Response to Reviewers

Below we respond to each comment by the four Referees. We fully reproduce all comments in black italic text, paired with our responses in blue text. References to specific lines in the main text are highlighted.

Reviewer #1

I have only two minor comments (plus a few suggestions).

L94-L118 & L454-L483: This seems to be a complicated way to explain that an Allee effect arises as soon as $b > 0$. I find the current explanation difficult to follow and I urge the authors to reformulate this part of the ms to clarify their results. It is currently still unclear why the interaction breakdown boundary changes as soon as $b > 0$, since eq 1 does not change for $b > 0$. I think the reason that it is confusing, is that initially the interaction breakdown is calculated by calculating the point where the insect can invade while rare (eq 1), while in case of $b > 0$, the interaction boundary is calculated by finding the condition for an internal equilibrium (eq 2b). A bifurcation graph might maybe help? E.g., showing the plant & insect densities as a function of the defense (h_{max-hi}) both in the absence and in the presence of pollination benefits. Also, it might be worth to mention that in case the insect disappears (for whatever reason), it might be difficult for the insect to invade again due to the bistability.

We thank the Reviewer for this helpful comment. The Reviewer is correct that an Allee effect arises when $b_i > 0$; however, the interaction breakdown boundary is still essential because it determines the boundary above which the insect can persist (provided its density is above the Allee threshold) as a net mutualist of the plant (green region of Fig. 2) and below which the insect always goes extinct (white region of Fig. 2). In the pollination-herbivory system, Equation 1 gives the invasion criterion above which the insect can always increase from low density and persist as a net antagonist of the plant despite being co-opted as a pollinator. The interaction breakdown boundary (approximated by Eq. 3) changes because pollination benefits, b_i , effectively allow the insect to persist as a net mutualist by buoying plant density and thereby lowering the interaction breakdown boundary. Within this mutualistic region, however, the insect cannot invade from very low density because, when the insect is rare, it cannot buoy plant density sufficiently to maintain a positive per capita growth rate (mathematically, the insect's invasion criterion – Eq. 1 – cannot hold when the condition for mutualism – Eq. 2 – is satisfied). The mutualistic region is thus characterized by bistability and an Allee effect. Following the Reviewer's comment, we have revised the main text (lines 114-127) and Methods (lines 516-531) to better show how Equation 3 arises and its consequences for the ecological outcomes of the interactions. We have also created Supplementary Figure 1, which shows bifurcation plots for the one-plant species communities as suggested by the Reviewer.

How exactly are the alternative larval host plant and alternative nectar plant modelled? The equations (6a,b,c), don't specify what happens in case a plant is only a nectar plant/only a host plant. Based on the tables in the sup mat, I guess that for the alternative host plant it is assumed that pollination benefits can't evolve, please specify this somewhere in the text. Is an additional nectar plant only increasing parameter epsilon, or are the dynamics of this plant also modelled? In the first case, please state this somewhere, since the text (e.g., L429-430) suggest that the alternative nectar plant is explicitly taken into account. In the second case, the parameters of this plant are missing in the manuscript.

We specify (in lines 482-484) that Equation 9a also gives the dynamics of the alternative larval host plant, which can coevolve attraction and defense. The alternative nectar source is incorporated within the model via ϵ .

Minor issues/typos

Thank you for including the video, it would be nice to make videos for all figures.

We have made videos for all of the figure panels.

*I like the new method, where the authors start with explaining the general model. However, notation changes in the *M. sexta* model, e.g., *I* is changed to *V*, *A* to *H*. This makes it a bit confusing and I think it would be better to keep the notation as similar as possible between the general and specific model.*

We agree and have changed all parameters in the general model so that they correspond with those in the pollination-herbivory model.

*Why do eq 3c and 8c differ from each other? I guess the minus before parameter *A'* should not be there.*

Given our changes in the parameters of the general model in response to the previous comment, these equations would be identically. We have removed the second version of these equations in the revised manuscript.

The authors changed r_i to b_i , but at some places in the text this hasn't happen yet (eq 7a for example, also in supplementary table 3).

We thank the Reviewer for identifying these mistakes.

L461: should it not be > 1 ?

Again, we thank the Reviewer for carefully reading the manuscript and identifying these small mistakes.

*In the supplementary table 3, the values of the coESS seems to be slightly off. E.g., for the final v_d in *D. wrightii* only, the value is 5.4, but if I fill in all parameters in the equations, I get 5.3.*

We have double-checked the values in Supplementary Table 3, which can also be determined in the code that we have provided with the manuscript (Johnson_et_al_codes.zip). In this case, the final value of v_w in the *D. wrightii* one-plant species community is 5.36, which we have rounded to 5.4 in Supplementary Table 3. We greatly appreciate that the Reviewer took the time to carefully check these values.

Supplementary table 1: the superscript should be 'a', not 'c'.

We have corrected this error.

Reviewer #2

Dear authors and editors,

I was Reviewer 2. I like the new version of the manuscript. The improvements are excellent. The changes and comments in the rebuttal letter I feel adequately address my own and other reviewers' comments. I am looking forward to seeing this in print soon.

I think there may be one typo, or perhaps I misunderstood. I do not need to see any change the authors make if the authors agree there is a typo. I also do not need any explanation of how I have misunderstood, if that's the case.

Line 154 "Comparing the predicted coESS with independent data provides a strong validation of the model – at least of the selection gradients on attraction and defense around the contemporary state." This line seems to imply that selection gradients were measured empirically, but having read the whole MS, my understanding is that it is the coESS and empirically measured current trait values that are compared?

The Reviewer is correct that the values of the coESS were validated with empirically-measured current trait values. The selection gradients, however, are not measured empirically – they are quantified via the parameterized model and yield the coESSs (where they intersect). The point of this line in the manuscript is that comparing the coESSs predicted by the model with empirical data provides a validation of the model selection gradients around the contemporary state (contemporary coESSs).

Reviewer #4

The modifications in the structure of the paper made it more readable. Most importantly the authors additionally presented results and mathematical argumentation convinced me that the model serves robust results and thus their conclusions are reliable.

All in all I think that the authors considered all of my critical points accordingly.

We thank the Reviewer for all of the helpful comments on previous versions of the manuscript.